# The muscle-enriched myokine Musclin impairs beige fat thermogenesis and systemic energy homeostasis via Tfr1/PKA signaling in male mice

Lu Jin [1,2,3,14], Shuang Han [1,3,4,5,14], Xue Lv [1,3,5,14], Xiaofei Li [6,14], Ziyin Zhang [1,3,5], Henry Kuang [7], Zhimin Chen [7], Cheng-an Lv [1], Wei Peng [2], Zhuoying Yang [1,3,5], Miqi Yang [1,3,5], Lin Mi [7], Tongyu Liu [7], Shengshan Ma [6], Xinyuan Qiu [8], Qintao Wang [1,3,5], Xiaowen Pan [9], Pengfei Shan [9], Yu Feng [10], Jin Li [11], Fudi Wang [11], Liwei Xie [12], Xuyun Zhao [13], Jun-Fen Fu [2] ✉, Jiandie D. Lin [7] & Zhuo-Xian Meng [1,3,4,5] ✉

Skeletal muscle and thermogenic adipose tissue are both critical for the maintenance of body temperature in mammals. However, whether these two tissues are interconnected to modulate thermogenesis and metabolic homeostasis in response to thermal stress remains inconclusive. Here, we report that human and mouse obesity is associated with elevated Musclin levels in both muscle and circulation. Intriguingly, muscle expression of Musclin is markedly increased or decreased when the male mice are housed in thermoneutral or chronic cool conditions, respectively. Beige fat is then identified as the primary site of Musclin action. Muscle-transgenic or AAV-mediated overexpression of Musclin attenuates beige fat thermogenesis, thereby exacerbating diet-induced obesity and metabolic disorders in male mice. Conversely, Musclin inactivation by muscle-specific ablation or neutralizing antibody treatment promotes beige fat thermogenesis and improves metabolic homeostasis in male mice. Mechanistically, Musclin binds to transferrin receptor 1 (Tfr1) and antagonizes Tfr1-mediated cAMP/PKA-dependent thermogenic induction in beige adipocytes. This work defines the temperature-sensitive myokine Musclin as a negative regulator of adipose thermogenesis that exacerbates the deterioration of metabolic health in obese male mice and thus provides a framework for the therapeutic targeting of this endocrine pathway.

Skeletal muscle comprises ~40% of the human body's total weight, and is responsible for up to 80% of postprandial glucose disposal. Our recent study demonstrated that skeletal muscle glucose-sensing plays an essential role in controlling muscle glucose uptake and systemic energy homeostasis[1]. Impairment of skeletal muscle insulin action and glucose sensing is recognized as an early hallmark of obesity-associated metabolic disorders[2–4]. In addition to serving as an important exercise and metabolic tissue, skeletal muscle has recently been identified as a secretory organ, producing and releasing 'myokines' to regulate systemic energy metabolism and glucose homeostasis via autocrine, paracrine, and endocrine mechanisms[5,6]. More than 600 potential myokines have now been

identified using label-free quantitative proteomic approaches[7]. Since skeletal muscle is known to play important roles in systemic homeostasis, much attention has been focused on the identification of beneficial myokines such as IL6, BDNF, irisin, IL-15, detailing their positive effects[5,8,9]. However, far less is known about any potential negative metabolic regulators that are secreted from skeletal muscle.

Adipocytes in mammals can be classified into two categories: energy-storing white adipocytes and energy-dissipating thermogenic adipocytes. The latter can be further divided into two types, namely, the classical brown adipocytes and the newly identified beige/brite adipocytes[10–13]. Despite the remarkable morphological, biochemical, and transcriptional similarities between brown and beige adipocytes, recent studies have revealed distinguishing features and mechanisms underlying the developmental origins and functional regulation of these two cell types[14,15]. Brown adipocytes convert chemical energy into heat mainly through uncoupling protein 1 (Ucp1)-mediated uncoupled respiration[16]. Beige adipocytes are an inducible form of thermogenic adipocyte that arises within white adipose tissue in response to cold exposure. Recent studies have suggested that both Ucp1-dependent and -independent mechanisms are involved in beige adipocyte thermogenesis[15,17]. Thermogenic brown and beige adipocytes have been extensively studied over the last decade due to their therapeutic potential for type 2 diabetes and other prevalent metabolic disorders. While adrenergic signaling secondary to the sympathetic outflow is undoubtedly the critical activator of brown and beige fat thermogenesis, recent studies have identified several other local and systemic secreted factors that activate the brown and beige fat thermogenic metabolism and improve systemic energy homeostasis[18,19]. Ironically, it seems that evolutionarily selection pressure may have contributed to the enrichment of mechanisms in humans that inhibit energy expenditure and facilitate increased fat storage. Whilst historically this may have facilitated survival during prolonged periods of hunger and cold, in today's modern life, often characterized by overnutrition and far lower levels of physical activity, such energy-saving mechanisms may have turned against us, being potentially the prime contributors towards obesity and associated metabolic diseases[20]. Although a recent study has identified the hepatokine Tsukushi (TSK) as a negative regulator of energy expenditure by suppressing brown fat sympathetic innervation and thermogenesis[21], any specific and naturally occurring endocrine secreted factors that can directly inhibit both brown and beige fat thermogenesis and energy expenditure remain to be identified.

Here, we report the identification and characterization of Musclin, a temperature-regulated myokine, as a critical negative regulator of beige fat thermogenesis and systemic glucose homeostasis. The skeletal muscle expression of Musclin is markedly increased in obese human subjects and mice, leading to elevation of Musclin levels in circulation. A SEAP binding assay revealed subcutaneous white adipose tissue (scWAT) as the primary target of Musclin. Interestingly, increased circulating Musclin, via either muscle-specific transgenic or AAV-mediated overexpression attenuates beige fat thermogenic metabolism, rendering the mice more susceptible to HFD-induced obesity and metabolic disorders. Conversely, inhibition of Musclin action by muscle-specific ablation or anti-Musclin neutralizing antibody treatment promotes beige fat thermogenesis and improves systemic glucose homeostasis. Mechanistically, Musclin directly binds to transferrin receptor 1 (Tfr1) on the plasma membrane of beige adipocytes and antagonizes cAMP/PKA-dependent thermogenic induction in a cell-autonomous manner. This work identifies Musclin as a critical negative regulator of beige fat thermogenesis that acts to fine-tune systemic energy balance under both physiological and pathophysiological conditions.

## Results

### Identification of Musclin as a risk factor for human obesity

We firstly developed a transcriptomic screening strategy to uncover any novel muscle-enriched secreted factors that may be important in the pathogenesis of obesity and its associated metabolic disorders (Fig. 1a). Initially, RNA sequencing (RNA-Seq) on skeletal muscles from 10 human subjects with a BMI of <24 and another 10 human subjects with a BMI of ≥24 was performed. We then analyzed the mRNA expression of genes encoding secreted protein based on our previously described secretome gene list[22]. We identified a set of 39 genes encoding putative secreted proteins that were significantly altered between these two groups (Cutoff: $p$-value < 0.05 & |Log$_2$ (fold change)| > 0.3). Between these representative 10 upregulated and 10 downregulated candidate factors, *Musclin*, also known as *Osteocrin*[23,24], exhibited significantly higher expression in subjects with BMI ≥ 24 (Fig. 1b) and highly restricted expression in skeletal muscle (Fig. 1c). The predominant expression of *Musclin* in skeletal muscle was also validated through qPCR analysis across different mouse tissues (Fig. 1d), in accordance with a previous study[24]. Western blotting analysis of conditioned medium (CM) from transiently transfected HEK293T cells confirmed that Musclin protein was secreted and readily detectable in the culture medium (Supplementary Fig. 1a). We next examined skeletal muscle samples from 54 human donors, as described in Supplementary Fig. 1b, to validate the correlation of *MUSCLIN* mRNA expression levels with BMI. Intriguingly, the mRNA levels of *MUSCLIN* in human skeletal muscles were positively correlated with BMI ($r = 0.2927$, $p = 0.0317$, Fig. 1e). Accordingly, compared to normal human individuals (BMI < 24), mRNA expression of *MUSCLIN* in skeletal muscles was robustly elevated in overweight (24 ≤ BMI < 28) and obese (BMI ≥ 28) human subjects (Fig. 1f). Further western blotting analysis confirmed that the protein levels of MUSCLIN in skeletal muscle were significantly increased in obese human subjects (Fig. 1g and Supplementary Fig. 1c). As expected, the plasma concentrations of MUSCLIN were significantly higher in obese human subjects (Fig. 1h). Consistent with the enriched expression of Musclin in skeletal muscle, mRNA expression of *Musclin* was markedly induced during the differentiation of C2C12 myotubes in a time-dependent manner (Fig. 1i). We next examined the expression of Musclin in skeletal muscles from obese and diabetic mouse models. Consistent with the observations in humans, *Musclin* mRNA expression was markedly induced in skeletal muscles from high-fat diet (HFD)-induced obese mice, genetically obese (*ob/ob*) mice, and diabetic (*db/db*) mice (Fig. 1j). Accordingly, the plasma concentrations of Musclin were also significantly elevated in obese and diabetic mice (Fig. 1k). Collectively, these findings demonstrate that muscle expression and circulating levels of Musclin are increased in cases of obesity and diabetes in both humans and mice, and that Musclin might play a pivotal role in the pathogenesis of obesity and its associated metabolic disorders through crosstalk between muscles and other metabolic tissues.

### Muscle expression of Musclin is regulated by environmental temperature and suppresses adipose tissue thermogenesis

Ambient temperature is one of the key factors affecting systemic energy expenditure and homeostasis. Interestingly, we observed that *Musclin* expression in skeletal muscle was dynamically regulated by housing temperature in mice. Both the mRNA and protein levels of Musclin in skeletal muscle were significantly increased when the mice were housed in thermoneutrality at 30°C for 1 week, compared to mice housed at room temperature (RT) (Fig. 2a, **left** and 2b). Conversely, chronic cold acclimation treatment, a highly thermogenic condition, robustly decreased the muscle expression of Musclin in mice (Fig. 2a, **right** and 2c). In accordance, plasma levels of Musclin were significantly increased or decreased in response to thermoneutrality or cold acclimation treatments as compared to their respective control mice (Fig. 2d). These results suggest that skeletal

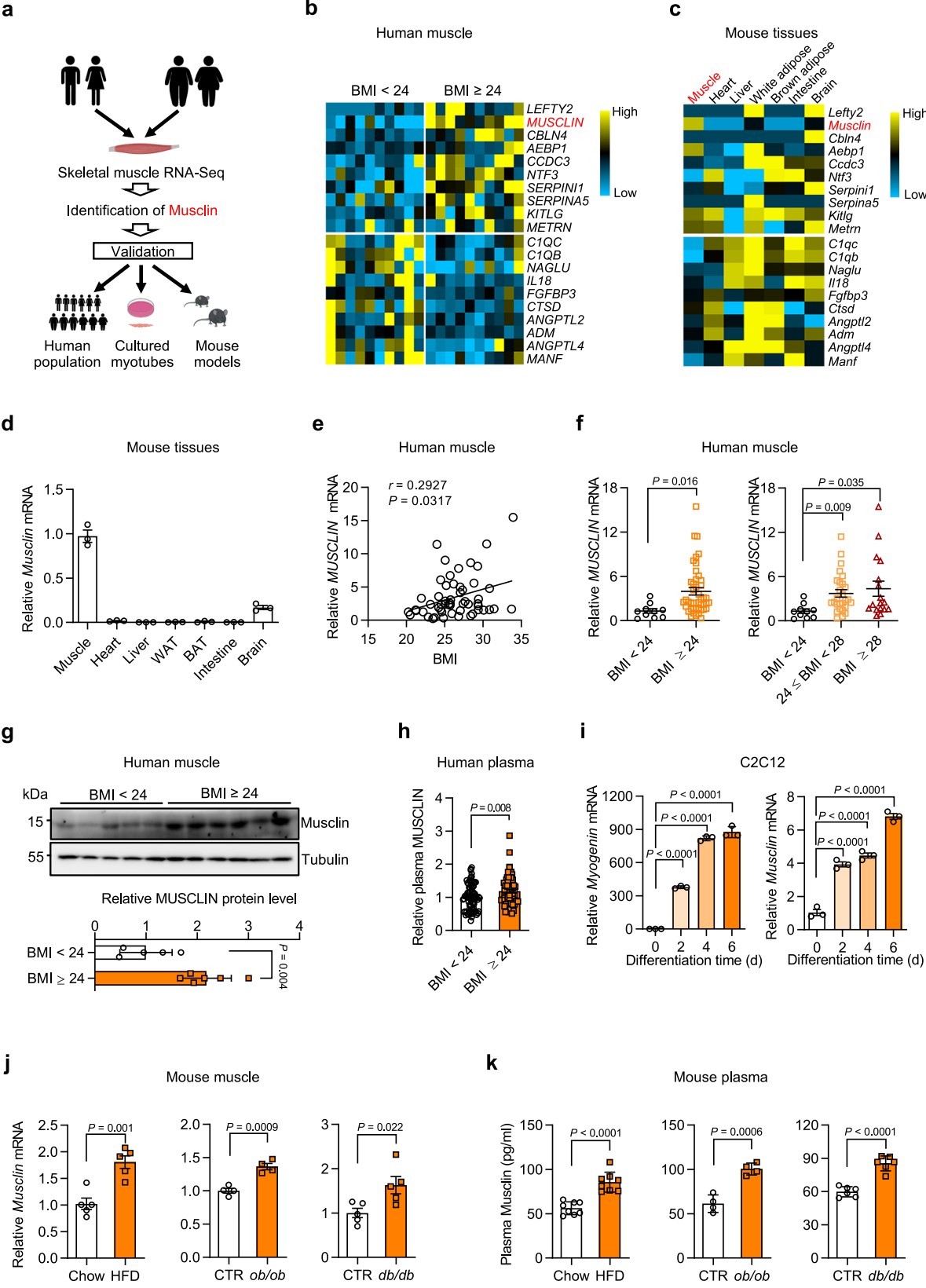

muscle might be actively engaged in the regulation of whole-body thermogenesis and energy homeostasis through Musclin-mediated tissue crosstalk. To test this, we generated a fusion protein of SEAP and Musclin (SEAP-Musclin) and performed binding assays on frozen sections of metabolic tissues to identify its downstream target(s) in vivo. The binding sites in tissues were determined by histochemical staining

for alkaline phosphatase activity. Interestingly, the strongest binding activities were observed on tissue sections of inguinal white adipose tissue (iWAT), a subcutaneous white adipose tissue (scWAT) depot containing thermogenic beige adipocytes, accompanied by much weaker binding signals on epididymal white adipose tissue (eWAT), interscapular brown adipose tissue (BAT) and liver sections, while the

**Fig. 1 | Identification of muscle secreted protein Musclin as a potential risk factor for metabolic diseases.** All mice used here are male. **a** Schematic representation of muscle secreted protein screening. **b** Heatmap of expression of 10 upregulated and 10 downregulated secreted protein-encoding genes in muscles from subjects with BMI ≥ 24 and BMI < 24 (Cutoff: $P < 0.05$, |Log$_2$ (fold change)| > 0.3 & average reads count > 30). BMI, body mass index. **c** Expression levels of genes listed in (**b**) in mouse tissues. **d** qPCR analysis of *Musclin* expression (3 months old, $n = 3$). **e** Association of relative *MUSCLIN* mRNA levels to human BMI ($n = 54$), *Spearman* correlation analysis with 95% confidence interval. The statistical test was two-sided with α = 0.05. **f** qPCR analysis of *MUSCLIN* gene expression. BMI < 24, $n = 10$; 24 ≤ BMI < 28, $n = 27$; BMI ≥ 28, $n = 17$. **g** Immunoblots of human muscle lysates (top) and quantification of Musclin levels (bottom). BMI < 24, $n = 5$; BMI ≥ 24, $n = 6$. **h** Relative human plasma Musclin levels. BMI < 24, $n = 83$; BMI ≥ 24, $n = 61$. **i** qPCR analysis of *Myogenin* and *Musclin* gene expression in C2C12 during differentiation. $n = 3$ biologically independent cell samples. **j** qPCR analysis of *Musclin* gene expression. Chow vs HFD, $n = 5$ (HFD feeding at 3-month-old, continued for 4 months); CTR vs *ob/ob*, $n = 4$ (4 months old, chow diet); CTR vs *db/db*, $n = 5$ (5 months old, chow diet). **k** Radioimmunoassay (RIA) of plasma Musclin levels. Chow vs HFD, $n = 8$ (7 months old); CTR vs *ob/ob*, $n = 4$ (4 months old); CTR vs *db/db*, $n = 6$ (5 months old). **d**, **f**–**h**, **j**, **k**, data represent mean ± SEM; **i** data represent mean ± SD. **d-h**, **j-k**, $n$ represents biologically independent animals/human samples. **f**–**h**, **j**, **k**, two-tailed unpaired Student's *t*-test; **i** one-way ANOVA with Tukey's multiple comparisons. **b**, **c** Differential expression analysis of RNA-Seq data was performed using *Deseq2* package (*P*-value by Wald test). Experiments in **d**–**g** and **i**–**k** were repeated independently three times with similar results. Source data are provided as a Source Data file.

binding activities on muscle and pancreas tissue sections were almost undetectable (Fig. 2e). These data suggest that adipose tissues, especially scWAT, might serve as the primary targets of Musclin signaling to exert its effect on thermogenesis.

To investigate the physiological role of Musclin in maintaining whole-body energy homeostasis, we generated muscle-specific Musclin transgenic mice driven by the muscle creatine kinase promoter (MCK-Musclin). Successful elevation of Musclin protein levels in skeletal muscle and the circulation of MCK-Musclin mice compared to wild-type (WT) controls was validated by western blotting (Fig. 2f) and radioimmunoassay (RIA) analysis (Fig. 2g), respectively. Interestingly, MCK-Musclin mice gained a slightly but significantly higher body weight than WT mice even on a standard chow diet, while food intake and treadmill running performance remained indistinguishable between the two groups (Fig. 2h and Supplementary Fig. 2a, b). Further tissue weight and morphological analysis of the major metabolic tissues revealed that tissue weight of iWAT was significantly increased in MCK-Musclin mice (Fig. 2i, j and Supplementary Fig. 2c, d). No significant difference in tissue weight, appearance, or histology of skeletal muscle was observed between control and MCK-Musclin mice (Supplementary Fig. 2c–e). These studies further demonstrate that muscle-specific transgenic expression of Musclin may influence systemic energy expenditure through muscle-adipose tissue crosstalk, while having a modest effect on the development, metabolism, and exercise performance of skeletal muscle per se.

To directly validate the role of Musclin in thermogenesis, we subjected control and MCK-Musclin mice to cold challenge or treatment with CL316,243, a potent selective β3-adrenoceptor agonist, to stimulate thermogenesis in adipose tissues. Compared to WT control mice, upon acute cold exposure at 4°C, MCK-Musclin mice exhibited impaired thermogenesis and were unable to maintain normal body temperature (Fig. 2k, l). A more robust difference in body temperature was observed between the two groups when the mice were housed at 8°C (Fig. 2m). Consistently, body surface temperature was significantly lower in MCK-Musclin mice than controls following intraperitoneal injection of CL316,243, as visualized by infrared imaging (Fig. 2n). Accordingly, MCK-Musclin mice tended to have higher blood glucose following cold exposure, whereas plasma insulin levels remained similar between the two groups (Fig. 2o, p). Taken together, these results confirm the important role of Musclin in regulating adipose tissue thermogenesis and systemic energy homeostasis.

## Musclin acts cell-autonomously to inhibit thermogenic metabolism in beige adipocytes

To further evaluate the effect of Musclin on the thermogenic metabolism in adipose tissues, we performed RNA-Seq analysis of BAT and iWAT from control and MCK-Musclin mice following cold exposure at 8°C for 1 week (Fig. 3a, b and Supplementary Fig. 3b). Consistent with the results from phenotypic studies, a large set of genes (Set I, 350 genes) were substantially downregulated in iWAT from MCK-Musclin mice as compared to controls, and these genes are mainly involved in

fatty acid metabolism, cellular respiration, glucose metabolism and adaptive thermogenesis (Fig. 3b, c). By contrast, a set of genes (Set II, 561 genes) mainly associated with immune response were upregulated (Fig. 3b, c). Since chronic cold exposure is known to induce beiging gene program and thermogenesis in subcutaneous iWAT, we performed overlapping analysis on the differentially expressed genes (DEGs) in iWAT from comparisons between MCK-Musclin and control mice, and DEGs in iWAT from comparisons between cold-acclimated mice and control mice according to RNA-Seq datasets (Fig. 3d). A set (142 out of 1179) of the genes upregulated in the iWAT of WT mice in response to cold acclimation were reversed by Musclin transgenic expression, which are mainly energy metabolism-related (Fig. 3d). Downregulation of mRNA and protein levels of genes related to lipid/ glucose metabolism and thermogenesis in iWAT of MCK-Musclin mice was further confirmed by qPCR and western blotting analysis (Fig. 3e, f, Supplementary Fig. 3a). However, despite the weak binding of SEAP-Musclin fusion protein to brown fat tissue sections (Fig. 2e), we did not observe any significant difference in canonical thermogenic gene programming through transcriptomic analysis of BAT from control and MCK-Musclin mice following 1 week of cold exposure at 8°C (Supplementary Fig. 3b, c). The creatine metabolism was recently uncovered to play an important role in thermogenesis and whole-body energy expenditure in both beige and brown fat independent of Ucp1-mediated canonical thermogenic mechanisms[25–27]. However, mRNA levels of those genes in either BAT or iWAT remained similar between the two groups (Supplementary Fig. 3d). Moreover, the expression of lipid and glucose metabolism related genes in eWAT and quadriceps skeletal muscle remains comparable between the two groups (Supplementary Fig. 3e, f), as consistent with the mild differences observed in appearance or tissue weight of eWAT and skeletal muscle between MCK-Musclin and control mice. These data indicate that Musclin selectively suppresses iWAT metabolic and thermogenic gene reprogramming upon cold challenge, leading to attenuation of beige fat thermogenesis.

To further examine whether Musclin has a direct effect on beige adipocytes, C3H10T1/2-derived adipocytes were treated with Musclin fused with the crystallizable fragment (Fc) domain of immunoglobulin G (IgG) on its C terminus (Musclin-Fc), purified from CM of Expi293F cells transfected with a plasmid expressing Musclin-Fc (Supplementary Fig. 3g, h) or fused with 6 x his tag on its C terminus (Musclin-His). Successful differentiation of C3H10T1/2 cells was confirmed by the accumulation of multiple multilocular lipid droplets and by high expression levels of genes related to adipogenesis and thermogenesis. Similar to the observations in vivo, Musclin treatment (0.8 μM) markedly downregulated a spectrum of lipid catabolism-related genes including *Atgl*, *Hsl*, *Mgll*, *Lpl*, *Acat1*, and *Acaa1b*, as well as the genes involved in glucose metabolism and thermogenesis such as *Aldoa*, *Ldhb*, *Pdhb*, *Prdm16*, and *Ucp1* (Fig. 3g). These results indicate that Musclin suppresses the transcriptional program related to thermogenic metabolism in beige adipocytes in a cell-autonomous manner. In accordance, seahorse studies revealed that both basal and forskolin (FSK, an adenylate cyclase agonist)-induced cell respiration rates were

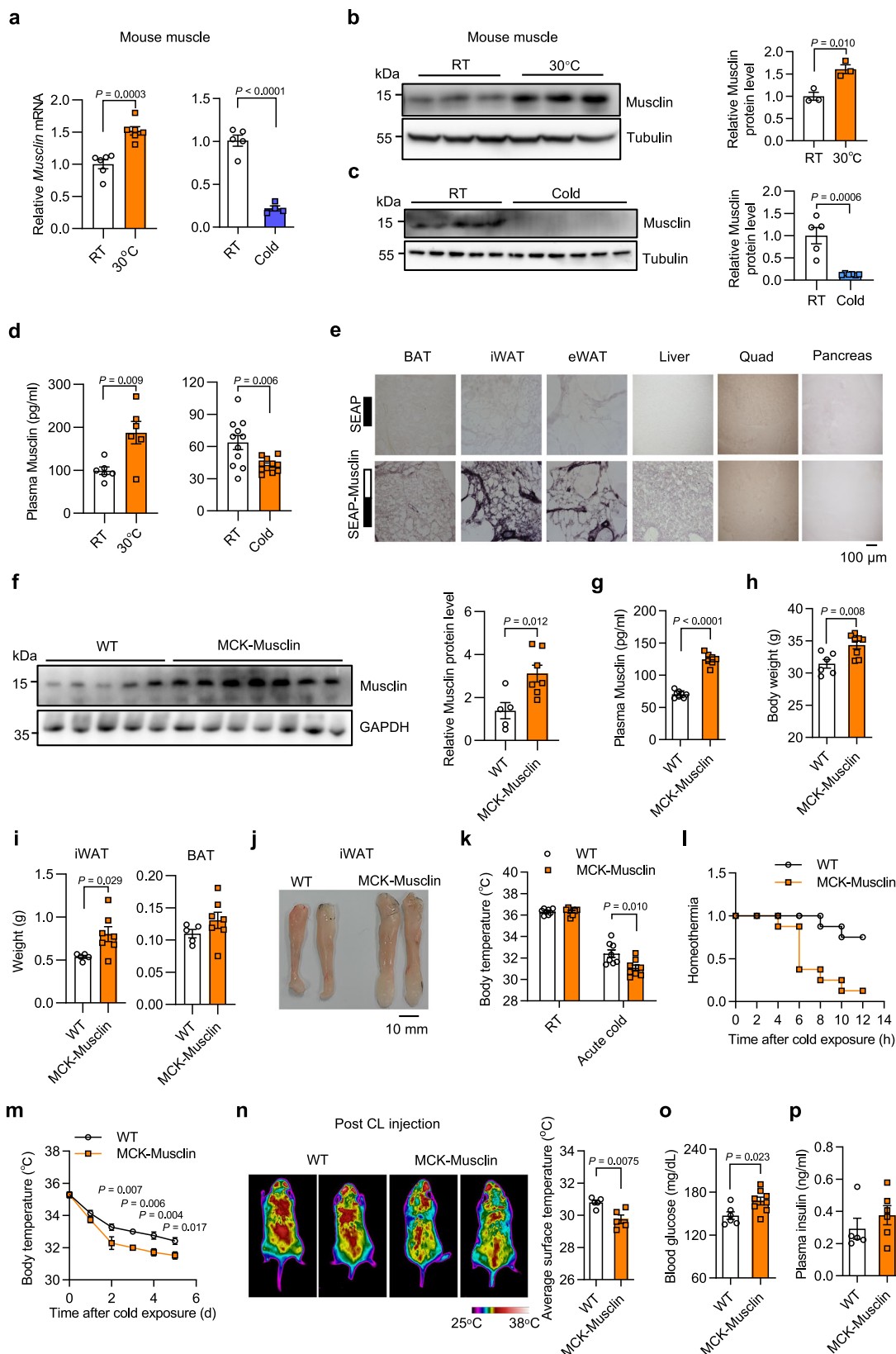

robustly attenuated by Musclin (Fig. 3h, **left**). Further quantitative analyses confirmed the inhibitory effects of Musclin on basal, proton leaking associated, and maximal respiration rates in the presence or absence of FSK stimulation (Fig. 3h, **right**). Taken together, these data suggest that Musclin suppresses thermogenic metabolism in beige adipocytes in a cell-autonomous manner.

## Musclin exacerbates diet-induced obesity and metabolic defects

Since adipose tissue thermogenesis had been established as a pivotal determinant in the progression of diet-induced obesity and its associated metabolic syndrome, we next subjected the MCK-Musclin and control mice to HFD feeding. As expected, MCK-Musclin mice gained significantly more body weight upon HFD feeding than control mice

**Fig. 2 | Transgenic overexpression of Musclin inhibits adipose tissue thermogenesis.** All mice used here are male and fed with chow diet. **a** qPCR analysis of *Musclin* mRNA in muscles from indicated mice (3 months old). Room temperature (RT) vs 30 °C (1 week), $n = 6$; RT vs Cold, $n = 5$ vs 4. **b** Immunoblots of muscle lysates (left, 3 months old) and quantification of Musclin protein levels (right). $n = 3$. **c** Immunoblots of muscle lysates (left, 3.5 months old), and quantification of Musclin protein levels (right). RT vs Cold, $n = 5$ vs 6. **d** Plasma Musclin levels. RT vs 30 °C, $n = 6$ (3 months old); RT vs Cold, $n = 11$ (5 months old). **e** Binding assay on mouse tissue sections using SEAP or SEAP-Musclin fusion proteins. SEAP, secreted alkaline phosphatase; iWAT, inguinal white adipose tissue; eWAT, epididimal white adipose tissue. Scale bar: 100 μm. **f** Immunoblots of muscle lysates (left), and quantification of Musclin protein levels (right). WT vs MCK-Musclin, $n = 5$ vs 7. MCK, muscle creatine kinase. **g** RIA of plasma Musclin levels. WT vs MCK-Musclin, $n = 8$ vs 7. **h** Body weight (6 months old). WT vs MCK-Musclin, $n = 6$ vs 8. **i** iWAT and BAT weight (7 months old). WT vs MCK-Musclin, $n = 5$ vs 7. **j** Representative images of iWAT (7 months old). **k** Core body temperature before and 6 h following cold at 4 °C (2.5 months old), $n = 8$. **l** Ratio of homeothermic mice during cold at 4 °C (2.5 months old) (Homeothermia is defined as the ratio of mice with a body temperature above 31 °C), $n = 8$. **m** Core temperature during cold at 8 °C (7 months old). WT vs MCK-Musclin, $n = 7$ vs 6. **n** Representative infrared images (left) and the average body surface temperature (right) of mice following 3 h of CL316,243 (CL) injection (4.5 months old). $n = 5$. **o** Blood glucose following cold at 8 °C for 2 days (6.5 months old). WT vs MCK-Musclin, $n = 6$ vs 8. **p** Plasma insulin levels (5 months old). WT vs MCK-Musclin, $n = 5$ vs 6. Data represent mean ± SEM ($n$ represents biologically independent animals). **a**–**d**, **f**–**i**, **k**, **n**–**p** Two-tailed unpaired Student's *t*-test; **m** two-way ANOVA with Sidak's multiple comparisons. All experiments here were repeated independently three times with similar results. Source data are provided as a Source Data file.

(Fig. 4a). In addition, blood glucose levels were significantly higher in HFD-fed MCK-Musclin mice following overnight starvation (Fig. 4b), while the plasma insulin level remained comparable between the two groups (Fig. 4c). Further glucose tolerance test (GTT) and insulin tolerance test (ITT) demonstrated that the elevation of circulating Musclin levels in MCK-Musclin mice impaired both glucose tolerance and insulin sensitivity (Fig. 4d, e). Plasma triglycerides (TG) and non-esterified fatty acid (NEFA) levels remained similar (Supplementary Fig. 4a). Body composition analysis revealed that elevation of Musclin increased body fat storage, whilst exhibiting no detectable effect on lean mass (Fig. 4f). Specifically, the tissue weight of iWAT, but not eWAT, BAT or liver, was markedly increased in MCK-Musclin mice (Fig. 4g). This further confirmed the role of Musclin in beige fat thermogenesis. Histological analysis revealed a dramatic enlargement of adipocyte size in iWAT and BAT, but not in eWAT or liver, from MCK-Musclin mice compared to control mice (Fig. 4h, i and Supplementary Fig. 4b). Taken together, these results demonstrate that muscle-specific transgenic overexpression of Musclin exacerbates diet-induced obesity and metabolic dysfunction.

To rule out the possibility that MCK-Musclin may influence energy metabolism and thermogenesis in the skeletal muscle itself, thereby contributing to the aforementioned impaired thermogenesis and energy expenditure, we employed adeno-associated viruses (AAVs) expressing Musclin (AAV-Musclin) or GFP (AAV-GFP, used as the control) to elevate circulating Musclin levels through tail vein injection (Fig. 5a). Musclin was successfully expressed in the liver from AAV-Musclin injected mice, leading to a significant elevation of Musclin levels in plasma (Fig. 5b, c). As expected, AAV-Musclin injected mice displayed impaired thermogenesis and energy metabolism as evidenced by lower body core and surface body temperature (Fig. 5d, e), as well as significantly higher blood glucose levels in response to cold acclimation compared to AAV-GFP injected mice (Supplementary Fig. 4c). By contrast, hepatic expression of genes involved in glucose and lipid metabolism remained similar between the two groups (Supplementary Fig. 4d). We then subjected the AAV-Musclin transduced and control mice to HFD feeding. Similar to the observations in HFD-fed MCK-Musclin mice, AAV-Musclin injected mice gained significantly higher increases in body weight upon HFD feeding than control mice (Fig. 5f), accompanied by higher fasting blood glucose levels (Fig. 5g), as well as greater impairment of glucose tolerance (Fig. 5i) and insulin sensitivity (Fig. 5j). In contrast, plasma insulin levels remained similar between two groups (Fig. 5h). We next applied the Comprehensive Lab Animal Monitoring System (CLAMS) study to examine the effects of Musclin on non-shivering thermogenesis and systemic energy expenditure. Whole-body metabolic rates were examined either under basal conditions or upon sympathetic nervous activation through intraperitoneal injection of the CL316,243 compound. Compared to control mice, AAV-Musclin mice displayed significantly lower $O_2$ consumption rates and energy expenditure (Fig. 5k and Supplementary Fig. 4g) following CL316,243 treatment. In

contrast, whole-body metabolic rates between the two groups were similar before the CL316,243 injection (Supplementary Fig. 4e, f). Food intake and total locomotor activity remained comparable either before or after CL316,243 injection (Supplementary Fig. 4h–k). Accordingly, iWAT weight from AAV-Musclin injected mice was also higher than that of AAV-GFP injected mice (Fig. 5l). Taken together, these results confirm that the elevation of circulating Musclin aggravates diet-induced metabolic disorders through inhibiting beige fat thermogenesis and whole-body energy expenditure.

Given that skeletal muscle expression and circulating levels of Musclin had been positively associated with BMI and seen to be markedly elevated in obese human subjects compared to lean controls (Fig. 1e–h), it was then speculated that the elevation of Musclin in circulation may also impair thermogenic metabolism in human adipose tissues, and thereby playing an essential role in the development of obesity and its associated metabolic defects. To test this, we harvested scWAT from human subjects with different BMIs for RNA-Seq analysis. The basic information of the recruited human subjects is shown in Supplementary Fig. 5a. We identified two sets of genes that were markedly upregulated or downregulated in scWAT from obese human subjects (BMI ≥ 28) as compared to controls (BMI < 24) (Supplementary Fig. 5b). Further GO analysis revealed that the downregulated genes were mainly enriched in pathways associated with fatty acid metabolism, lipid oxidation, cellular respiration, and temperature homeostasis, while the upregulated genes were mainly related to immune response and inflammation (Supplementary Fig. 5c). Notably, these gene expression profiles were quite similar to those obtained in iWAT RNA-Seq data from the comparison between MCK-Musclin and control mice (Fig. 3b, c). These data suggest an important role of Musclin elevation in the pathogenesis of obesity and its associated metabolic disorders through suppressing subcutaneous beige fat thermogenesis in both mice and humans.

## Musclin acts on Tfr1 to antagonize cAMP/PKA pathway-dependent thermogenic induction in adipocytes

cAMP/PKA signaling is known to play a central role in non-shivering thermogenesis in brown and beige fat. Consistent with the in vivo phenotype and transcriptional regulation, Musclin treatment in beige adipocytes resulted in marked decreases in the phosphorylation levels of PKA substrates (Fig. 6a) and cellular levels of cAMP (Fig. 6b). However, this inhibitory effect could be blocked by pretreatment with 8-Br-cAMP, a cell-permeable cAMP analog (Fig. 6c). These data indicate that Musclin inhibits thermogenic metabolism in beige adipocyte through antagonizing cAMP/PKA signaling. To further identify the potential receptor mediating the effect of Musclin on thermogenic adipocytes, we first conducted a proximity-dependent biotin identification (BioID) assay, developed by Roux and colleagues[28], to obtain the interactome of Musclin on the plasma membrane of beige adipocytes (Fig. 6d). Fully differentiated C3H10T1/2-derived adipocytes were treated with

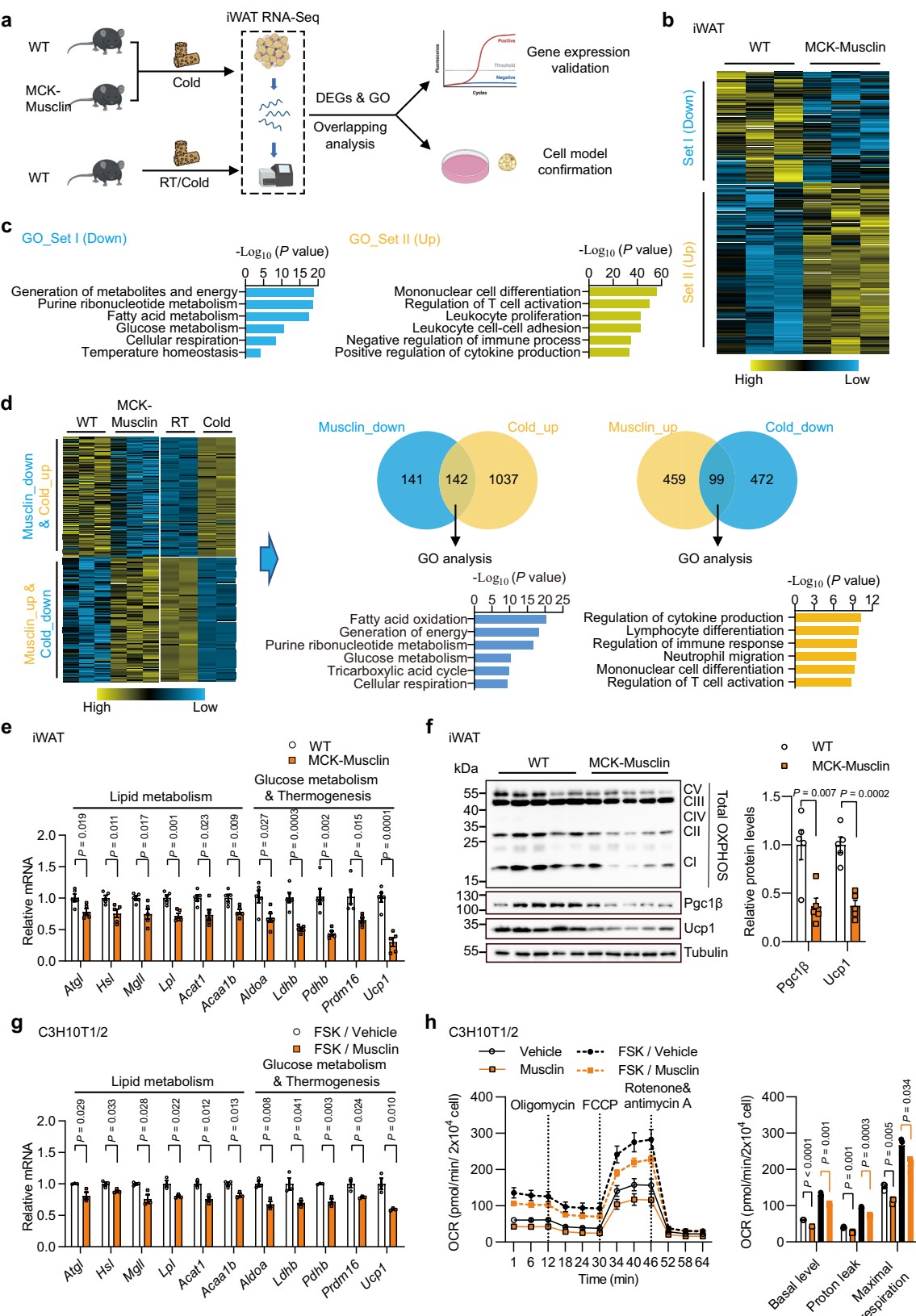

CM from HEK293T cells transfected with pCMV-BirA* or pCMV-Musclin-BirA*, followed by streptavidin pulldown of the cell membrane-associated proteins. After SDS-PAGE gel electrophoresis, the distinct gel bands between the two groups were collected for protein identification by mass spectrometry (M.S.) (Fig. 6e). Among all the proteins obtained from M.S. analysis, proteins localized in the endoplasmic reticulum and mitochondria; residual non-membrane proteins including nucleoproteins and cytoplasmic proteins; and those also present in the control group were all excluded. We finally obtained 11 membrane-associated proteins with enrichment score >2.0 (Supplementary Fig. 6a), among which only transferrin receptor protein 1 (Tfr1), hormone-sensitive lipase (HSL), and P2x purinoceptor 4

**Fig. 3 | Musclin cell-autonomously inhibits adipocyte thermogenic metabolism.** All mice used here are male and fed with chow diet. **a** Schematic of transcriptional profiling analysis in iWAT from chronic cold (8°C, 1 week)-treated WT and MCK-Musclin mice (5 months old) and overlapping analysis between differentially expressed genes (DEGs) in iWAT from MCK-Musclin and cold acclimated male mice as compared to their respective controls (2.5 months old). GO, Gene Ontology. **b** Transcriptomics of iWAT from MCK-Musclin and control mice (Cutoff: $p < 0.05$, |log$_2$ (fold change)| > 0.5). **c** GO analysis of gene sets I and II in (**b**). **d** Overlapping analysis of DEGs in iWAT from indicated mice (Cutoff: $P < 0.05$, |log$_2$ (fold change)| > 0.5). Heatmap, Venn diagram, and GO analysis are shown. **e** qPCR analysis of gene expression in iWAT from MCK-Musclin and control mice as indicated in (**a**). **f** Immunoblots of total lysates of iWAT from chronic cold (8°C, 1 week)-treated mice (left), and quantification of indicated protein levels (right). **e-f** Data represent mean ± SEM ($n = 5$ biologically independent animals per group, 5 months old). **g** qPCR analysis of gene expression in differentiated C3H10T1/2 cells pretreated with vehicle or Musclin-His, followed by forskolin treatment. Data represent mean ± SD ($n = 3$ biologically independent cell samples per group). FSK, forskolin. **h** Oxygen consumption rates (OCRs) of adipocytes with indicated treatments at each timepoint (left) and average basal, proton leak, and maximal respiration levels (right). OCRs were normalized to the cell numbers. Data represent mean ± SD at each timepoint (left panel; $n = 6$ for Vehicle, 5 for Musclin, 6 for FSK/Vehicle, 7 for FSK/Musclin) and mean ± SEM (right panel; each symbol represents the average OCR from one treatment at each detection timepoint, 3 detections for each condition). **e–h** two-tailed unpaired Student's $t$-test. Differential expression analysis of RNA-Seq data was performed using *Deseq2* package (*P*-value by Wald test) in **b–d**. Experiments in **e–h** were repeated independently three times with similar results. Source data are provided as a Source Data file.

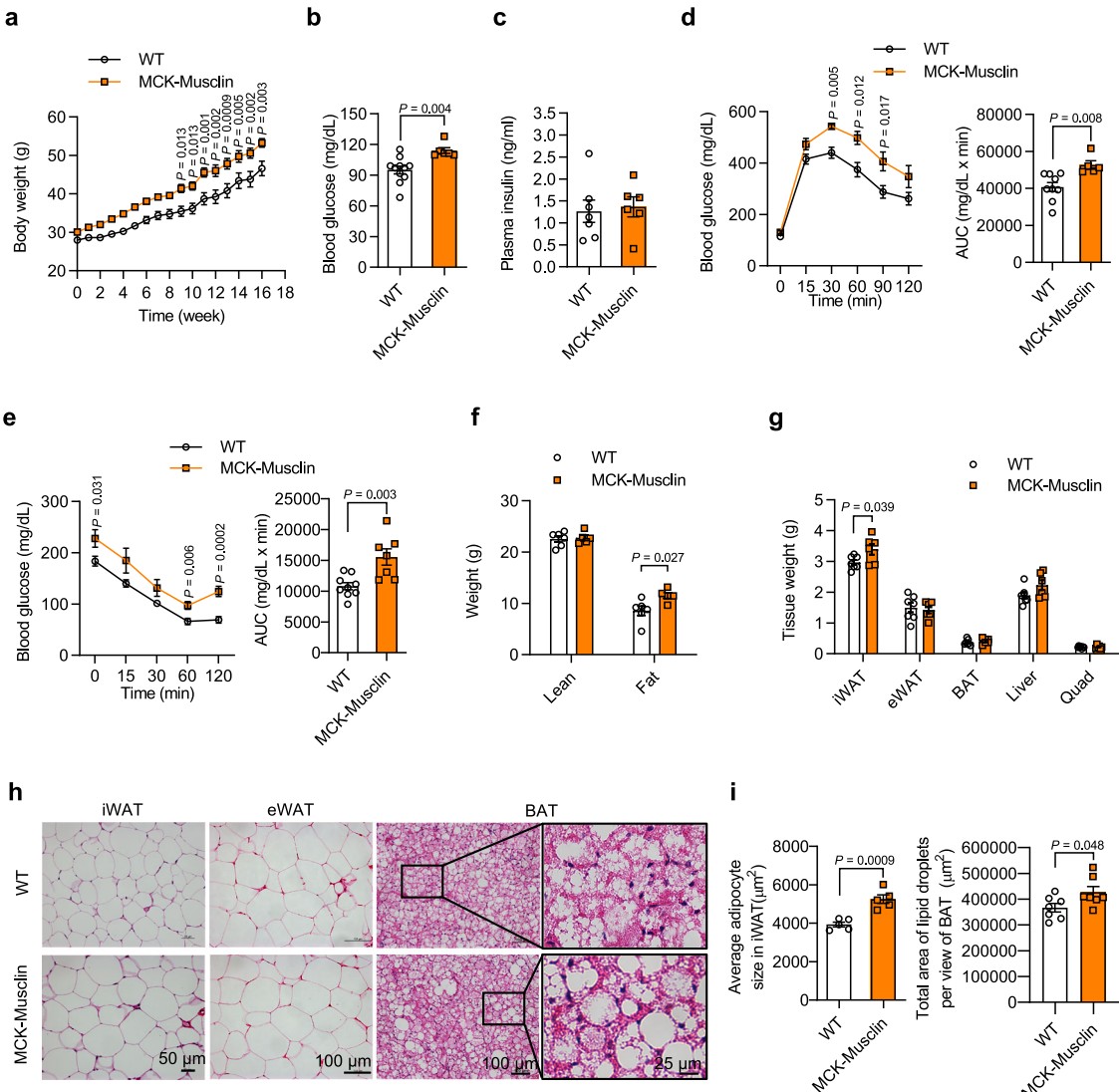

**Fig. 4 | Elevation of muscle-secreted Musclin exacerbates HFD-induced obesity and metabolic dysfunction.** MCK-Musclin and WT male mice were fed with HFD beginning at 3 months old. **a** Body weight of MCK-Musclin and WT mice housed at 16°C. WT vs MCK-Musclin, $n = 8$ vs 7. **b** Fasting blood glucose. WT vs MCK-Musclin, $n = 11$ vs 6. **c** Plasma insulin concentration of mice housed at 16°C. WT vs MCK-Musclin, $n = 7$ vs 6. **d** Glucose tolerance test (left) and the area under curve (AUC, right). WT vs MCK-Musclin, $n = 9$ vs 5. **e** Insulin tolerance test (left) and AUC (right) of mice housed at 16°C. WT vs MCK-Musclin, $n = 9$ vs 7. **f** Lean mass and fat mass after HFD feeding for 2 months at 16°C. WT vs MCK-Musclin, $n = 6$ vs 5. **g** Weight of indicated tissues from mice housed at 16°C. WT vs MCK-Musclin, $n = 7$ vs 6. **h** Representative H&E staining images of iWAT, eWAT, and BAT (WT vs MCK-Musclin, $n = 4$ vs 5). **i** The cell size of iWAT ($n = 5$ per group, left panel) and total lipid droplet area per view in BAT H&E sections ($n = 7$ per group, right panel) as shown in (**h**). **a–g, i** data represent mean ± SEM; $n$ represents biologically independent animals. **a** Two-way ANOVA with Sidak's multiple comparisons. **b–g, i** Two-tailed unpaired Student's $t$-test. Experiments in **a–g** were repeated independently three times, and experiments in **h** and **i** were repeated independently twice with similar results. Source data are provided as a Source Data file.

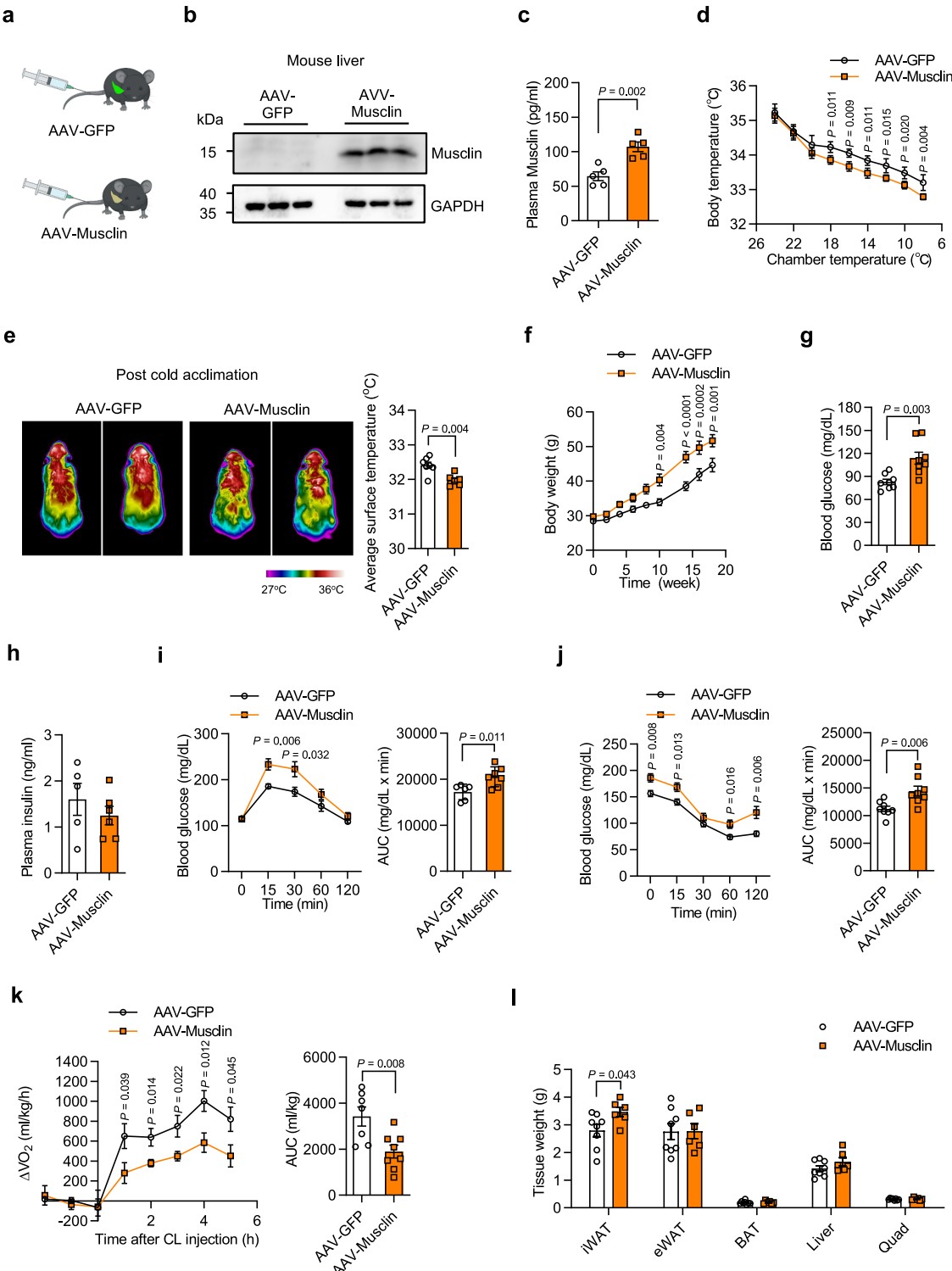

**Fig. 5 | Adeno-associated virus-mediated elevation of Musclin suppresses thermogenesis and systemic energy expenditure. a** Schematic representation of tail vein injection of Adeno-associated virus (AAV) expressing GFP or Musclin in mice. **b** Immunoblots of total liver lysates from AAV-GFP/Musclin transduced male mice (*n* = 3 per group). **c** Plasma Musclin levels in AAV-GFP/Musclin transduced male mice (6 months old). *n* = 5 per group. **d** Core body temperature of chow diet-fed AAV-GFP/Musclin transduced female mice during cold acclimation (5 months old). *n* = 5 per group. **e** Representative infrared images and average body temperature of chow diet-fed AAV-GFP/Musclin transduced male mice following cold acclimation (3.5 months old). AAV-GFP vs AAV-Musclin, *n* = 7 vs 6. **f–l** Three-month-old AAV-GFP/Musclin transduced male mice were fed with HFD at 16 °C. **f** Body weight. *n* = 8 per group. **g** Fasting blood glucose. *n* = 8 per group. **h** Plasma insulin

concentration. AAV-GFP vs AAV-Musclin, *n* = 5 vs 6. **i** Oral glucose tolerance test (left) and AUC (right). AAV-GFP vs AAV-Musclin, *n* = 6 vs 7. **j** Insulin tolerance test (left) and AUC (right). *n* = 8 per group. **k** CL-induced oxygen consumption at RT (24 °C) (left) of indicated male mice fed with HFD for 4 weeks, represented as changes over values measured before CL treatment, and AUC (right). AAV-GFP vs AAV-Musclin, *n* = 7 vs 8. **l** Weight of indicated tissues from male mice. AAV-GFP vs AAV-Musclin, *n* = 8 vs 6. **c–l** Data represent mean ± SEM; *n* shown in this figure represents biologically independent animals. **d**, **f** Two-way ANOVA with Sidak's multiple comparisons. **c**, **e**, **g–l** Two-tailed unpaired Student's *t*-test. All experiments here were repeated independently three times with similar results. Source data are provided as a Source Data file.

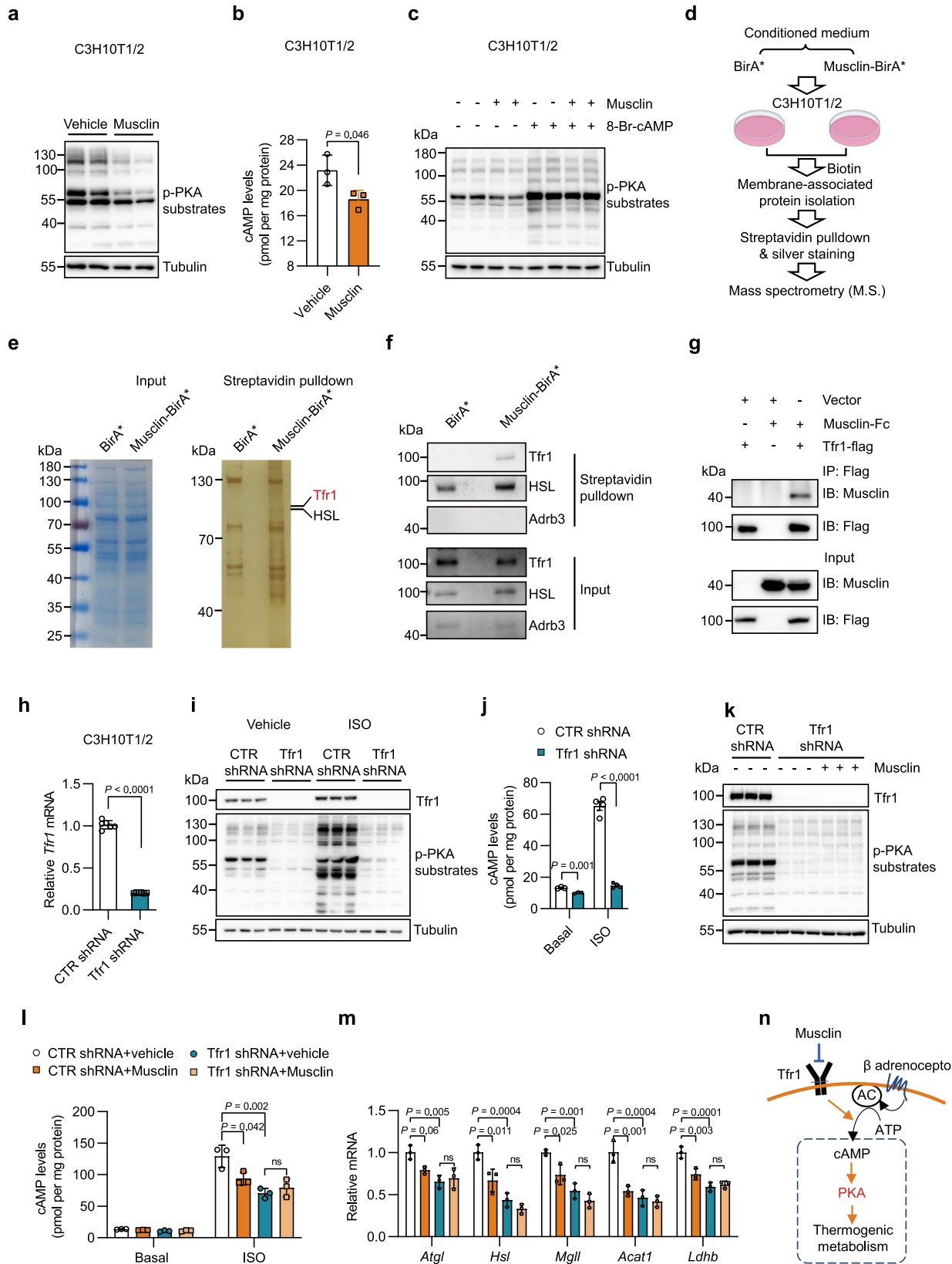

are known to function as receptors[29–32]. Due to the much lower enrichment score of P2x purinoceptor 4 than those of Tfr1 and HSL, our primary focus then turned to the validation of the interaction between Musclin and Tfr1 or HSL. Western blotting analysis demonstrated that Tfr1 was specifically present in the pulldown sample treated with CM from cells expressing Musclin-BirA*, but not in the

sample treated with CM from cells expressing BirA* alone, while HSL was observed in both samples (Fig. 6f). Since BirA* alone in the pCMV-BirA* vector does not contain a signal peptide and may not be able to secrete well into the culture medium. To solve this issue, we also constructed a vector that expressed BirA* with a flag tag (for the western blotting analysis of BirA*) on its C-terminus and the same

**Fig. 6 | Musclin directly binds to Tfr1 and antagonizes cAMP/PKA signalling-dependent thermogenic induction in adipocytes. a** Immunoblots of differentiated C3H10T1/2 cells treated with vehicle or Musclin for 1 h. **b** cAMP levels of differentiated C3H10T1/2 cells treated with vehicle or Musclin-Fc for 4 h. $n = 3$ per group. **c** Immunoblots of differentiated C3H10T1/2 cells pretreated with vehicle or Musclin-Fc, followed by treatment of 8-Br-cAMP or vehicle. **d** Schematic of screening for Musclin-interacted membrane-associated proteins using BioID assay. Musclin-BirA*, Musclin and BirA* fusion protein. **e** Coomassie blue staining of total membrane-associated proteins isolated from differentiated C3H10T1/2 cells (left, shown as input), and silver staining of proteins pulled down from total membrane-associated proteins by streptavidin beads (right, shown as streptavidin pulldown). **f** Immunoblots of input and streptavidin pulldown samples as indicated in (**e**). **g** Physical interaction of Musclin and Tfr1 in transiently transfected HEK293T cells. IP, immunoprecipitation; Musclin-Fc, Fc-tagged Musclin; Tfr1-flag, Flag-tagged Tfr1. **h** qPCR analysis of *Tfr1* gene expression in adipocytes derived from C3H10T1/2 transduced with retrovirus expressing CTR shRNA or *Tfr1* shRNA. $n = 6$ per group.

**i** Immunoblots of whole-cell lysates of adipocytes derived from C3H10T1/2 treated with CTR shRNA or *Tfr1* shRNA followed by treatment with vehicle or ISO (50 nM). ISO, Isoproterenol. **j** cAMP levels in indicated adipocytes following treatment with vehicle or ISO (100 nM). $n = 3$ for basal condition and $n = 4$ for ISO treatment. **k** Immunoblots of whole-cell lysates of adipocytes derived from C3H10T1/2 treated with CTR shRNA or *Tfr1* shRNA followed by treatment with vehicle or Musclin-Fc. **l** The basal and ISO-induced cAMP levels in adipocytes derived from C3H10T1/2 transduced with CTR shRNA or *Tfr1* shRNA following treatment with vehicle or Musclin-Fc. $n = 3$ per group. ns, no significance. **m** qPCR analysis of gene expression in indicated C3H10T1/2-derived adipocytes. $n = 3$ per group. **n** Model depicting the thermogenic inhibition effect of Musclin. **b, h, j** Data represent mean ± SD; $n$ shown here represents biologically independent cell samples. **b, h, j** two-tailed unpaired Student's *t*-test. **l, m** One-way ANOVA with Tukey's multiple comparisons. All experiments here are repeated independently three times with similar results. Source data are provided as a Source Data file.

signal peptide (Sp-BirA*-flag) as Musclin-BirA* on its N-terminus. As shown in Supplementary Fig. 6b (**Left**), Sp-BirA*-flag is secreted and readily detectable in the CM. We then applied the conditioned media containing either Sp-BirA*-flag or Musclin-BirA* to treat the fully differentiated C3H10T1/2-derived adipocytes, followed by streptavidin pulldown of the cell membrane-associated proteins and cytoplasmic proteins. Tfr1 was confirmed to be present in the pulldown sample of membrane-associated protein fraction treated with Musclin-BirA*, but barely detectable in the sample treated with Sp-BirA*-flag (Supplementary Fig. 6b, **middle**). As expected, no signal was detected in the pulldown samples of cytoplasmic protein fractions treated with CM containing either Musclin-BirA* or Sp-BirA*-flag (Supplementary Fig. 6b, **right**). Moreover, the Co-IP assay using transient transfection in HEK293T cells further confirmed the interaction between Musclin and Tfr1, but not HSL (Fig. 6g and Supplementary Fig. 6c). Tfr1 is well-known as an essential receptor in mediating iron uptake through internalizing transferrin-bound iron and has also been recently shown to be important for regulating brown and beige fat thermogenesis[29]. To examine the role of Tfr1 in mediating the effect of Musclin on the adipocyte thermogenic metabolism, we knocked down *Tfr1* in C3H10T1/2-derived beige adipocytes, as confirmed by qPCR and western blotting analyses of Tfr1 expression (Fig. 6h, i). Notably, both basal and isoproterenol (ISO, a nonselective β-adrenoceptor agonist)-induced PKA signaling was dramatically attenuated by Tfr1 inactivation in C3H10T1/2-derived beige adipocytes (Fig. 6i). This was accompanied by remarkable decreases in cellular cAMP levels under both conditions (Fig. 6j). Interestingly, Musclin treatment exhibited no further inhibitory effect on PKA signaling in Tfr1 knockdown beige adipocytes (Fig. 6k), suggesting that Musclin and Tfr1 function in the same pathway. Consistently, ISO-induced elevation of cAMP levels was robustly decreased by either Musclin treatment or Tfr1 knockdown, and no further decrease was observed by Musclin treatment in Tfr1 knockdown beige adipocytes (Fig. 6l). In addition, the thermogenic gene program displayed a similar expression pattern in control and Tfr1 knockdown adipocytes treated with or without Musclin (Fig. 6m). Taken together, these data suggest that Musclin inhibits cAMP/PKA signaling in adipocyte thermogenesis through antagonizing its receptor Tfr1 on the plasma membrane (Fig. 6n).

To further test the physiological function of Tfr1 as the receptor for Musclin in vivo, we first examined the protein levels of Tfr1 in mature adipocytes derived from BAT, iWAT, and eWAT of WT mice following chronic cold exposure. Consistent with the selectivity of Musclin on the thermogenesis of beige adipocytes, we observed that Tfr1 protein levels in mature adipocytes derived from iWAT were significantly higher than those from eWAT and BAT (Supplementary Fig. 6d). We then generated inducible adipose tissue-specific Tfr1 knockout (Tfr1-iAKO) mice by crossing Tfr1^{flox/flox} mice with Adipoq-CreERT2 mice (Supplementary Fig. 6e). The mRNA levels of *Tfr1* in

iWAT and BAT were substantially decreased following tamoxifen (TMX) treatment (Supplementary Fig. 6f), and Tfr1 ablation largely abolished the binding signal of SEAP-Musclin fusion protein to iWAT tissue sections (Supplementary Fig. 6g). Compared to controls, Tfr1-iAKO mice displayed significantly lower core body temperature during cold acclimation (Supplementary Fig. 6h). Moreover, Tfr1 inactivation (Tfr1-iAKO) decreased the core body temperature to a similar extent as AAV-Musclin treatment (Tfr1^{flox/flox} + AAV-Musclin) did, whereas AAV-Musclin treatment in Tfr1-iAKO mice (Tfr1-iAKO +AAV-Musclin) exhibited no further effect on body temperature compared to the effects of their treatment alone (Supplementary Fig. 6h). These in vivo functional studies further confirm that Musclin antagonizes Tfr1-mediated activation of the thermogenic metabolism in subcutaneous beige fat.

## Musclin inactivation promotes thermogenesis and improves systemic energy homeostasis

Given that elevated Musclin level in circulation plays an important role in diet-induced metabolic disorders through inhibiting the thermogenic metabolism in adipose tissues, we considered that Musclin blockade might have therapeutic potential for obesity-associated metabolic diseases. To test this, two independent approaches were employed. Firstly, we generated muscle-specific Musclin knockout (Musclin-MKO) mice by crossing Musclin^{flox/flox} mice with MLC-Cre mice (Supplementary Fig. 7a). Successful knockout of Musclin in skeletal muscle was confirmed by qPCR and western blotting analyses of Musclin mRNA and protein levels, respectively (Supplementary Fig. 7b and Fig. 7a), leading to a significant decrease of Musclin in circulation (Fig. 7b). Running performance and muscle morphology remained indistinguishable between Musclin-MKO and control mice (Supplementary Fig. 7c, d). Interestingly, Musclin-MKO mice displayed significantly higher capacity in maintaining body temperature during cold exposure assays under either chow or HFD feeding conditions (Supplementary Fig. 7e and Fig. 7c, d). Fasting blood glucose levels were also significantly lower in HFD-fed Musclin-MKO mice than those in the control group (Fig. 7e). Further GTT and ITT assays revealed that Musclin-MKO mice exhibited improved glucose tolerance and insulin sensitivity (Fig. 7f, g). While body weight and fat & lean mass were similar between Musclin-MKO and control mice (Supplementary Fig. 7f, g), tissue weight and average adipocyte size of iWAT were significantly lower in Musclin-MKO mice compared to control mice (Fig. 7h, i). We next subjected the control and Musclin-MKO mice to a whole-body metabolic study to examine the effect of Musclin deficiency on systemic energy expenditure. The whole-body metabolic rates were monitored under thermoneutrality at 30°C for 1 week followed by adaptation at 24°C for an additional 4 days. Compared to control mice, Musclin-MKO mice displayed significantly higher $O_2$ consumption rates and energy expenditure in response to adaptation

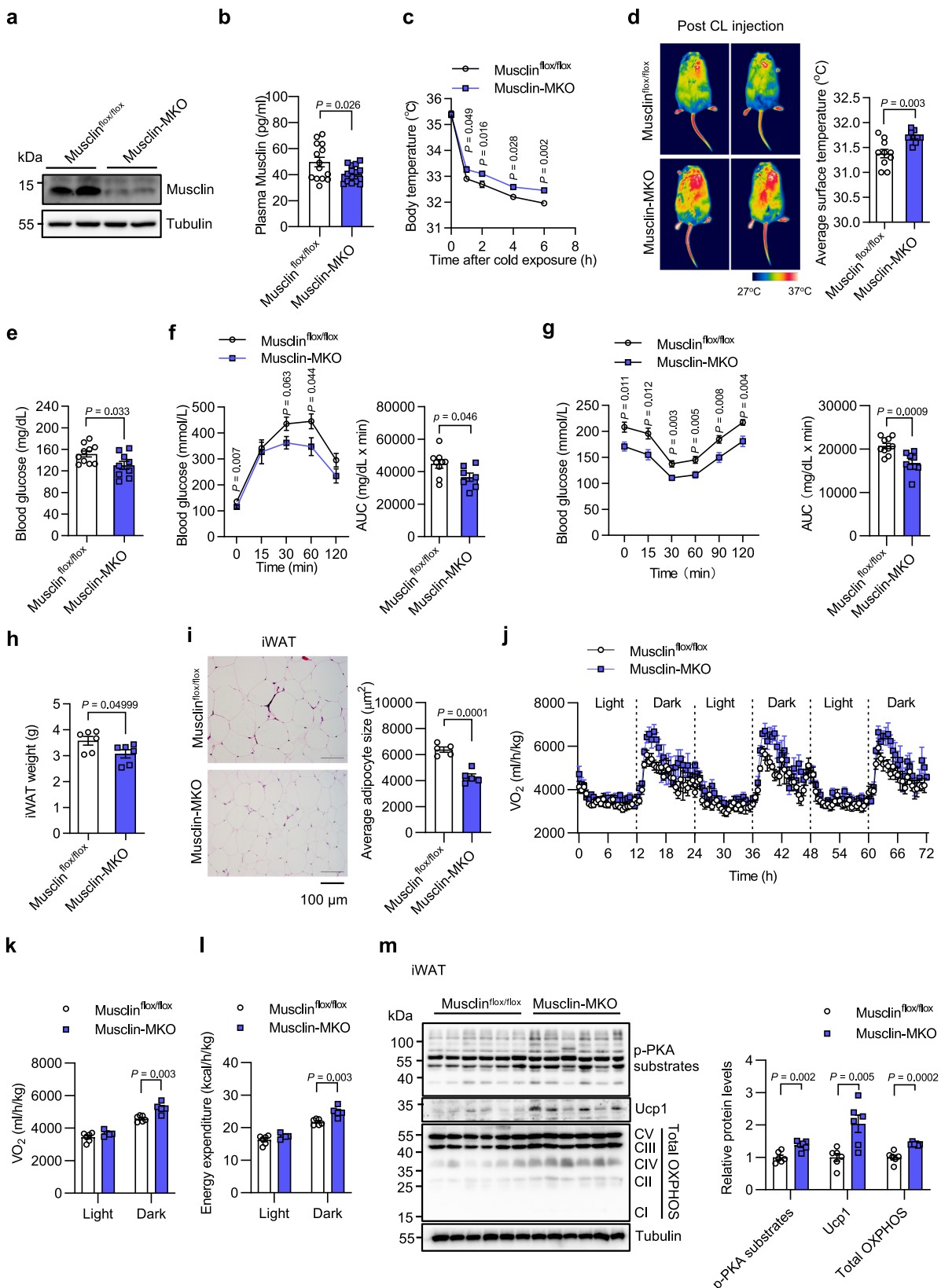

at 24 °C, especially during the dark phase (Fig. 7j–l, Supplementary Fig. 7h), while the food intake and running activity remained similar between the two groups (Supplementary Fig. 7i, j). Moreover, the levels of phosphorylated PKA substrates and thermogenesis-related protein Ucp1, as well as the protein levels of mitochondrial OXPHOS complexes were markedly elevated in iWAT of Musclin-MKO mice

(Fig. 7m). These results demonstrate that the genetic inactivation of Musclin in skeletal muscle enhances beige fat thermogenesis and improves systemic glucose homeostasis.

To further explore the therapeutic potential of Musclin blockage for metabolic diseases, we next attempted to inactivate Musclin in the circulation using a rabbit anti-Musclin neutralizing antibody

**Fig. 7 | Muscle-specific Musclin knockout promotes thermogenesis and energy expenditure.** Male Musclin[flox/flox] and Musclin-MKO were used in studies shown in this figure. **a** Immunoblots of total protein lysates from muscle. **b** Plasma Musclin levels. Musclin[flox/flox] vs Musclin-MKO, $n = 14$ vs 15. **c-m**, Mice fed with HFD began at 3 months old and continued for 3–5 months. **c** Body temperature of HFD-fed mice during acute cold exposure. Musclin[flox/flox] vs Musclin-MKO, $n = 8$ vs 7. **d** Representative infrared images (left) and the average body surface temperature (right) of mice following 6.5 h of CL316,243 injection. Musclin[flox/flox] vs Musclin-MKO, $n = 10$ vs 9. **e** Blood glucose following 6 h of fasting at 16 °C. Musclin[flox/flox] vs Musclin-MKO, $n = 10$ vs 9. **f** Glucose tolerance test (left) and AUC (right). Musclin[flox/flox] vs Musclin-MKO, $n = 9$ vs 7. **g** Insulin tolerance test (left) and AUC (right). Musclin[flox/flox] vs Musclin-MKO, $n = 10$ vs 9. **h** iWAT weight. $n = 6$ per group. **i** Representative H&E staining images of iWAT (left) and adipocyte size analysis (right). $n = 5$

per group. **j** Oxygen consumption rates of mice were monitored at RT (24 °C) for 3 consecutive days following thermoneutrality at 30 °C (1 week) and an additional 4 days of adaptation at RT. Lean mass was applied for normalization. **k** Average oxygen consumption rate of mice at day time or night time during the 3 days of monitoring as in (j). **l** Averaged energy expenditure tested at day time and night time. **j–l** Musclin[flox/flox] vs Musclin-MKO, $n = 6$ vs 5. **m** Immunoblots of total iWAT protein lysates (left) and quantification of protein levels (right). $n = 6$ per group. **b–m** Data represent mean ± SEM ($n$ represents biologically independent animals). **c**, two-way ANOVA with Sidak's multiple comparisons. **b, d–i, k–m** Two-tailed unpaired Student's $t$-test. Experiments in **b–e, m** were repeated independently three times, and results in **f–l** were repeated independently twice with similar results. Source data are provided as a Source Data file.

(Musclin Ab) and examined the subsequent effects of this Musclin inactivation on the thermogenic metabolism in mice (Fig. 8a). As shown in Fig. 8b, Musclin Ab successfully abolished the downregulation of thermogenic gene program by Musclin in differentiated C3H10T1/2 adipocytes. We then utilized this neutralizing antibody to treat both chow and HFD-fed mice through intraperitoneal injection (using saline as a control). Interestingly, a relatively low dose of Musclin Ab (0.12 mg Kg⁻¹ BW) was sufficient to decrease body weight gain and WAT (both iWAT and eWAT) weight and to reduce lipid droplet size in iWAT of chow diet-fed mice (Fig. 8c–e), while no obvious side effects were observed, including upon food intake or wheel-running activity, during the entire long-term treatment period (Supplementary Fig. 7k, l). Accordingly, the relative levels of p-PKA substrates and p-HSL were elevated in iWAT by Musclin Ab treatment (Fig. 8f). Furthermore, Musclin Ab treatment effectively promoted oxygen consumption (Fig. 8g, h), energy expenditure (Fig. 8i) and thermogenesis (Fig. 8j) of HFD-fed WT mice, leading to a significant decrease in blood glucose levels and improved glucose tolerance (Fig. 8k, l). Together, these findings suggest that the inactivation of Musclin is a practical approach to activating beige fat thermogenesis and improving systemic energy homeostasis in mice, and thereby holds therapeutic potential for obesity and other associated metabolic diseases in humans.

## Discussion

Skeletal muscle, the central organ for postprandial glucose disposal, is critical for controlling the glucose metabolism and energy homeostasis of the whole-body[1–4]. Beyond nutrient and fuel metabolism, skeletal muscle has also emerged as an endocrine organ to maintain systemic energy balance and disease progression through myokine-mediated muscle-tissue crosstalk[5,6]. Brown and beige fat thermogenesis and function have attracted considerable attention over the past decade due to their energy dissipating features and therapeutic potential for obesity-associated metabolic disorders. Study of their related regulatory mechanisms have led to the identification of several local and systemic secreted factors that activate brown and beige fat energy expenditure and improve systemic energy homeostasis[18]. These include several myokines such as Irisin[33], Meteorin-like (Metrnl)[34], and β-aminoisobutyric acid (BAIBA)[35]. However, whether skeletal muscle and adipose tissue are horizontally interconnected in response to environmental temperature challenges to precisely control systemic thermogenesis and metabolic homeostasis remains unclear. In the present work, we identified Musclin as a temperature sensitive myokine that can directly bind to beige adipocyte and inhibit its thermogenesis through antagonizing Tfr1-PKA mediated thermogenic induction. Muscle expression of Musclin is notably increased in obese and diabetic mice and humans, leading to a corresponding elevation of Musclin levels in the circulation. Under chow diet-fed conditions, Musclin is critical for the regulation of murine body temperature and glucose homeostasis in response to cold exposure.

Moreover, muscle-specific transgenic or AAV-mediated elevation of circulating Musclin levels inhibits beige fat metabolism and exacerbates HFD-induced obesity, insulin resistance and glucose intolerance. More importantly, inhibition of Musclin action by genetic ablation or neutralizing antibody treatment promotes beige fat thermogenesis, energy expenditure, and ameliorates HFD-induced metabolic defects. In this way, we have revealed temperature-regulated myokine Musclin as a potential therapeutic target for treating obesity and associated metabolic diseases.

Musclin has also been termed Osteocrin since it was initially found to be expressed in osteoblasts and young osteocytes that modulates osteoblast function, developmental bone growth[23,36], and physiological load-induced bone formation[37]. Musclin was subsequently also identified as a novel secretory factor preferentially expressed in skeletal muscle in adult rodents[24]. However, as a muscle-enriched secretory factor, the role and underlying mechanisms of Musclin in regulating systemic thermogenesis and metabolic health remain largely unknown. Here we uncovered Musclin as a critical negative regulator of beige fat thermogenesis and energy expenditure, linking muscle bioenergetic and nutrient-sensing functions to the control of systemic metabolic homeostasis under both physiological (cold exposure) and pathophysiological (diet-induced obesity) conditions through muscle-beige fat crosstalk. There is mounting evidence suggesting the influences exerted by different bone-derived molecules in systemic metabolism[38]. It is possible that Musclin from bone may also play a role in the regulation of systemic energy metabolism in adult mice. To this end, we have also examined Musclin expression levels in skeletal muscle and bone from WT C57BL/6J adult mice. Consistent with the previous study[24], the tibia, a long bone in the hindlimb, displayed a markedly lower expression of Musclin gene than that in skeletal muscle in adult mice. In addition, we also examined the Musclin mRNA levels in both skeletal muscle and long bone from Musclin-MKO and control mice, and showed that muscle-specific knockout of Musclin does not influence its expression in bone. These data indicate that skeletal muscle-derived Musclin plays a pivotal role in the regulation of adipose tissue thermogenesis and systemic energy homeostasis. Nonetheless, we also noticed that the plasma concentration of Musclin was partially decreased in Musclin-MKO mice, suggesting that other tissues may also contribute to the circulating levels of Musclin. The roles of Musclin secreted from these tissues in the regulation of tissue homeostasis and energy metabolism remain to be explored in future studies. Interestingly, Musclin can be repurposed in primates through the evolutionary acquisition of DNA regulatory elements to regulate neuronal structure and function features that are unique to primates[39], suggesting a species-specific function of Musclin. More recently, Musclin was shown to function as an exercise-responsive myokine[40], to attenuate the pathogenesis and progression of several cardiovascular diseases including heart failure during pathological overload[41], hypertension[42], cardiac remodeling and congestive heart failure after myocardial infarction[43], and chronic doxorubicin-induced

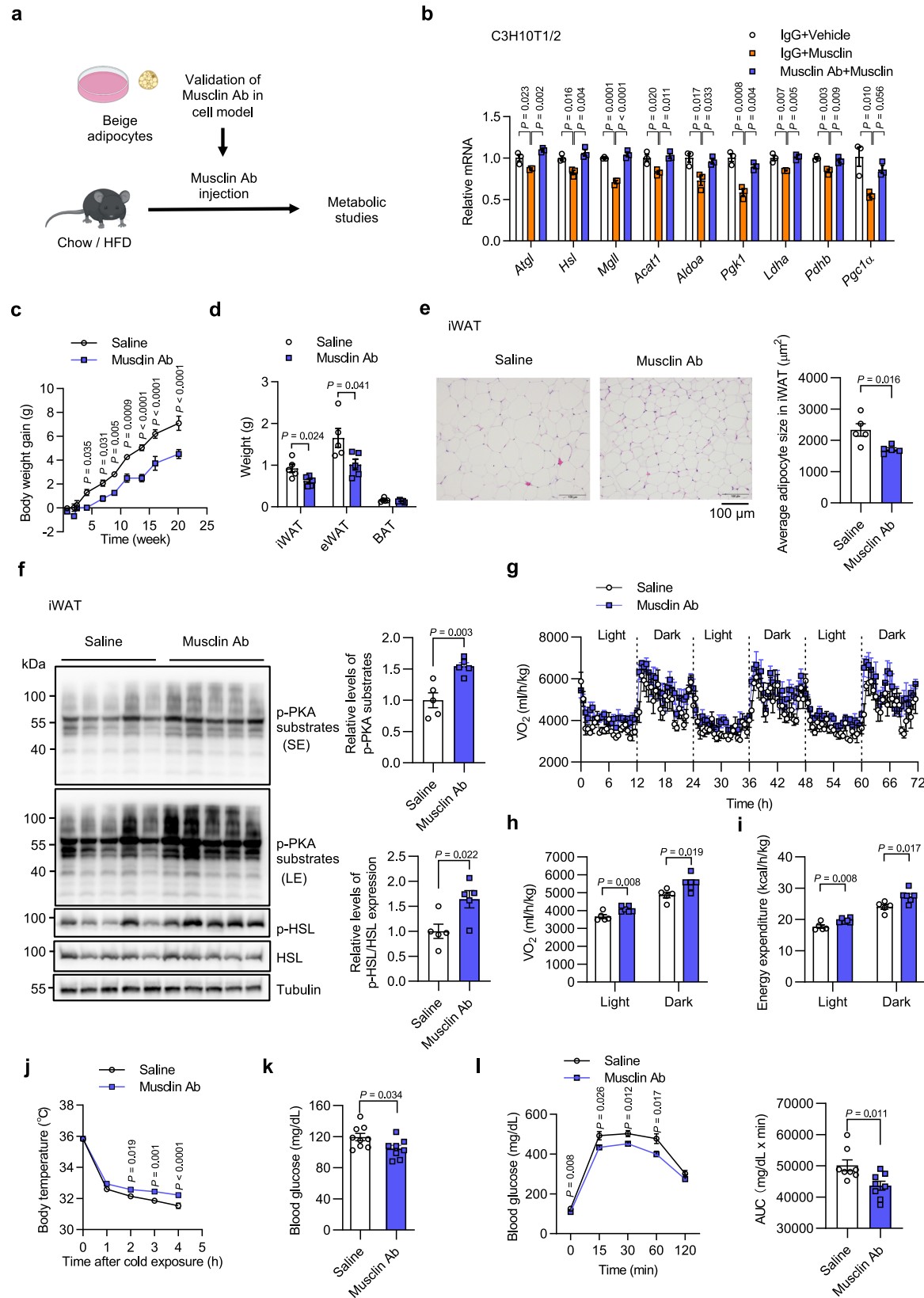

cardiotoxicity[44] in animal models. In addition, low circulating Musclin levels are associated with adverse prognosis of patients undergoing transcatheter aortic valve implantation (TAVI)[45], and potential atrial fibrillation in non-diabetic patients[46]. Taken together, these findings demonstrate the tissue-specific and context-dependent roles of Musclin under physiological and pathological conditions.

Musclin was previously reported to impair glucose uptake in cultured myotubes via its repression of insulin-stimulated activation of the Akt/protein kinase B pathway[24]. Using SEAP binding assay, we demonstrated that Musclin mainly binds to adipose tissues, especially subcutaneous iWAT, rather than other metabolic tissues, such as skeletal muscle and liver. Further studies revealed that elevated

**Fig. 8 | Neutralizing antibody-mediated Musclin blockade improves energy homeostasis.** WT male mice were treated with saline or Musclin Antibody (Musclin Ab). **a** Schematic of in vitro validation of Musclin antibody and in vivo Musclin neutralization assay. **b** qPCR analysis of C3H10T1/2-derived adipocytes pretreated with control IgG or Musclin Ab, followed by treatment with forskolin. $n = 3$ per group. **c**–**f** Chow diet-fed mice were treated with saline or Musclin Ab beginning at 3 months old. $n = 5$ per group. **c** Body weight gain. **d** Weight of indicated tissues. **e** Representative H&E staining images of iWAT (left) and average adipocyte size analysis (right). **f** Immunoblots of total lysates from iWAT (left), and quantification of protein levels (right). **g**–**i** Chow diet-fed mice were treated with saline or Musclin Ab beginning at 2.5 months old and continued for 3 months. Saline vs Musclin Ab, $n = 5$ vs 6. **g** Oxygen consumption rate of 3 consecutive days monitored at RT (24°C) following thermoneutrality at 30°C (1 week) and an additional 2 days of adaptation at RT. Lean mass was applied for normalization. **h** Average oxygen consumption rate at day time or night time during the 3 days of monitoring as in (**g**). **i** Average energy expenditure at day time or night time. **j** Core body temperature of HFD-fed mice during acute cold exposure at 4°C. $n = 8$ per group. HFD feeding began at 2.5 months old with antibody treatment (for 1 month) after 5.5 months old. **k** Blood glucose levels. Saline vs Musclin Ab, $n = 9$ vs 8. HFD feeding began in mice at 3.5 months old with antibody treatment (for 2 weeks) at 4.5 months old. **l** Glucose tolerance test (left) and the AUC (right). $n = 8$ per group. HFD feeding began in mice at 3.5 months old with antibody treatment (for 1 month) at 4.5 months old. **b** Data represent mean ± SD ($n$ represents biologically independent cell samples); **c**–**l** data represent mean ± SEM ($n$ represents biologically independent animals). **b**, one-way ANOVA with Tukey's multiple comparisons. **c**, **j** two-way ANOVA with Sidak's multiple comparisons. **d**-**f**, **h**, **i**, **k**, **l** Two-tailed unpaired Student's $t$-test. Experiments in **b**, **f**, **j**, **k** were repeated independently three times, and results in **c**–**e**, **g**–**i**, **l** were repeated independently twice with similar results. Source data are provided as a Source Data file.

circulating Musclin selectively acts on iWAT to suppress thermogenic gene expression programs, resulting in impairment of beige fat thermogenesis, lower energy expenditure, and augmentation of HFD-induced obesity and metabolic dysfunction. Consistently, both MCK-Musclin and Musclin-MKO mice used in this study exhibited no detectable differences in muscle morphology, exercise performance, or energy metabolism-related gene expression profiles compared to their respective controls. However, a previous study demonstrated that Musclin whole-body knockout mice displayed impaired oxidative metabolism in skeletal muscle and reduced exercise tolerance compared to controls following five consecutive days of exercise training[40]. Three reasons might contribute to this discrepancy. Firstly, to exclude the potential confounding effects of Musclin on muscle development and bone growth[23,36], we generated mice carrying muscle-specific transgenic expression and knockout of Musclin using MCK promoter[47] and MLC-Cre[48], respectively. This lies in contrast to the Musclin whole-body knockout mice utilized in their study. Secondly, different running protocols and testing conditions were used in the two studies. Whilst our evaluation incorporated all the muscle-related parameters in sedentary mice, in their research mice were subjected to 5 consecutive days of exercise training prior to assessment[40]. They actually found that differences in endurance running capacity and markers of mitochondrial biogenesis were much less evident in sedentary WT and Musclin-KO mice than those in exercise-trained WT and Musclin-KO mice. This is consistent with their hypothesis that running endurance affected by Musclin is mediated by its effect on ANP/Pgc1α pathway-dependent regulation of exercise training-related mitochondrial biogenesis. Thirdly, different receptors and associated downstream signal pathways might be another reason for the tissue-specific effects of Musclin on mitochondrial biogenesis in skeletal muscle and adipose tissue. In the previous study, Musclin was shown to bind to Npr3, a natriuretic peptide (NP) clearance receptor, leading to the enhancement of activity-related ANP/cGMP/Pgc1α-dependent skeletal muscle mitochondrial biogenesis and endurance running activity. However, in this study, we identified Tfr1 as a receptor for Musclin in adipocytes, and revealed that Tfr1-mediated cAMP signaling plays a pivotal role in the regulation of thermogenic metabolism in subcutaneous white adipose tissue.

Brown and beige/brite adipocytes share remarkable morphological and transcriptional similarities. However, recent studies have demonstrated distinguishing features and mechanisms underlying the developmental origins and functional regulation of these two cell types[14,15]. This study demonstrated that Musclin selectively repressed gene expression networks related to lipid and glucose metabolism in the subcutaneous beige fat depot, resulting in the impairment of thermogenesis and augmentation of HFD-induced obesity and metabolic dysfunction. However, tissue morphology and thermogenic metabolism of interscapular "classical" BAT remained largely unaltered by Musclin gain-of-function. This highlighted a distinct

regulatory mechanism related to beige fat thermogenic metabolism (beiging) with a unique role in the pathogenesis of obesity and its associated metabolic dysfunction.

Receptors are generally indispensable for mediating the vital role of secreted proteins in the crosstalk among tissues. Musclin has been shown to function as a ligand to Npr3, competing with natriuretic peptides (NPs), to amplify the NP signaling[36,49]. As such, it is possible that Musclin may elicit its effects on adipocyte thermogenesis and metabolism through regulating the circulation levels of NPs and the downstream cGMP levels, a key player in mediating NP-associated responses[50]. It has been reported that Musclin transgenic mice exhibited a significant increase in plasma CNP levels at 28 days after myocardial infarction[43]. Consistently, a recent study by Szaroszyk M et al. also demonstrated that plasma CNP levels were significantly elevated in mice with AAV6-mediated overexpression of Musclin in skeletal muscle at 3 weeks after transverse aortic constriction (TAC) surgery, as compared to control mice[41]. Conversely, plasma CNP levels were lower in muscle-specific Musclin deficient mice versus control mice after TAC surgery[41]. However, plasma ANP levels were either trending lower or higher in Musclin overexpressing mice after myocardial infarction[43] or TAC surgery[41], respectively. Moreover, it has been reported that plasma ANP levels were trending lower, but did not reach statistical significance, in Musclin whole-body KO mice compared to WT controls following exercise training[40]. These results revealed an overall modest effect of Musclin overexpression or deficiency on plasma ANP levels in mice. Consistent with these findings, we observed that plasma ANP levels remained largely unchanged in MCK-Musclin mice compared to control mice in response to cold exposure or HFD feeding (Supplementary Fig. 8a, b). Similarly, plasma levels of ANP were similar between Musclin^flox/flox and Musclin-MKO mice under either cold exposure or HFD feeding conditions (Supplementary Fig. 8a, b). Notably, plasma CNP levels were significantly increased in MCK-Musclin mice compared to controls under either cold exposure or HFD feeding conditions (Supplementary Fig. 8c, d). In contrast, plasma concentrations of CNP were lower in Musclin-MKO mice than those in Musclin^flox/flox mice in response to either cold exposure or HFD feeding conditions (Supplementary Fig. 8c, d).

The cGMP, generated by the guanylyl cyclase domain of NPRA (receptor for ANP and BNP) and NPRB (receptor for CNP), is a key mediator of the NP-associated biological responses[50]. It has been reported that Musclin could enhance cardiomyocyte cAMP generation through cGMP-mediated inhibition of cAMP-degrading phosphodiesterase 3 (PDE3)[41]. Notably, it has been reported that Musclin ($10^{-5}$ M) treatment could significantly increase the intracellular cGMP levels in the presence of $10^{-8}$ M ANP in human umbilical vein endothelial cells (HUVECs)[43]. We next went on to test the involvement of the cGMP/PDE3 signaling in the regulation of thermogenic metabolism in adipocytes by Musclin. As a positive control in our experimental system, we first treated the HUVECs with increasing concentrations of Musclin

plus $10^{-8}$ M ANP. Similar to the results shown in the previous study[43], we also showed that Musclin ($10^{-5}$ M) significantly elevated the intracellular cGMP levels in the presence of $10^{-8}$ M ANP in HUVECs (Supplementary Fig. 8e). Previous studies showed that both ANP and CNP could increase the intracellular cGMP levels in adipocytes[51,52]. Consistently, we also demonstrated that both ANP (Supplementary Fig. 8f) and CNP (Supplementary Fig. 8g) elevated intracellular cGMP levels in differentiated C3H10T1/2-derived adipocytes in a dose-dependent manner. However, co-treatment with Musclin exhibited no obvious effect on intracellular cGMP levels in response to increasing amounts of ANP or CNP (Supplementary Fig. 8f, g). To further examine the involvement of cGMP signaling in the regulation of adipocyte thermogenic metabolism in vivo, we also measured the cGMP levels in iWAT and BAT tissues from control and MCK-Musclin mice in response to either cold exposure or HFD feeding, as well as in iWAT from Musclin$^{flox/flox}$ and Musclin-MKO mice upon HFD feeding. As shown in Supplementary Fig. 8h–j, cGMP levels in iWAT or BAT remained largely unchanged after the manipulation of Musclin in mice upon HFD feeding or cooling. Taken together, these data indicate that the cGMP/PDE3 pathway downstream of ANP or CNP signaling might not be involved in the regulation of adipocyte thermogenic metabolism and systemic energy homeostasis by Musclin.

Interestingly, using the proximity-dependent biotin identification (BioID) assay and mass spectrometry analysis, we identified Tfr1 as the membrane receptor for Musclin in mediating its inhibitory effect on the thermogenic metabolism in beige adipocytes. Tfr1, as a member of the transferrin receptor family, is known to play an important role in cellular iron homeostasis through the endocytosis of transferrin-bound iron[53,54]. One recent study has demonstrated that Tfr1-mediated control of cellular iron levels is important for regulating white adipose tissue homeostasis[55]. Intriguingly, accumulating evidence indicates that other iron-independent mechanisms might also be involved in the regulation of biological function by Tfr1[56]. Accordingly, Tfr1 was uncovered to play an essential role in regulating brown and beige fat development and thermogenesis via both iron-dependent and -independent mechanisms[29,57]. Consistent with the selectivity of Musclin on the thermogenesis of beige adipocytes, we observed that the protein levels of Tfr1 in adipocytes from iWAT were significantly higher than those from eWAT and BAT in WT mice following chronic cold exposure (Supplementary Fig. 6d). However, whether iron-dependent and -independent mechanisms are involved in this process warrants further investigation.

In summary, this work identifies the myokine Musclin as a critical negative regulator of beige fat thermogenesis that acts in concert with other thermogenesis activators to fine-tune systemic energy balance under both physiological and pathophysiological conditions. Muscle expression of Musclin is regulated by ambient temperature and is essential for maintaining body temperature upon acute and chronic cold stresses, thus indicating its physiological role in gauging systemic energy expenditure. Musclin gain-of-function inhibits beige fat thermogenesis and augments HFD-induced obesity and metabolic disorders. More importantly, blocking Musclin actions, either by genetic ablation or neutralizing antibody treatment, promotes energy expenditure and alleviates HFD induced-obesity and metabolic dysfunction. These findings highlight the therapeutic potential of Musclin inactivation for treating obesity and its associated metabolic disorders.

## Methods

### Ethics statement
All animal studies were performed in compliance with the Guide for the Use and Care of Laboratory Animals by the Medical Experimental Animal Care Committee of Zhejiang University. All animal studies were performed following the protocols approved by the Animal Ethics Committee of the Second Affiliated Hospital, School of Medicine, Zhejiang University (Approval number: 2019-90). The studies on

human skeletal muscle and scWAT were approved by the First People's Hospital of Lianyungang City (Approval number: LW-20180321001), and the Second Affiliated Hospital of Zhejiang University School of Medicine (Approval number: 2020-528), respectively. The study on human plasma was approved by the Second Affiliated Hospital of Soochow University (Approval number: JD-LK-2020-038-01). This study is compliant with the "Guidance of the Ministry of Science and Technology (MOST) for the Review and Approval of Human Genetic Resources" (Approval number: 2023BAT0826).

### Human study
Human biological samples including skeletal muscle (from a total of 54 donors; 41 male and 13 female; data shown in Fig. 1b, e, f, g, and Supplementary Fig. 1b, c), scWAT (from a total of 34 donors; 26 male and 8 female; data shown in Supplementary Fig. 5), and plasma samples (from a total of 144 donors; 89 male and 55 female; data shown in Fig. 1h) were analyzed in this study. According to the BMI criteria by the Working Group on Obesity in China (WGOC), human samples were divided into control (BMI < 24), overweight (24 ≤ BMI < 28), and obesity (BMI ≥ 28) groups. Sex was comparable between groups, as shown in Supplementary Fig. 1b, c and Supplementary Fig. 5a. Sex information for human sample-related results shown in Fig. 1b, e–h are provided in 'Source Data' file. Only the participants who had provided written informed consent were included in the study. All participants representing donor tissue samples were subjected to medical history inquiries before hospitalization. Blood biochemical tests were taken after overnight starvation.

The human muscle samples analyzed were gracilis and semitendinosus muscles collected between 2018–2021 from patients undergoing ligament repair or reconstruction treatment due to ligamentous injury of the knee joint or patella injury at the First People's Hospital of Lianyungang City. Donors with BMI <18, deficiency in muscle development, skeletal muscle-related diseases or displaying disability during regular physical activity, or those with abnormal body height were all excluded from this study. Subjects with other severe diseases, including malignant tumors and severe obesity-independent cardiovascular disorders, were also excluded. Moreover, to minimize the confounding effect of previous injury in the current findings, the human muscle samples were only collected from patients who have fully recovered from acute injury with confirmation of the absence of acute inflammation prior to ligament repair or reconstruction treatments. BMI is the only criterion to divide the human muscle samples into different groups. As such, we assumed that the physical activity should be similar between groups. The scWAT samples were collected between 2019-2020 from the fat layer that is located underneath the Scarpa's Fascia and outside of the abdominal cavity of patients who underwent surgical repair because of oblique inguinal hernia, incisional hernia, or appendicitis at the Second Affiliated Hospital of Zhejiang University School of Medicine. All the patients for scWAT collection were checked for full recovery from acute injury and confirmation of the absence of inflammation before surgery. Fat samples from subjects with malignant tumors were excluded. All the tissue samples were freshly collected and immediately frozen in liquid nitrogen. Human blood samples were collected from subjects with differing BMIs, while subjects with malignant tumors, organic lesions, or having previously undergone surgery within several months to half a year were excluded. Plasma samples were harvested after centrifugation and then stored at −80 °C before use.

### Animal study
Mice were housed in 12/12 h light/dark cycles with a humidity of 50-60%, and fed with either a regular chow diet (10 kcal% fat, 70 kcal% carbohydrate, and 20 kcal% protein; 1010088; Jiangsu Xietong Pharmaceutical Bio-engineering Co., Ltd.) or an HFD (60 kcal% fat, 20 kcal% carbohydrate, and 20 kcal% protein; D12492; Research Diets). Mice

were bred and housed at 23-24 °C except for cold exposure assay. Male mice were used for all the experiments unless otherwise indicated. On rare occasions, mice in bad health conditions, such as severe fighting wounds or exhibiting sickness, according to the predefined criteria listed in the Institutional Animal Care and Use Committee (IACUC) protocol were excluded. The sample size for mouse experiments was >4 per group. All in vivo experiments were independently repeated at least twice.

To generate skeletal muscle-specific Musclin transgenic (MCK-Musclin) mice by pronuclear microinjection, a full-length Musclin coding sequence was placed downstream of the 4.8-Kb murine MCK promoter[47]. Five independent founder lines were established and backcrossed to C57BL/6J for at least eight generations to generate stable transgenic lines. Skeletal muscle-specific Musclin knockout (Musclin-MKO) mice were generated by breeding Musclin[flox/flox] mice with MLC-Cre mice[48]. Tfr1[flox/flox] mice (JAX stock #028363) and Adipoq-CreERT2 mice (JAX stock #025124), were shared by Dr. Liwei Xie (Guangdong Institute of Microbiology). Adipoq-Cre (JAX stock #028020) mice were generously provided by Dr. Fudi Wang (Zhejiang University), and have been previously described[29,58]. To directly inactivate Tfr1 in adipocytes, Tfr1[flox/flox] mice were crossed with Adipoq-Cre mice. Inducible adipose tissue-specific Tfr1 knockout mice were generated by crossing Tfr1[flox/flox] mice with Adipoq-CreERT2 mice. Cre expression in Adipoq-CreERT2 mice was activated through intraperitoneal administration of tamoxifen at the dose of 0.1 g Kg[-1] body weight for seven consecutive days. The *ob/ob* (leptin deficient) and *db/db* (leptin receptor-deficient) mice were purchased from the Jackson Laboratory. All mouse strains used in this study were born at the expected Mendelian ratios with normal fertility.

All the C57BL/6J WT mice were obtained from GemPharmatech (Nanjing, China). All chow diet-fed mice used in this study were 2–8 months old. For diet-induced obesity in mice, HFD feeding was maintained for 2–6 months, during which the experiments were performed. For Musclin neutralizing Ab treatment studies, chow diet-fed mice or 3-month-old C57BL/6J mice fed with HFD were used. Musclin neutralizing Ab preparation was performed by ABclonal Technology Co., Ltd. (Wuhan). Through epitope prediction, the peptide sequence C-HSKKRFGIP-Nle-DRIGRNR (corresponding to 108-124aa of Musclin) was synthesized and used for immunizing rabbits to generate antibodies. Polyclonal antibodies against Musclin were obtained from inoculated rabbits. Antibodies were purified using affinity chromatography on columns containing the corresponding peptides. In the metabolic studies involving HFD-induced obese mice, male mice were used since female mice don't gain as much body weight as male mice do according to our previous experience[1,2,59,60], and studies from other laboratories in the metabolism field[61]. Female mice were only used for the cold exposure study shown in Fig. 5d and metabolic gene expression study in liver following AAV tail vein injection shown in Supplementary Fig. 4d. The mouse age and sex information for each experiment has also been included in the corresponding figure legend.

## Cell culture models

HEK293T, C3H10T1/2, C2C12, and HUVEC cell lines were obtained from ATCC. Expi293F cell line was obtained from Thermo Fisher. AAV293 cell line was purchased from Agilent. The authentication of C3H10T1/2 and C2C12 cells was confirmed through cell morphology and global mRNA and protein expression analyses, such as RNA-Sequencing, qPCR, and/or western blotting, after full differentiation. HEK293T, Expi293F, and AAV293 cell lines were authenticated and routinely used in our previous studies[2,59,60,62]. HUVEC cell line was kindly provided by Dr. Nan Xu (Henan University), and has been authenticated and successfully used in their previous study[63]. HEK293T, C3H10T1/2, C2C12, AAV293, and HUVEC cells were cultured at 37 °C with 5% $CO_2$, and Expi293F cells were cultured at 37 °C with 8% $CO_2$.

## Adeno-associated virus production and transduction

AAV production and purification were performed by ChuangRui Bio (Lianyungang, China). In brief, AAV293 cells (Agilent, 240073) were cultured in Dulbecco's Modified Eagle Medium (DMEM; Gibco, 11995065) with 10% (vol./vol.) fetal bovine serum (FBS; VISTECH, SE100-011) in a humidified incubator containing 5% $CO_2$ at 37 °C. One day before transfection, AAV293 cells were seeded on 20 × 15 cm culture dishes and grown to appropriate 90% confluency before transfection. For each 15 cm dish, 7 μg of AAV shuttle vector carrying the target gene, 20 μg of Delta F6 helper vector, 7 μg of RC2/9 vector, and 250 μL of Polyethylenimine (PEI; Polysciences, Cat# 23966-2) was added to 1 ml DMEM, and then incubated at room temperature for 15 min before being adding to the cells. Culture media were replaced with DMEM plus 0.5% FBS at 24 h post-transfection. Cells were harvested at 72 h post-transfection and centrifuged at 200 g for 10 min. Cell pellets were washed once with 20 ml PBS, resuspended in 5 ml cell lysis buffer (150 mM NaCl, 20 mM Tris pH 8.0), and frozen and thawed three times. Stock solutions containing 1 M $MgCl_2$ and 25 KU ml[-1] Benzonase (Sigma, E8263-25k) were then added to the above cell lysate to achieve the final concentrations of 1 mM and 250 U ml[-1], respectively. Cell lysates were incubated in a 37 °C water bath for 15 min, and centrifuged at 1,500 g at 4 °C for 30 min. Iodixanol solutions (Sigma, D1556-250ML) were prepared at four different concentrations, and then added into the ultracentrifuge tube (Beckman, 361625) in the order of 60%, 40%, 25%, and 17%. Cell lysates were gently transferred onto the top layer, and the remaining volume of the ultracentrifuge tube was filled with cell lysis buffer. The virus was isolated from the lysates by centrifugation at 205,000 g using Beckman Optima XPN-100 Ultracentrifuge with Type 70Ti rotor at 14 °C for 2 h and 40 min. The viral titer was determined by qPCR assay with a standard curve generated by serial dilutions of the AAV shuttle vector. C57BL/6J WT, Tfr1-iAKO and their respective littermate control mice were transduced with AAV-GFP or AAV-Musclin (-5×10[10] viral particles (vp) ml[-1] in 100 μL per mouse) through tail vein injection.

## Cold exposure assay

For acute cold exposure assays, mice were singly housed in pre-chilled cages without bedding in a cold chamber set at 4 °C. Mice were granted free access to pre-chilled food and water during the whole assay. Core body temperature was monitored at indicated time points using a portable intelligent digital thermometer (TH212). For cold acclimation assays, mice were subjected to a temperature-controlled chamber in which the temperature was decreased two degrees every two days until reaching 8 °C and then maintained at 8 °C for five more days. During this period, metabolic parameters including mouse body temperature, body weight, blood glucose, as well as plasma TG and NEFA levels were monitored. In this study, mice subjected to cold exposure at 8 °C for 1 week, chronic cold at 16 °C under HFD feeding condition, and thermoneutrality at 30 °C for 1 week were also used.

## Infrared imaging

Body surface temperature was measured using a thermal imaging camera (FLIR Systems, T430sc InfraRed Camera). Mice were anesthetized with isoflurane and quietly laid on a whiteboard with their back up, followed by image capturing with the camera anchored at the same height for all mice in the same batch. FLIR Tools 5.7 was used for the quantification of average body surface temperature.

## Whole body metabolic study

For the measure of the metabolic rates in AAV-GFP/Musclin mice, the comprehensive Lab Animal Monitoring System (CLAMS) from Columbus Instruments was used at GemPharmatech (Nanjing, China). Briefly, mice were singly housed in metabolic chambers at 24 °C under a 12/12 h light/dark cycle. Food and water were freely accessible to mice. $O_2$ consumption, $CO_2$ production, energy expenditure, total

locomotor activity, and food intake were monitored under both normal conditions and under adrenergic stimulation. For measuring metabolic rates upon adrenergic stimulation, 1 mg Kg$^{-1}$ of CL dissolved in saline was intraperitoneally injected into mice following their adaptation to the metabolic chambers. The metabolic parameters for each mouse were measured two days before CL injection and were continued to be monitored for one more day after CL injection. For the measurement of metabolic rates in HFD-fed Musclin-MKO & Musclin$^{flox/flox}$ and Musclin Ab/saline-treated mice, the PhenoMaster (TSE systems) was used. Mice were also singly housed under a 12/12 h light/dark cycle with food and water freely accessible. Metabolic parameters were monitored after the mice were housed in thermoneutrality at 30 °C for 1 week followed by adaptation at 24 °C for an additional 2–4 days.

### GTT and ITT
For GTT, following overnight starvation for 12-16 h, mice were intraperitoneally injected with a glucose solution in saline at 1.5 g Kg$^{-1}$ body weight, and oral gavage with a glucose solution in saline at 2.6 g Kg$^{-1}$ body weight. Blood glucose levels were determined directly before and 15, 30, 60, and 120 min after glucose treatment. For ITT, after starvation for 3 h, mice were intraperitoneally injected with insulin in saline at 1 unit Kg$^{-1}$ body weight. Immediately before and 15, 30, 60, and 120 min post insulin injection, blood glucose levels were measured.

### Treadmill running assay
Mouse running performance was assessed using the treadmill running system from Columbus Instruments. The inclination angle was level. Before the running tests, mice were trained for three consecutive days at a speed of 10 m min$^{-1}$ to acquire running skills. During the tests, the speed was started from 10 m min$^{-1}$ for 20 min and increased every 20 min at 2 m min$^{-1}$ increments until reaching and being maintained at 20 m min$^{-1}$. Total running distance and running time were recorded at the point when the mice reached exhaustion.

### Adipocyte culture, differentiation, viral transduction, and siRNA transfection
C3H10T1/2 cells were obtained from ATCC and cultured in DMEM (Gibco, C11995500BT) containing 10% (v/v) FBS (VISTECH, SE100-011). After reaching 100% confluency, cell differentiation was induced by replacing the culture medium with an induction medium (DMEM containing 10% FBS, 0.5 mM IBMX, 125 µM indomethacin, 1 µM dexamethasone, 1 nM T3, and 20 nM insulin). Two days after induction, cells were cultured in a differentiation medium (DMEM containing 10% FBS, 1 nM T3, and 20 nM insulin). The cells were switched to a fresh differentiation medium every 2 days until fully differentiated. To establish Tfr1 or Npr3 knockdown stable cell lines, C3H10T1/2 cells were transduced with lentiviruses expressing shRNA targeting *Tfr1*, *Npr3*, and control vector, respectively, followed by subjection to puromycin selection and differentiation. For *Npr1* or *Npr2* knockdown in C3H10T1/2-derived adipocytes, *NPR1*, *NPR2* or *CTR* siRNA was transfected using RNAiMAX (Invitrogen) after differentiation for 6 days.

*Tfr1* shRNA targeting sequence: 5′-GACAATAACATGAAGGCTACT-3′;

*Npr3* shRNA targeting sequence: 5′-GATCGAGGTGCTTGTTCTATT-3′;

si*Npr1*−1 targeting sequence: 5′-CGCAAGGGCAGAGTTATCTACATCT-3′;

si*Npr1*−2 targeting sequence: 5′-CCAGGCTGAGGCATTTGATAGTGTT-3′;

si*Npr2*−1 targeting sequence: 5′-TAGCACTTCGAAGTGGTCCTTTCTA-3′;

si*Npr2*−2 targeting sequence: 5′- CCATCCCTGATGAACCTTATT-3′;

### Muscle cell culture and differentiation
C2C12 myoblasts were purchased from ATCC and cultured in DMEM containing 10% FBS (VISTECH, SE100-011). After reaching >90% confluency, myotube differentiation was induced by switching the culture medium to differentiation medium (DMEM containing 2% FBS). After full differentiation, myotubes were subject to RNA isolation and gene expression analysis.

### Preparation of Fc-tagged Musclin protein from Expi293F cells
Expi293F cells were transfected with pcDNA3.0-Musclin-Fc using PEI (Polysciences, 23966-2). At 24 h after transfection, 1 M sodium butyrate (1:1000) was added to suppress cell proliferation, followed by incubation for another 4–5 days. The residual cells and debris in the collected media were removed by centrifuging at 450 g (5 min) and 12,000 g (5 min), respectively. Musclin-Fc was purified with protein A/G agarose resin (YEASEN, 36403ES08), filtered with a 0.22 µm filter (Millipore), and stored at −80 °C for future use.

### OCR assay
Seahorse XFe96 Extracellular Flux Analyzer (Agilent) was used to measure oxygen consumption rates (OCRs) in C3H10T1/2 cells. In brief, fully differentiated C3H10T1/2 adipocytes were digested and seeded into the XFe96 cell culture microplate (Agilent) at a density of ~2 × 10$^4$ cells/well. After being pretreated with Musclin or vehicle for 5 h, the cells were treated with FSK (forskolin) in XF-DMEM for 1 h in a 37 °C incubator before analysis. The XF96 microplate was then loaded into a Seahorse XFe96 analyzer for equilibration and determination of the basal respiration rate. Through automatic pneumatic injection, oligomycin (2 µM), carbonyl cyanide 4-(trifluoromethoxy) phenylhydrazone (FCCP, 2 µM), and rotenone/actinomycin A (0.5 µM/0.5 µM) were added into the well in sequence. OCRs were recorded and the ATP production, maximal respiration, and the spare respiratory capacity-dependent OCRs were calculated. Cell viability was also quantified using the CCK8 kit (Beyotime) for normalization. The XF Cell Mito Stress test kit (Agilent) was used in this assay. Seahorse Wave Desktop and Controller 2.6.1 (Agilent) was used for data acquisition.

### Metabolic measurements
An NMR analyzer (NIUMAG, QMN06-090H) was applied to measure the body fat and lean mass. According to the manufacturer's protocols, the concentrations of Triglyceride (Sigma, TR0100), and NEFA (Wako Diagnostics, 294-63601) in plasma were assessed using commercial assay kits. An enzyme-linked immunosorbent assay (ELISA) kit from Crystal Chem (90080) was used to measure plasma insulin concentration.

### Detection of plasma Musclin, ANP, and CNP
For the detection of Musclin levels in mouse plasma, the plasma samples were pretreated with aprotinin (500 KIU ml$^{-1}$), and then measured using a radioimmunoassay kit (Beijing Sino-UK Institute of Biological Technology, HY-076). For the detection of Musclin levels in human plasma, a Human Musclin ELISA kit (CUSABIO, CSB-E12021h) was applied. For the detection of ANP levels in mouse plasma, a mouse ANP ELISA kit (Elabscience, E-EL-M0166c) was used. For the detection of CNP levels in mouse plasma, a Mouse CNP ELISA kit (Beijing Sino-UK Institute of Biological Technology, HY-NE022) was applied. The assays were performed according to the manufacturer's instructions.

### Histological analysis
Mouse tissues including liver, skeletal muscle, BAT, eWAT, and iWAT were dissected, fixed in formalin, followed by embedding in paraffin and cutting for tissue sections at 3-4 µm for liver, skeletal muscle, and BAT, or 6-8 µm for eWAT and iWAT. Tissue sections were then stained with hematoxylin and eosin, and subjected to image acquisition using

NIS Elements F 4.60 (Nikon). The cell size of iWAT was quantified using Image J 1.52q software (NIH).

## SEAP-Musclin binding assay

The SEAP-binding assay was performed as described[64]. Firstly, the vectors expressing SEAP or SEAP-Musclin fusion protein were transiently transfected into HEK293T cells. At 12 h after transfection, the culture medium was switched to serum-free. Conditioned medium (CM) was collected after incubation for an additional 2 days, followed by concentration using Centricon filters (Millipore). The concentrated CM containing SEAP or SEAP-Musclin was then applied to treat the indicated frozen tissue sections followed by incubation in a wet chamber at room temperature for 45 min. The sections were washed with PBS containing 0.1% Tween-20 for four times and fixed in a solution (3% formaldehyde, 20 mM HEPES (pH 7.4), and 60% acetone) for 10 min. After endogenous alkaline phosphatase was inactivated in a 65 °C water bath for 30 min, the signal derived from the enzymatic activity of SEAP-Musclin fusion protein was developed with NBT/BCIP substrate (Roche, 11681451001). CellSens Standard (Olympus) was used for image acquisition and data collection.

## RNA isolation and RT-qPCR

Total RNA from WAT was isolated using a commercially available kit (TIANGEN Biotech, DP430). Total RNA from other tissues including skeletal muscle, heart, brain, small intestine, liver, BAT, bone, and cultured cells was isolated following the standard method using TRIzol. RNA was then reverse transcribed using HiScript II Q RT SuperMix (Vazyme, R222-01), followed by qPCR analysis using SYBR Green (Roche). The relative gene expression level was calculated using the $2^{-\Delta\Delta Ct}$ method after normalization to ribosomal protein, large, P0 (Rplp0). The qPCR primers used are listed in Supplementary Data.

## RNA sequencing and bioinformatics

For 7 tissues from C57BL/6J WT mice, iWAT and BAT from MCK-Musclin and control mice and human muscles, RNA sequencing was performed in KAITAI-BIO (Hangzhou, China). Sequencing libraries were constructed from total RNA using SMART-RNAseq Library Prep Kit (Hangzhou KaiTai, AT4201). In brief, mRNA was isolated from total RNA with Sera-Mag Magnetic Olido(dT) particles, and then chemically fragmented. The fragmented RNA was reverse-transcribed into cDNA using a random primer containing a tagging sequence at their 3′ends. And the cDNA libraries were subsequently amplified using the KAPA high-fidelity DNA polymer. Quality of the libraries was validated by the 2100 Bioanalyzer (Agilent Technologies). Subsequently, high-throughput sequencing was performed using a NovaSeq 6000 (Illumina). Raw reads were filtered with *fastp* (V0.20.0, https://github.com/OpenGene/fastp) and *bowtie2* (V2.3.5, https://bowtie-bio.sourceforge.net/bowtie2/index.shtml). Filtered data were then aligned with *HISAT2* (V2.1.0, https://daehwankimlab.github.io/hisat2/download/)[65] under recommended parameters (--dta –phred64 unstranded –new-summary -x index −1 read_r1 −2 read_r2(PE)) to mm10 reference genome (Mus_musculus.GRCm38.90, https://hgdownload.soe.ucsc.edu /goldenPath/mm10/bigZips/) (for mouse tissue data) or to Homo_sapiens.GRCh38.p13 (for human data). The FPKM (fragments per kilobase of exon per million fragments mapped) was calculated using StringTie (V2.0, https://ccb.jhu.edu/software/stringtie/) to determine gene expression levels.

For iWAT from cold-acclimated and room temperature-housed control WT mice, RNA sequencing was performed in Majorbio Biopharm Biotechnology Co., Ltd. (Shanghai, China). The library was prepared using TruSeqTM RNA Sample Prep Kit (Illumina). Shortly, mRNA was isolated by oligo(dT) beads and then fragmented. After that, a SuperScript double-stranded cDNA synthesis kit (Invitrogen) with random hexamer primers (Illumina) was used to synthesize double-stranded cDNA which was subjected to end-repair,

phosphorylation and 'A' base addition according to library construction protocol. Libraries were size selected for cDNA target fragments of 300 bp on 2% Low Range Ultra Agarose, followed by PCR amplification using Phusion DNA polymerase (NEB). After quantified by TBS380, paired-end RNA sequencing was performed using the NovaSeq 6000 (Illumina). The raw reads were processed similarly and the TPM (transcripts per million reads) values were used to determine gene expression levels.

For human scWAT, RNA sequencing was performed in BGI (Shenzhen, China). Similarly, mRNA was purified using oligo(dT)-attached magnetic beads and was then fragmented. cDNA was generated by random hexamer-primed reverse transcription, and end-repaired through incubation with a-Tailing Mix and RNA Index Adapters. The obtained cDNA fragments were amplified using PCR followed by purification with Ampure XP Beads, which was validated on the 2100 Bioanalyzer (Agilent Technologies). Distinctively, further heat-denaturation and circularization by the splint oligo sequence of the PCR products from previous step were needed to get the final library. The library was finally amplified by phi29 to make DNA nanoball (DNB), >300 copies of one molecular, for sequencing on BGIseq500 platform. Pair end 100 base reads were generated for analysis. The sequencing data was filtered with *SOAPnuke* (V1.5.6)[66] to obtain clean data which were then aligned to reference genome (GCF_000001405.38_GRCh38.p12) with *HISAT2* (V2.1.0). Expression level of gene (FPKM value) was calculated by *RSEM* (V1.3.1, http://deweylab.biostat.wisc.edu/rsem)[67] to determine gene expression levels.

For all these sequencing data, differential expression analysis was performed using the *Deseq2* (v.1.20.0) package (*P*-value by Wald test)[68]. Gene Ontology (GO) and pathway grouping and enrichment studies were performed by *clusterProfiler* (V3.12.0, https://bioconductor.org/packages/release/bioc/html/clusterProfiler.Html)[69]. Results were visualized by *ggplot2* (V3.2.1, https://tidyverse.github.io/ggplot2-docs/index.html)[70] and *pheatmap* (V1.0.10, https://cran.r-project.org/web/packages/pheatmap/index.html). The RNA-Seq data of 7 tissues from C57BL/6J WT mice, and the RNA-Seq data of iWAT and BAT from MCK-Musclin and control mice have been deposited into Gene Expression Omnibus (GEO) database under accession code: GSE228218, and the RNA-Seq data of iWAT from cold-acclimated mice and control mice housed at room temperature have been deposited into Sequence Read Archive (SRA) database under accession code: PRJNA948712. The RNA-Seq datasets of human muscle and scWAT samples have been deposited in the Genome Sequence Archive[71] in National Genomics Data Center[72], China National Center for Bioinformation / Beijing Institute of Genomics, Chinese Academy of Sciences (GSA-Human: HRA004352).

## Immunoblotting analysis

Protein lysates from skeletal muscles and adipose tissues, or whole-cell lysates from cultured cells were quantified using BCA protein assay (Beyotime). Protein samples with equal amounts were separated by SDS-PAGE gels, followed by transferring to polyvinylidene difluoride (PVDF) membrane (Millipore) and membrane incubating with primary antibodies 4 °C overnight. After that, membranes were incubated with anti-rabbit or anti-mouse or anti-Rat IgG (whole molecule)-Peroxidase secondary antibodies diluted in TBS-T containing 5% milk. The final results were visualized with chemiluminescence (ECL) western blotting substrates. MiniChemi™610 (Mini Chemiluminescent/Fluorescent Imaging and Analysis System) with Sage Capture™ Imaging Software was used for immunoblot data collection. The primary antibodies used in this study were listed as follows: Rabbit polyclonal antibody to Musclin, generated with mouse Musclin protein and affinity purified (Abcam); total OXPHOS rodent WB antibody cocktail (1:1000, ab110413), and antibodies to Pgc1(α + β) (1:1000, ab72230), from Abcam; antibody to GAPDH (1:1000, sc-25778), from Santa Cruz;

antibodies to phospho-PKA substrates (1:1000, 9624), HSL (1:1000, 4107 s), phospho-HSL (ser660) (1:1000, 4126 s), and Hsp90 (1:1000, C45G5) from Cell Signaling; antibodies to Flag (1:1,000, M2, A8592) and α-tubulin (1:1,000, T6199), from Sigma; antibody to Tfr1 (1:1000, 13-6800), from Invitrogen; antibody to Ucp1 (1:1000, UCP11-A) from Alpha Diagnostic; antibody to Adrb3 (1:1000, A8607), from Abclonal; antibody to His tag (1:1000, A00186s), from GenScript; anti-rabbit (1:10000, A6154) or anti-mouse (1:10000, A4416) IgG (whole molecule)-Peroxidase secondary antibodies produced in goat, from Sigma.

### BioID and mass spectrometry

A proximity-dependent biotin identification (BioID) assay was used to identify the receptors for Musclin on the plasma membrane of adipocytes. In brief, the pCMV-BirA*, pCMV-signal peptide (Sp)-BirA*-flag or pCMV-Musclin-BirA* plasmids were transfected into the HEK293T cells. After transfection for 12 h, cells were switched to a serum-free medium for another 48 h, followed by the collection of CM enriched in BirA* or Sp-BirA*-flag alone, or Musclin-BirA* fusion protein. Then, the fully differentiated C3H10T1/2 cells were incubated with the CM supplemented with 80 µM biotin for 24 h. Cell membrane-associated proteins were separated through differential centrifugation and then pulled down using streptavidin agarose (Thermo Fisher, S951) at 4 °C with an overnight end-over-end rotation. The agarose beads were pelleted by centrifugation and then subjected to five times washing with washing buffer, followed by protein denaturation, SDS-PAGE gel electrophoresis, and Coomassie blue staining or silver staining (Pierce Silver Stain Kit #24612, Thermo Fisher). The gel fragments in pulldown samples containing differential protein bands as compared to control were collected for electrospray ionization tandem M.S. analysis on a Thermo Finnigan LTQ Orbitrap Instrument[73]. Proteome Discoverer version 1.4 (Thermo Scientific™) was used for mass spectrometry data analysis. The mass spectrometry proteomic data have been deposited to the ProteomeXchange Consortium via the PRIDE[74] partner repository with the dataset identifier PXD041019.

### Preparation of protein fractions from cell culture media

HEK293T cells were transiently transfected with plasmids expressing control or Musclin. Culture media were removed 24 h after transfection, and cells were then washed with PBS and incubated in serum-free DMEM (Gibco, C11995500BT) for another 24 h. Finally, cells and culture media were harvested separately. Cells were subjected to total protein lysate preparation. Culture media were centrifuged (12,000 g) for 5 min at 4 °C to get rid of cell debris. Trichloroacetic acid was then added to a final concentration of 10%, followed by incubation on ice for 1 h and centrifugation (12,000 g) for 20 min at 4 °C to precipitate the protein pellet. The obtained protein pellet was washed in cold acetone, air-dried, and finally resuspended in SDS-containing lysis buffer.

### Co-immunoprecipitation (Co-IP) assay

HEK293T cells were transiently transfected with plasmids expressing Musclin or Musclin-Fc, and Flag-tagged Tfr1 (Flag-Tfr1) or HA-tagged HSL (HA-HSL) for 48 h, followed by preparation of whole-cell protein lysates and immunoprecipitation. The input and IP samples were subjected to immunoblotting using antibodies against Musclin (Abcam), Flag (Sigma, A8592, M2), or HSL (Cell signaling, 4107s).

### Detection of cellular cAMP levels

cAMP was extracted with lysis buffer (50 mM Tris-HCl, pH7.5; 150 mM NaCl; 5 mM EDTA; 1% Triton-X 100; 0.5 mM IBMX) from differentiated C3H10T1/2 adipocytes and was then quantified using a competitive enzyme immunoassay kit according to the manufacturer's instructions (GenScript, L00460). Cell proteins were quantified using a BCA protein assay kit (Beyotime) for normalization.

### Detection of tissue and cellular cGMP levels

For the detection of cGMP levels in mouse iWAT and BAT, a cGMP ELISA kit (Cayman, 581021) was applied according to the manufacturer's instruction. Tissue weight was used for normalization.

For the detection of cellular cGMP levels, cGMP was extracted with lysis buffer (20 mM Tris-HCl (pH7.5), 150 mM NaCl, 1 mM EDTA, 1 mM EGTA, 1% Triton-X 100, 2.5 mM sodium pyrophosphate, 1 mM β-glycerophosphate, 1 mM $Na_3VO_4$, 1 µg/ml leupeptin, 0.5 mM IBMX, 2 mM PMSF) from HUVECs or differentiated C3H10T1/2 adipocytes, and was then quantified using a competitive enzyme immunoassay kit according to the manufacturer's instructions (GenScript, L00461).

### Tissue iron assay

A standard chromogen method was used to measure the tissue non-heme iron described as previously described[75]. Briefly, about 50 mg tissue was homogenized in Millipore water (100 µL/10 mg tissue), then mixed with equal volume acid solution (10% trichloroacetic acid, 1 M HCl). The mixture was then incubated at 95 °C for 1 h, followed by centrifugation to pellet the debris. 50 µL supernatant was then loaded into a transparent 96-well plate followed by adding 50 µL assay reagent (1 mM Ferrozine, 3 M sodium acetate, 1% mercaptoacetic acid). After mixing, the plate was incubated at 37 °C for 1 h. Optical density was measured at 562 nm using a spectrophotometer plate reader (Thermo Fisher, Multiskan™ SKY). The amount of iron was normalized to the wet tissue weight, and presented as $\mu g\ g^{-1}$ wet tissue.

### Statistical analysis

All statistical analyses were performed using GraphPad Prism 9 software. Data are presented as mean ± SD or mean ± SEM. A two-tailed unpaired Student's t-test was used to evaluate the statistical differences for comparisons between two groups, and one-way analysis of variance (ANOVA) with Tukey's multiple comparisons test was used for comparisons of more than two groups. Two-way ANOVA with Sidak's multiple comparisons test was applied for multiple comparisons involving two independent variables. For human studies, data represent mean ± SEM, and *Spearman* correlation analysis was applied for the correlation between muscle *MUSCLIN* expression and BMI. n values represent biological replicates for cell experiments, or mouse number for in vivo animal studies, or human subject number unless otherwise indicated. Specific details for the n value are noted in each figure legend.

### Reporting summary

Further information on research design is available in the Nature Portfolio Reporting Summary linked to this article.

## Data availability

RNA-Seq data of 7 tissues from C57BL/6 J WT mice, and the RNA-Seq data of iWAT and BAT from MCK-Musclin and control mice are available in GEO database under accession code: GSE228218. RNA-Seq data of iWAT from cold-acclimated mice and their controls are available in SRA database under accession code: PRJNA948712. RNA-Seq datasets of human muscle samples and of human scWAT samples have been deposited in the Genome Sequence Archive in National Genomics Data Center, China National Center for Bioinformation/Beijing Institute of Genomics, Chinese Academy of Sciences under accession code GSA-Human: HRA004352 that are publicly accessible. The mass spectrometry proteomics data have been deposited to the ProteomeXchange Consortium via the PRIDE[74] partner repository with the dataset identifier PXD041019, which can be freely accessed. The images representing human participants, mouse models, skeletal muscle, myotube, adipose tissue, RNA-Sequencing, qPCR analysis, culture dish, and adipocyte shown in Figs. 1a, 3a, 5a, 6d, 8a and Supplementary Fig. 1b, 5a are created with BioRender.com. All other data and image files are

available from the "Source Data" file. Source data are provided with this paper.

## Code availability

All software and codes used in this study are open-source and publicly available.

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

## Acknowledgements

The authors thank Drs. Dante Neculai (Zhejiang University), Christopher R Wood (Zhejiang University), and Zhe Yu Zhang (Zhejiang University) for critical proofreading and editing of the manuscript. We thank the Meng lab members for the helpful discussion and technical support for this study. We also thank Drs. Peng Wu and Wei Chen (Zhejiang University) and Dr. Cheng Ma (the Core Facilities of Zhejiang University School of Medicine) for technical support. We also appreciate Dr. Nan Xu (Henan University) for providing the HUVECs (purchased from ATCC). This work was supported by grants from the National Natural Science Fund for Excellent Young Scholars of China (81722012), the National Key Research and Development Programme of China (2018YFA0800403, 2021YFC2701903), the Training Program of the Major Research Plan of the National Natural Science Foundation of China (91857110), the National Natural Science Foundation of China (81670740, 32000817), the Zhejiang Provincial Natural Science Foundation of China (LZ21H070001, LHDMD22H02001), the Construction Fund of Medical Key Disciplines of Hangzhou (OO20200055), the Innovative Institute of Basic Medical Sciences of Zhejiang University, and the Fundamental Research Funds for the Central Universities to Z.-X.M. This study was also supported by grants from the National Natural Science Foundation of China (82100904), the Zhejiang Provincial Natural Science Foundation of China (LQ21C110001), and the China Postdoctoral Science Foundation (2020M671696) to S.H.. This work was also supported by grants from the National Key Research and Development Programme of China (2016YFC1305301, 2021YFC2701900), the National Natural Science Foundation of China (81570759, 81270938), the Key Disciplines of Medicine (Innovation discipline, 11-CX24) and the Fundamental Research Funds for the Central Universities (2017XZZX001-01) to J.F. The authors gratefully acknowledge the support from K.C. Wong Education Foundation.

## Author contributions

Z.-X.M. conceived and designed the research. L.J., S.H., X.Lv, X. Li, H.K., Z.C., W.P., M.Y., L.M., C.L., S.M., and X.P. performed the experiments. Z.Z., Z.Y., T.L., X.Q., and Q.W. conducted the bioinformatics analysis. P.S., Y.F., J.L., F.W., L.X., and X.Z. provided critical reagents for this study. J.D.L. contributed to the discussion and data interpretation. J.F. contributed to the discussion. Z.-X.M., J.D.L., L.J., and S.H. wrote the manuscript with help from the other authors.

## Competing interests

The authors declare no competing interests.

## Additional information

[1]Department of Pathology and Pathophysiology and Department of Cardiology of the Second Affiliated Hospital, Zhejiang University School of Medicine, Hangzhou, China. [2]Department of Endocrinology, Children's Hospital, Zhejiang University School of Medicine, Hangzhou, China. [3]Key Laboratory of Disease Proteomics of Zhejiang Province, Zhejiang University School of Medicine, Hangzhou, China. [4]Department of Geriatrics, Affiliated Hangzhou First People's Hospital, Zhejiang University School of Medicine, Hangzhou, China. [5]Chronic Disease Research Institute, Zhejiang University School of Public Health, Hangzhou, China. [6]Department of Sport Medicine, The Lianyungang First People's Hospital, Affiliated Hospital of Xuzhou Medical University, Affiliated Hospital of Kangda College of Nanjing Medical University, Lianyungang, China. [7]Life Sciences Institute and Department of Cell and Developmental Biology, University of Michigan, Ann Arbor, USA. [8]Department of Biology and Chemistry, College of Science, National University of Defense Technology, Changsha, China. [9]Department of Endocrinology and Metabolism, The Second Affiliated Hospital, Zhejiang University School of Medicine, Hangzhou, China. [10]Department of Endocrinology, The Second Affiliated Hospital of Soochow University, Suzhou, China. [11]The Second Affiliated Hospital, School of Public Health, Zhejiang University School of Medicine, Hangzhou, China. [12]Guangdong Provincial Key Laboratory of Microbial Culture Collection and Application, State Key Laboratory of Applied Microbiology Southern China, Institute of Microbiology, Guangdong Academy of Sciences, Guangzhou, China. [13]Department of Biochemistry and Molecular Cell Biology, Shanghai Key Laboratory for Tumor Microenvironment and Inflammation, Key Laboratory of Cell Differentiation and Apoptosis of National Ministry of Education, Shanghai Jiao Tong University School of Medicine, Shanghai, China. [14]These authors contributed equally: Lu Jin, Shuang Han, Xue Lv, and Xiaofei Li. ✉e-mail: fjf68@zju.edu.cn; zxmeng@zju.edu.cn

