## [Peer Review File · Nature Communications]

The muscle-enriched myokine Musclin impairs beige fat thermogenesis and systemic energy homeostasis via Tfr1/PKA signaling in male miceREVIEWER COMMENTS

Reviewer #1 (Remarks to the Author):

Jin et al. reported that muscle-derived myokine Musclin is a negative regulator of beige fat thermogenesis in mice. The authors first showed in humans and mice that circulating levels of Musclin were higher in obese subjects than lean/non-obese subjects. Then the authors showed that overexpression of Musclin attenuated adaptive thermogenesis and led to body weight gain and glucose intolerance on a high-fat diet. Similarly, AAV-induced overexpression of Musclin in mice impaired adaptive thermogenesis and induced obesity on a high-fat diet. The authors next employed BirA-based proteomics and identified transferrin receptor 1 (Trf1) as a receptor that mediates the effect of Musclin on inhibiting PKA signaling in adipocytes. Finally, the authors demonstrated the role of Musclin in energy metabolism by two independent loss-of-function studies – by studying muscle-specific KO mice and using a neutralizing antibody for Musclin.

The paper provided a comprehensive story regarding the role of Musclin as a negative regulator of adaptive thermogenesis and energy homeostasis. A possible link to human metabolism was also significant. However, several issues are identified in the present manuscript. The authors should provide compelling evidence to address these concerns.

1. The present paper showed that skeletal muscle expressed musclin at the highest levels. On the other hand, previous papers showed the expression in other cell types. Skeletal muscle-specific Musclin KO mice provided a terrific opportunity for the authors to address the source of Musclin in the circulation quantitatively. The authors should measure plasma Musclin levels in muscle KO mice under a regular chow diet and a high-fat diet. Similarly, the authors should examine the plasma levels at thermoneutrality and cold exposure. These results are crucial to validate the title and conclusion of this paper.
2. The authors should determine the affinity of Musclin to Trf1. This is important because previous work by Moffatt et al. (Journal of Biological Chemistry Vol.282, No.50, pp.36454–36462, 2007) reported that Osteocrin/Musclin binds specifically to the NP peptide receptor-C (NPR-C) with a Kd of ~5 nM. This is another critical experiment to show that Trf1 mediates the inhibitory effect of Musclin on fat thermogenesis.
3. The authors mostly focused on iWAT as a primary target of Musclin. However, infrared imaging showed changes in iBAT thermogenesis. The authors should examine iBAT phenotype of muscle-Musclin transgenic mice and KO mice. If the effect of Musclin is selective to iWAT, the authors should explore the underlying mechanism – e.g., are the receptors (Trf1 or NPR-c) more abundant in iWAT than iBAT?
4. The data consistency: the authors should show energy expenditure and body weight of Musclin-MKO mice. Similarly, the authors should examine body-weight and tissue weight of fat-specific Trf1 KO mice.
5. Given the current consensus regarding the energy expenditure analysis, the authors should show 1) EE without normalization by body weight, and 2) assess if a difference in EE depends on temperature (e.g., Speakman et al. Front. Physiol., 14 March 2013).

Reviewer #2 (Remarks to the Author):

Jin et al. evaluated in depth the possible role of musclin in fat thermogenesis. The authors used several approaches to upregulate or downregulate the gene/protein, showing concordant changes and in the direction expected.

Major comments

*Being age and sex important contributors to muscle mass, no mention is made regarding this critical confounding factors in the recruiting of human samples. In mice experiments, it is stated that male mice were used for all experiments unless otherwise indicated, but at the same time that "...those with the same genotype, gender and similar body weight were allocated into different groups". A thorough statement of age and sex would need to be specified in all experiments in vivo or regarding the samples obtained from human subjects.

*The phenotype and potential confounding factors of human donors is very poor. As the subjects recruited underwent ligament repair or reconstruction, did the previous injury play a role in the current findings? In fact, findings shown in Figure 1b are quite heterogeneous.

*Given the alternative name of musclin (osteocrin), it is possible that the bone could also be playing a role in the observed effects. There is mounting literature about the influences exerted by different bone-derived molecules in systemic metabolism. It seems that virtually no data has been reported about this possible influence in this paper.

Minor suggestions

*Irisin is known to promote thermogenesis but their circulating levels are increased in sedentary subjects with obesity. Ideally, the information about physical activity would need to be provided.

*Page 7, line 181: It is mentioned that muscle-specific transgenic expression of Musclin had a modest effect on the development, metabolism, and exercise performance of skeletal muscle per se, but no information virtually is provided to sustain this statement.

*Page 11, line 298: harvested scWAT. Did the authors harvest scWAT or simply obtained the adipose tissue samples.

*Page 14, line 401. Data not shown. Why?

Reviewer #3 (Remarks to the Author):

Lu Jin et al. investigated the impact of the myokine musclin on beige fat thermogenesis. They found increased Musclin serum levels in individuals with a BMI>24 and in mouse models of diabetes and obesity. Using transgenic muscle specific (but also systemic AAV- based) Musclin overexpression as well as muscle specific Musclin knock-out mice, the authors demonstrate that Musclin inhibits the development of beige fat during cold exposure and high fat diet administration, triggering exacerbation of obesity under the latter condition. Mechanistically, the authors suggest endocrine crosstalk between skeletal muscle and inguinal white adipose tissue, which they found to be rather selectively bound by Musclin. Using an unbiased proteomics based approach, the authors identified the transferrin receptor as binding target of Musclin at inguinal WAT. Intracellularly Musclin inhibits the cAMP formation and PKA activation.

This is a very interesting manuscript addressing a timely topic using elegant genetic and translational mouse models with results that could be of high interest for treating obesity or metabolic syndrome. A previously unknown function of Musclin is described. This being said, the authors almost completely

neglect previous knowledge describing Musclin as inhibitor of the natriuretic peptide degrading receptor NPRC, and therefore as an amplifier of natriuretic peptide signaling. The authors need to address these points. In addition, some parts of the mechanism remain unclear. More specifically, the authors need to address the following points:

Major:

- 1) Musclin competes with ANP, BNP and CNP for binding to NPRC. All natriuretic peptide receptors are also expressed in adipose tissue and have a known role in lipolysis and perhaps fat browning (e.g. Bordicchia et al., JCI, 2012). How are the serum levels of ANP (and CNP?) in Musclin overexpressing or knock-out mice? How are cGMP levels changed in iWAT or BAT after manipulation of Musclin during high fat diet or cooling? Are cAMP level changes by Musclin the result of changes of cGMP levels, which could happen for example via PDE3.
- 2) Are the effects of Musclin on adipocytes dependent on the expression of Npr1 or Npr2?
- 3) How does Musclin selectively target iWAT, but not eWAT or other tissues? Is Tfr1 expression different between the different tissues (for example higher in iWAT than eWAT?)
- 4) How are serum Glucose levels changed by Musclin? Is this mediated by its effect on iWAT or is this through its effect on skeletal muscle?
- 5) How are adipocyte cAMP levels controlled by Tfr1?
- 6) The authors suggest that Musclin triggered effects via Tfr1 are iron independent, have they checked some iron homeostasis (e.g. ferritin levels)?
- 7) I am still struggling with the seemingly contradictory effect of systemic musclin KO mice (Subbotina E et al., PNAS, 2015) showing less mitochondria in skeletal muscle and the current study implying more mitochondria in adipose tissue. Can the authors elaborate on this?

Minor: The authors do not mention at all that Musclin was also discovered as exercise dependent myokine with cardioprotective features and they neglect some of the relevant recent literature (Kattih B et al., JAHA, 2022; Szaroszyk et al., Nat comm, 2022).

Reviewer #4 (Remarks to the Author):

In their manuscript entitled "The muscle-enriched myokine Musclin impairs beige fat thermogenesis and systemic energy homeostasis via Tfr1/PKA signaling" the authors report that mouse and human obesity is associated with elevated muscle expression and circulating levels of Musclin. The muscle expression of Musclin was also detected to be environment temperature sensitive, and beige fat was identified as primary site of Musclin action. Overexpression of Musclin was also detected to inhibit beige fat thermogenesis mimicking the obesity and metabolic disorders, whereas Musclin ablation promoted beige fat thermogenesis and improved metabolic homeostasis. The authors also tried to find the molecular mechanisms behind this and find that Musclin bind to transferrin receptor 1 (Tfr1) and antagonises Tfr1-mediated cAMP/PKA-dependent thermogenic induction in beige adipocytes. Overall, this is a very large and extensive work utilising multiple different sample types and methods. However, several issues need to be solved before the manuscript is ready for publication.

Major criticism:

Figure 1. b-c: The authors should comply with standard guidelines for RNA-Seq data reporting and should include the actual data as a supplementary table and also should ensure that the actual raw data is submitted by the time of publication to a public data repository (it is difficult to judge this, as no access was made available for the reviewers)

Figure 1. g: there is a horizontal white line present in the Musclin plot. Original uncropped images of this WB as well as other WBs should be included as a supplementary images.

Figure 2. d: could perhaps use some more replicates than one as $n \geq 6$ animals were most likely used in the actual analyses.

Figure 3. a-d: the whole RNA-Seq (quantitative) data should be presented in a supplementary table.

Additionally, I have serious concerns of the used cutoff values ($p < 0.05$, $\log_2(\text{fold change}) > 0.3$). These are significantly lower than what normally are used and it is impossible to say if the changes reported actually are significant or biologically relevant.

Figure 6. d: The proximity labeling is highly dependent on the localisation of the BirA*-tagged protein and choosing a appropriate control requires some thought. The authors use the BirA* tag alone as a control for secreted Musclin-BirA*. To my knowledge the BirA*-tag alone is not well secreted to the cell culture media and therefore is not a great control. Therefore the experiment should be redone using a secreted BirA* control.

Additionally the mass spectrometry data (after the database searches) should be added as a supplementary table.

Minor criticism:

At parts the manuscript would benefit from language editing.

Responses to the reviewers' comments (original comments are in blue)

We are deeply grateful to the reviewers for their enthusiasm and helpful comments and suggestions on improving the scope and impact of our original manuscript. Key issues raised include plasma Musclin levels in control and muscle-specific Musclin KO (Musclin-MKO) mice, as well as in WT mice at thermoneutrality or cold exposure, the binding affinity of Musclin to Tfr1, the effect of Musclin on iBAT, energy expenditure in control and Musclin-MKO mice, circulating levels of ANP and CNP in muscle-specific Musclin transgenic (MCK-Musclin) or Musclin-MKO mice and their respective control mice, effects of Musclin on cGMP content in iWAT and BAT during HFD feeding or cold exposure, the involvement of Npr1 or Npr2 in the metabolic regulation by Musclin in adipocytes, the potential reasons for the selectivity of Musclin action on iWAT, the underlying mechanism of how Tfr1 controls intracellular cAMP levels, the effect of Musclin on iron homeostasis, the impact of Musclin-MKO on mitochondria in skeletal muscle, and Musclin-BirA* BioID study using a secreted form of BirA* as the control. We have now performed extensive new studies to essentially address all the concerns raised by the reviewers and included the new data in **Figures 1g, 2b-2d, 2f, 2i, 3b-3d, 3f, 4h, 4i, 6l, 7b, 7g, 7j-7m, 8a-8l** and **Supplementary Figures 1c, 2a-2c, 4b, 6b, 6g, 7b, 7f-7l** in the revised manuscript. Furthermore, additional supporting data are presented as **Response Figures 1-21** in this letter.

Point-to-point responses to the reviewers' questions are detailed below.

Reviewer #1 (Remarks to the Author):

Jin et al. reported that muscle-derived myokine Musclin is a negative regulator of beige fat thermogenesis in mice. The authors first showed in humans and mice that circulating levels of Musclin were higher in obese subjects than lean/non-obese subjects. Then the authors showed that overexpression of Musclin attenuated adaptive thermogenesis and led to body weight gain and glucose intolerance on a high-fat diet. Similarly, AAV-induced overexpression of Musclin in mice impaired adaptive thermogenesis and induced obesity on a high-fat diet. The authors next employed BirA-based proteomics and identified transferrin receptor 1 (Tfr1) as a receptor that mediates the effect of Musclin on inhibiting PKA signaling in adipocytes. Finally, the authors demonstrated the role of Musclin in energy metabolism by two independent loss-of-function studies – by studying muscle-specific KO mice and using a neutralizing antibody for Musclin.

The paper provided a comprehensive story regarding the role of Musclin as a negative regulator of adaptive thermogenesis and energy homeostasis. A possible link to human metabolism was also significant. However, several issues are identified in the present manuscript. The authors should provide compelling evidence to address these concerns.

1. The present paper showed that skeletal muscle expressed musclin at the highest levels. On the other hand, previous papers showed the expression in other cell types. Skeletal muscle-specific Musclin KO mice provided a terrific opportunity for the authors to address the source of Musclin in the circulation quantitatively. The authors should measure plasma Musclin levels in muscle KO mice under a regular chow diet and a high-fat diet. Similarly, the authors should examine the plasma levels at thermoneutrality and cold exposure. These results are crucial to validate the title and conclusion of this paper.

We appreciate the reviewer's enthusiasm about our study and his/her insightful suggestions. According to the reviewer's suggestion, we measured plasma Musclin levels in control (Musclin^{fl^{ox}/fl^{ox}}) and Musclin-MKO mice under a regular chow diet and a high-fat diet (HFD), as well as the plasma levels of Musclin in wild-type mice at thermoneutrality and cold exposure. As expected, the Musclin concentrations in plasma significantly decreased in Musclin-MKO mice either fed with a regular chow diet (New **Fig. 7b**) or a HFD (**Response Fig. 1**). Consistent with the alterations of Musclin mRNA and protein levels in skeletal muscle (New **Fig. 2a-c**), the plasma Musclin levels were markedly increased at thermoneutrality, while significantly decreased after cold exposure (New **Fig. 2d**).

Response Fig. 1 Plasma Musclin levels in Musclin^{fl^{ox}/fl^{ox}} and Musclin-MKO mice fed with HFD. Data represent mean \pm SEM ($n = 9-14$ mice per group). * $p < 0.05$, two-tailed unpaired Student's t -test.

2. The authors should determine the affinity of Musclin to Tfr1. This is important because previous work by Moffatt et al. (Journal of Biological Chemistry Vol.282, No.50, pp.36454–36462, 2007) reported that Osteocrin/Musclin binds specifically to the NP peptide receptor-C (NPR-C) with a K_d of ~ 5 nM. This is another critical experiment to show that Tfr1 mediates the inhibitory effect of Musclin on fat thermogenesis.

We thank the reviewer for raising this critical issue. We have read the paper mentioned above carefully, in which Moffatt et al. used ^{125}I -Labeled Musclin-(83-133) for intact cell binding studies in pre-chondrocytic ATDC5 cells¹. In this study, we have also tried to measure the binding affinity of Musclin to Tfr1 and the results are shown in **Response Fig. 2**. Biosensor system has been extensively applied for the protein interaction study *in vitro*²⁻⁴. Here, we used the surface plasmon resonance (SPR) instrument (Biacore 3000, GE Healthcare) for measuring the binding affinity between Musclin and Tfr1. Since Tfr1 is well known as the transferrin receptor, we used human holo-transferrin (Sigma T0665) as the positive control. Human holo-transferrin, purified mouse Tfr1 extracellular domain (ECD) fragment with His-tag on its N-terminus (His-mTfr1-ECD), and Musclin with His-tag on its C-terminus (Musclin-His) were validated by Coomassie blue staining and western blot analyses (**Response Fig. 2a-c**). Previous studies indicated that the affinity of transferrin to Tfr1 was affected by the pH values of the binding reaction buffers and the approaches of Tfr1 immobilization^{3,4}. In this study, purified His-mTfr1-ECD in 20 mM sodium acetate (pH 6.0) was covalently immobilized using standard amine coupling chemistry on a CM5 chip (Biacore, BR100012). Holo-transferrin or Musclin containing solutions were injected at room temperature in HBS-EP+ (10 mM HEPES [pH 7.4], 150 mM NaCl, 3 mM EDTA, and 0.005% v/v polysorbate P20) buffer (BR100669, GE Healthcare). Consistent with the previous report², the human holo-transferrin binds specifically to Tfr1 with a K_d of 642 nM (**Response Fig. 2d**). We then measured the binding affinity of Musclin to Tfr1, and found that compared to holo-transferrin, Musclin-His showed a stronger binding affinity ($K_d = 36.6$ nM) to Tfr1 (**Response Fig. 2e**). These results demonstrate that both Npr3 and Tfr1 may serve as the receptor for Muslin in adipocytes.

To test the functional relevance of Npr3 in mediating the effects of Muslin on the thermogenic metabolism of adipocytes, we treated differentiated adipocytes derived from C3H10T1/2 cells transduced with retroviral vectors expressing scramble (Scrb) shRNA or shRNA targeting Npr3 (Npr3 shRNA) with or without Musclin. Consistent with previous studies, knockdown of Npr3 decreased the expression of adipocyte differentiation-associated genes such as *Cebpa* and *Cebpb* (**Response Fig. 2f**). However, similar to the effects in Scrb groups, Musclin significantly decreased the mRNA expression of genes involved in glucose and lipid metabolism in Npr3 knockdown adipocytes (**Response Fig. 2g**). By contrast, knockdown of Tfr1 almost completely abolished the inhibitory effects of Musclin on the expression of genes involved in glucose and lipid metabolism in adipocytes (New **Fig. 6m**). Together, these data provide evidence supporting the conclusion that Tfr1 mediates the inhibitory effects of Muslin on adipose tissue thermogenic metabolism.

Response Fig. 2 The binding affinity analysis of Musclin to Tfr1. a Coomassie blue staining of indicated purified proteins. **b** Immunoblots of purified His-mTfr1-ECD protein using anti-Tfr1 antibody (left) or anti-His antibody (right). **c** Immunoblots of purified Musclin protein using anti-Musclin antibody. **d-e** SPR analysis of the binding affinity of holo-transferrin (**d**) or Musclin-His (**e**) to Tfr1. His-mTfr1-ECD was directly immobilized on a CM5 chip. His-mTfr1-ECD, mouse Tfr1 extracellular domain (ECD) fragment with His-tag on its N-terminus, purified from insect baculovirus vector expression system in insect cells; Musclin-His, Musclin with His-tag in its C-terminus, purified from conditioned medium of 293F cells. **f** Relative gene expression in differentiated C3H10T1/2-derived adipocytes transduced with retroviral vectors expressing scramble (Scrb) or shRNA targeting *Npr3* (*Npr3* shRNA). Data represent mean \pm SD ($n = 3$ biological replicates). $**p < 0.01$, $***p < 0.001$, two-tailed unpaired Student's *t*-test. **g** Relative gene expression in differentiated C3H10T1/2-derived adipocytes transduced with retroviral vectors expressing scramble (Scrb) or shRNA targeting *Npr3* (*Npr3* shRNA) and treated with vehicle or Musclin-Fc. Fully differentiated adipocytes were pretreated with vehicle or Musclin-Fc for 30 min, followed by forskolin treatment for 6 h. Data represent mean \pm SD ($n = 3$ biological replicates). $***p < 0.001$, one-way ANOVA with Sidak's multiple comparisons.

3. The authors mostly focused on iWAT as a primary target of Musclin. However, infrared imaging showed changes in iBAT thermogenesis. The authors should examine iBAT phenotype of muscle-Musclin transgenic mice and KO mice. If the effect of Musclin is selective to iWAT, the authors should explore the underlying mechanism – e.g., are the receptors (Tfr1 or NPR-c) more abundant in iWAT than iBAT?

We thank the reviewer for the insightful comments. Both interscapular brown adipose tissue (iBAT) and subcutaneous iWAT are known to play critical roles in thermogenesis and body temperature homeostasis. In this study, we observed that skeletal muscle-specific transgenic expression of Musclin (MCK-Musclin) significantly inhibited body temperature maintenance in mice, and this inhibitory effect was more prominent during chronic cold exposure when the beige adipocytes in iWAT are more actively engaged to play an important role in thermogenesis and body temperature maintenance. These observations led us to focus more on iWAT. Consistent with this hypothesis, the mRNA and protein expression of thermogenic and metabolic genes was markedly decreased in iWAT from MCK-Musclin mice compared to control mice in response to one week of cold exposure at 8°C (**Fig. 3b, c, e, f**). However, RNA-Seq analysis revealed that expression of genes involved in lipid and glucose metabolism and thermogenesis remained largely unaltered in iBAT from MCK-Musclin mice following one week of cold exposure at 8°C as compared to controls (**Supplementary Fig. 3a-c**). These data suggest that iWAT is more likely to be the major target of Musclin in thermogenic regulation during chronic cold exposure.

Per the reviewer's suggestion, to further clarify the effects of Musclin on iBAT function in maintaining body temperature and metabolic homeostasis, during the revision, we extensively assess the metabolic parameters in iBAT from control and MCK-Musclin mice under different conditions: (1) Under normal housing and feeding condition, although there was no significant difference in iBAT tissue weight between control and MCK-Musclin mice (**Fig. 2i**, right), significantly larger lipid droplets and fewer multilocular structures were observed in MCK-Musclin mice (**Response Fig. 3a, b**). In addition, we observed a modest downregulation of the mRNA and protein expression of genes related to thermogenesis, such as *Ebf2*, *Ucp1*, and *Pgc1 β* (**Response Fig. 3c, d**). (2) Under HFD feeding condition, while tissue weight of iBAT was similar between the two groups (**Response Fig. 4a**), MCK-Musclin mice exhibited larger lipid droplets (**Fig. 4h**) and more profound inhibitory effects on thermogenic gene expression (**Response Fig. 4b**) in iBAT compared to those from control mice. (3) Moreover, we also examined the iBAT phenotype in HFD-fed Musclin flox/flox and Musclin-MKO mice. As expected, Musclin-MKO mice have lower iBAT weight, smaller lipid droplets, and more multilocular structures, as well as trending higher levels of *Ucp1* protein in iBAT (**Response Fig. 5**). Together, these new findings suggest that Musclin may also act on iBAT to contribute to thermogenic regulation during chow diet or HFD feeding at room temperature, while iWAT may

serve as the major target for Musclin in the thermogenic regulation and metabolic homeostasis.

As we know, adipose tissue contains not only adipocytes but also other cell types in the stromal vesicular fraction (SVF). To further examine adipocytes from which fat depots are targeted by Musclin, we isolated adipocytes from eWAT, iWAT, and iBAT, and performed western blotting analyses of its receptor Tfr1 (identified in the present study). Consistent with our hypothesis, in mice housed at room temperature, adipocytes from both iWAT and BAT express higher levels of Tfr1 protein than those from eWAT (**Response Fig. 6a**). Remarkably, in mice under chronic cold exposure condition, Tfr1 protein levels in adipocytes from iWAT were much more abundant than those in adipocytes from iBAT and eWAT (**Response Fig. 6b**).

Response Fig. 3 BAT phenotype in MCK-Musclin and control mice fed with a regular chow diet. **a** Representative H&E staining images of BAT sections from WT and MCK-Musclin mice fed with a normal chow diet at room temperature. **b** Quantification of the total area of lipid droplets per view. Two BAT sections were obtained from each mouse, and four views for each mouse were captured. Each dot in the graph represents the average total lipid droplet area per view. Data represent mean \pm SEM ($n = 5-7$ mice per group). * $p < 0.05$, two-tailed unpaired Student's t -test. **c** qPCR analysis of gene expression in BAT from control and MCK-Musclin mice. Data represent mean \pm SEM ($n = 4-7$ mice per group). * $p < 0.05$, ** $p < 0.01$, two-tailed unpaired Student's t -test. **d** Immunoblots of total protein lysates of BAT from control and MCK-Musclin mice.

Response Fig. 4 BAT phenotype in control and MCK-Musclin mice fed with HFD. **a** Tissue weights of BAT. Data represent mean \pm SEM ($n = 6-8$ mice per group). **b** qPCR analysis of indicated gene expression in BAT. Data represent mean \pm SEM ($n = 6-8$ mice per group). * $p < 0.05$, ** $p < 0.01$, *** $p < 0.001$, two tailed unpaired Student's t -test.

Response Fig. 5 BAT phenotypic study in control and Musclin-MKO mice fed with HFD. **a** Tissue weight of BAT. Data represent mean \pm SEM ($n = 8-11$ mice per group). * $p < 0.05$, two-tailed unpaired Student's t -test. **b** Representative H&E staining images of BAT tissue sections from Musclin-MKO and control mice fed with HFD (left panel) and average total area of lipid droplets per view (right panel). Two BAT sections were obtained from each mouse, and four views for each mouse were captured. Each dot in the graph represents the average of the total area of lipid droplets per view. Data represent mean \pm SEM ($n = 7$ mice per group). * $p < 0.05$, two-tailed unpaired Student's t -test. **c** qPCR analysis of gene expression in BAT from

Musclin^{flox/flox} and Musclin-MKO mice fed with HFD. Data represent mean \pm SEM ($n = 9$ mice per group). Two-tailed unpaired Student's *t*-test. **d** Immunoblots of total protein lysates of BAT from Musclin^{flox/flox} and Musclin-MKO mice.

Response Fig. 6 Immunoblot analysis of Tfr1 protein levels in mature adipocytes of eWAT, iWAT and BAT. **a** Immunoblots of Tfr1 protein in mature adipocytes of eWAT, iWAT and BAT from mice housed at room temperature. **b** Immunoblots of Tfr1 protein in mature adipocytes of eWAT, iWAT, and BAT from mice following cold exposure at 6°C for 1 week. Total protein lysates of iWAT from Tfr1^{flox/flox} and Tfr1^{flox/flox} Adipoq-CreERT2 (Tfr1-iAKO) mice were used as controls for the validation of Tfr1 antibody.

4. The data consistency: the authors should show energy expenditure and body weight of Musclin-MKO mice. Similarly, the authors should examine body-weight and tissue weight of fat-specific Tfr1 KO mice.

We appreciate the reviewer's good suggestions. We have monitored the body weight (BW) of control and Musclin-MKO mice during HFD feeding, and no significant difference was observed between the two groups (new **Supplementary Fig. 7f**).

For the energy expenditure measurement, Musclin-MKO and control mice were subjected to metabolic cage studies using the TSE PhenoMaster system. As shown in new **Fig. 7j-l** and **Supplementary Fig. 7h-j**, Musclin-MKO mice displayed significantly higher oxygen consumption and energy expenditure rates in response to adaptation at room temperature (24°C) for 1 week following housing in the thermoneutrality at 30°C for 1 week. However, no significant difference in food intake and wheel-running activity was observed between control and Musclin-MKO mice.

Per the reviewer's suggestion, we have also examined the body weight (BW) and weights of major metabolic tissues, including liver, iWAT, eWAT, and BAT, in control (Tfr1^{flox/flox}) and fat-specific Tfr1 KO (Tfr1-iAKO) mice fed with a regular chow diet. Tfr1^{flox/flox} and Tfr1^{flox/flox}-Adipoq-CreERT2 mice were treated with tamoxifen (0.1g Kg⁻¹ body weight for seven consecutive days) to inactivate Tfr1 in adipocytes in adult mice (Tfr1-iAKO). As shown in **Response Fig. 7a**, no significant difference was observed in BW between Tfr1^{flox/flox} and Tfr1-iAKO mice. In addition, the tissue

weights of the major metabolic tissues, including liver, iWAT, eWAT, and BAT, appear comparable between the two groups (**Response Fig. 7b**).

Response Fig. 7 Phenotypic studies in Tfr1^{flox/flox} and Tfr1-iAKO mice. **a, b** Body weight (**a**) and tissue weight (**b**) of Tfr1^{flox/flox} and Tfr1-iAKO mice fed with a regular chow diet. Data represent mean \pm SEM ($n = 4-6$ mice per group).

5. Given the current consensus regarding the energy expenditure analysis, the authors should show 1) EE without normalization by body weight, and 2) assess if a difference in EE depends on temperature (e.g., Speakman et al. Front. Physiol., 14 March 2013).

1) EE without normalization by body weight.

We appreciate the reviewer's comments. As discussed in the review article (Speakman et al. Front. Physiol., 14 March 2013), both body weight/mass and housing temperature are critical contributing factors for the assessment of whole-body energy expenditure (EE). According to the reviewer's suggestion, we have reanalyzed the EE data of AAV-Musclin and control mice without normalization to body weight. Similar to the body weight normalized EE (**Supplementary Fig. 4g**), while whole-body EE between the two groups remained indistinguishable before CL316,243 (CL) injection (**Response Fig. 8a, b**), CL induction of EE was significantly inhibited in AAV-Musclin mice compared to controls (**Response Fig. 8c**).

Response Fig. 8 Energy expenditure studies of AAV-GFP and AAV-Musclin mice. **a** Energy expenditure (EE, without normalization to body weight) of AAV-GFP and

AAV-Musclin mice determined before CL316,243 (CL) injection. Data represent mean \pm SEM ($n = 7-8$ mice per group). **b** Average EE in the daytime (Light) and nighttime (Dark) of AAV-GFP and AAV-Musclin mice. Data represent mean \pm SEM ($n = 7-8$ mice per group). **c** CL-induced increase in EE in AAV-GFP and AAV-Musclin mice, data are presented as changes over the values before CL treatment. * $p < 0.05$, two-tailed unpaired Student's t -test.

2) assess if a difference in EE depends on temperature (e.g., Speakman et al. *Front. Physiol.*, 14 March 2013).

We appreciate the reviewer's comments. During the CLAMS study of AAV-GFP and AAV-Musclin mice, we monitored the EE either before or after the CL316,243 injection. CL316,243 is a highly potent selective $\beta 3$ -adrenoceptor agonist, which partially mimics cold exposure to simulate adipocyte lipolysis and adipose tissue thermogenesis. As shown in **Supplementary Fig 4f** and **4g**, the EE remains comparable between AAV-GFP and AAV-Musclin mice before CL injection. However, the induction of EE in response to CL treatment was significantly lower in AAV-Musclin mice than that in AAV-GFP mice. These data suggest the difference in EE between AAV-GFP and AAV-Musclin mice is likely to be temperature dependent. To more specifically test this, we directly monitored the EE of control/Musclin-MKO mice and saline/Musclin Ab injected mice in different ambient temperatures. While the EE was similar between control and Musclin MKO groups when the mice were housed in thermoneutrality at 30°C for 1 week (**Response Fig. 9a, b**), Musclin-MKO mice exhibited a significantly higher EE in response to the temperature switch from 30°C to 24°C (**Response Fig. 9b**). Similar results were also obtained in mice treated with saline and Musclin Ab. Musclin Ab treatment increased EE in response to a temperature switch from 30°C to 24°C (new **Fig. 8i**), while displaying no obvious effect on EE when the mice were housed at 30°C (**Response Fig. 9c, d**). Taken together, these data suggest that the effects of Musclin gain-of-function and loss-of-function on EE are temperature dependent.

Response Fig. 9 Energy expenditure studies of Musclin^{flox/flox}/Musclin-MKO mice and saline/Musclin Ab treated mice. **a** Energy expenditure (EE, normalization to lean mass) of Musclin^{flox/flox} and Musclin-MKO mice monitored under thermoneutrality at 30°C. Data represent mean ± SEM ($n = 5-6$ mice per group). **b** Average EE of Musclin^{flox/flox} and Musclin-MKO and mice in the daytime (Light) and nighttime (Dark) during the 3 days of monitoring under thermoneutrality at 30°C as described in (a) and the 3 days of monitoring in response to temperature switch from 30°C to 24°C. Data represent mean ± SEM ($n = 5-6$ mice per group). ** $p < 0.01$, two-tailed unpaired Student's t -test. **c** EE (normalization to lean mass) of saline or Musclin Ab-treated mice monitored in thermoneutrality at 30°C. Data represent mean ± SEM ($n = 5-6$ mice per group). **d** Average EE of saline and Musclin Ab-treated mice in the daytime (Light) and nighttime (Dark) during the 3 days of monitoring under thermoneutrality at 30°C as described in (c) and the 3 days of monitoring in response to temperature switch from 30°C to 24°C. Data represent mean ± SEM ($n = 5-6$ mice per group). * $p < 0.05$ and ** $p < 0.01$, two-tailed unpaired Student's t -test.

Reviewer #2 (Remarks to the Author):

Jin et al. evaluated in depth the possible role of musclin in fat thermogenesis. The authors used several approaches to upregulate or downregulate the gene/protein, showing concordant changes and in the direction expected.

Major comments:

Being age and sex important contributors to muscle mass, no mention is made regarding this critical confounding factors in the recruiting of human samples. In mice

experiments, it is stated that male mice were used for all experiments unless otherwise indicated, but at the same time that "...those with the same genotype, gender and similar body weight were allocated into different groups". A thorough statement of age and sex would need to be specified in all experiments in vivo or regarding the samples obtained from human subjects.

We are grateful for the reviewer's encouraging and positive comments on our work. We totally agree with the reviewer that both age and sex are important contributors to muscle mass and function. According to the reviewer's suggestions, we have removed the sentence "...those with the same genotype, gender, and similar body weight were allocated into different groups" to avoid confusion. The information regarding both age and sex of all the mouse models used in this study has been specified in their corresponding figure legends. For human muscle samples, we have included the age and gender information in **Supplementary Table 5** (related to **Fig. 1b**) and **Supplementary Fig. 1b** (related to **Fig. 1e-g**) in the revision. For human fat samples, we have summarized the age and gender information in **Supplementary Fig. 5a** (related to **Supplementary Fig. 5b**) in the revised manuscript.

*The phenotype and potential confounding factors of human donors is very poor. As the subjects recruited underwent ligament repair or reconstruction, did the previous injury play a role in the current findings? In fact, findings shown in Figure 1b are quite heterogeneous.

We thank the reviewer for the insightful comments. For human muscle samples, donors with BMI < 18, deficiency in muscle development, skeletal muscle-related diseases, disability in normal physical activity, as well as abnormal body height were excluded in this study. In addition, subjects with other severe diseases, including malignant tumors and severe obesity-independent cardiovascular diseases, were also excluded. Moreover, at the beginning of the human muscle-related study, we were also aware of the potential confounding factors of previous acute injury on the gene and protein expression profiles in the muscle samples. As such, to minimize the confounding effect of previous injury in the current findings, the human muscle samples were only collected from patients who have fully recovered from acute injury and confirmation of the absence of acute inflammation prior to ligament repair or reconstruction treatment. Therefore, the heterogeneity of gene expression profiles shown in **Fig. 1b** might be due to the differences in gender (7 males and 3 females for BMI < 24 group; 6 males and 4 females for BMI ≥ 24) and age (18 to 55 years old for BMI < 24 group; 17 to 54 years old for BMI ≥ 24 group) among the subjects (**Supplementary Table 5**). Therefore, to further validate the correlation of *MUSCLIN* mRNA expression levels with BMI, we went on to examine skeletal muscle samples from 54 human donors (patient information including age and gender was described in **Supplementary Fig. 1b**). We found that the mRNA levels of *MUSCLIN* in human skeletal muscles were positively correlated with BMI (**Fig. 1e**). Accordingly, compared to normal human individuals (BMI < 24), mRNA expression of *MUSCLIN*

in skeletal muscle was robustly elevated in overweight ($24 \leq \text{BMI} < 28$) and obese ($\text{BMI} \geq 28$) human subjects (**Fig. 1f**). Further western blotting analysis confirmed that the protein levels of MUSCLIN in skeletal muscle were significantly increased in obese human subjects (**Fig. 1g**). We have added these human donors related information in the “Methods” section and incorporated these new data in the revised manuscript.

*Given the alternative name of musclin (osteocrin), it is possible that the bone could also be playing a role in the observed effects. There is mounting literature about the influences exerted by different bone-derived molecules in systemic metabolism. It seems that virtually no data has been reported about this possible influence in this paper.

We thank the reviewer for raising this important concern. In the previous study, Musclin (Osteocrin) was discovered to be expressed and secreted from bone, and it's restricted to be detected in long bones and calvaria of neonate and adult⁵. Moreover, Musclin expression in the long bones and calvaria was age-dependent, with the highest expression in newborn or young mice and much lower expression in elder mice (8 months old)⁵. However, another study showed that Musclin expression in skeletal muscle was much higher than that in the bone of adult mice (10-12 weeks old)⁶. Indeed, there is mounting evidence suggesting the influences exerted by different bone-derived molecules in systemic metabolism⁷. It is possible that Musclin from bone may also play a role in the regulation of systemic energy metabolism in adult mice. To test this possibility, according to the reviewer's suggestion, we have also examined Musclin expression levels in skeletal muscle and bone from WT C57BL/6J adult mice. As shown in **Response Fig. 10**, consistent with the study by Hitoshi Nishizawa et al.⁶, the tibia, a long bone in the hindlimb, displayed a markedly lower expression of Musclin gene than that in skeletal muscle in adult mice.

As depicted in **Fig. 7**, Musclin-MKO improved systemic energy homeostasis in mice. While we have shown that both mRNA and protein expression of Musclin was substantially decreased in skeletal muscle of Musclin-MKO mice (new **Supplementary Fig. 7b** and **Fig. 7a**), leading to a significant decrease of plasma Musclin levels (**Fig. 7b**), whether Musclin expression is also altered in bone, which in turn, affect systemic energy metabolism, remains unclear. Therefore, we also examined the Musclin mRNA levels in both skeletal muscle and long bone from Musclin-MKO and control mice. The results show that muscle-specific knockout of Musclin does not influence its expression in bone (**Supplementary Fig. 7b**). These data, together with the results shown in **Fig. 7**, indicate that skeletal muscle-derived Musclin plays a pivotal role in the regulation of adipose tissue thermogenesis and systemic energy homeostasis. Nonetheless, we also noticed that plasma concentrations of Musclin were partially decreased in Musclin-MKO mice, suggesting that other tissues may also contribute to the circulating levels of Musclin. The roles of Musclin secreted from these tissues in the regulation of tissue homeostasis and energy

metabolism remain to be explored in future studies. We have cited these references and updated the “Discussion” section in the revised manuscript.

Response Fig. 10 Musclin expression in skeletal muscle and bone. qPCR analysis of Musclin expression in quadriceps muscle and tibia bone from WT mice. Data represent mean \pm SEM ($n = 4$ mice per group). *** $p < 0.001$, two-tailed unpaired Student's t -test.

Minor suggestions:

*Irisin is known to promote thermogenesis but their circulating levels are increased in sedentary subjects with obesity. Ideally, the information about physical activity would need to be provided.

We thank the reviewer for this good suggestion. In this study, the human muscle samples were gracilis and semitendinosus muscles collected from patients who underwent ligament repair or reconstruction treatment due to ligamentous injury of the knee joint or patella injury at the First People's Hospital of Lianyungang City during the 2018-2021. Donors with BMI < 18 , deficiency in muscle development, skeletal muscle-related diseases and disability in normal physical activity, and abnormal body height were excluded from this study. In addition, subjects with other severe diseases, including malignant tumors and severe obesity-independent cardiovascular diseases, were also excluded. Moreover, to minimize the confounding effect of previous injury in the current findings, the human muscle samples were only collected from patients who have fully recovered from acute injury and confirmation of the absence of acute inflammation prior to ligament repair or reconstruction treatments. BMI is the only criterion to divide the human muscle samples into different groups, as such, we assume that the physical activity should be similar between BMI < 24 and BMI ≥ 24 groups.

To further confirm the upregulation of MUSCLIN in skeletal muscles in obese humans, we all examined the expression of Musclin in skeletal muscles from obese and diabetic mouse models. Consistent with the observations in humans, *Musclin* mRNA expression was markedly induced in skeletal muscles from high-fat diet (HFD)-induced and genetically obese (*ob/ob*) mice and diabetic (*db/db*) mice (**Fig. 1j**). All the mice involved in this study were housed in mouse cages with free physical activity, suggesting that physical activity may not contribute to the upregulation of

Musclin expression in skeletal muscles in obese humans and mice.

Moreover, we have also assessed the effects of Musclin overexpression or deficiency on physical activity in mice. As shown in **Supplementary Fig. 4j, k** and **Supplementary 7j**, Musclin gain-of-function or loss-of-function in AAV-Musclin mice or Musclin-MKO mice, respectively, exhibited no obvious effect on the physical activity in mice compared to their respective controls. We have added these information and new data in the revised manuscript.

*Page 7, line 181: It is mentioned that muscle-specific transgenic expression of Musclin had a modes effect on the development, metabolism, and exercise performance of skeletal muscle per se, but no information virtually is provided to sustain this statement.

We thank the reviewer for raising this important concern. we have determined the expression of mentalism-associated genes in adult MCK-Musclin and control mice under normal chow diet feeding and housing conditions. As shown in **Response Fig. 11**, no significant difference was detected in the mRNA expression of genes related to lipid metabolism, glucose metabolism, and mitochondrial function between the two groups. In addition, we have also examined the exercise performance and histology of skeletal muscles in control and MCK-Musclin mice. As shown in **Supplementary Fig. 2b**, the treadmill running distance and time are indistinguishable between control and MCK-Musclin mice. The tissue weights and appearance of skeletal muscles are also similar between the two groups (**Supplementary Fig. 2c, d**). Further H&E staining analysis of muscle tissue sections revealed indistinguishable muscle histology between control and MCK-Musclin mice (**Supplementary Fig. 2e**). Collectively, these data demonstrated that muscle-specific transgenic expression of Musclin has a modest effect on the development, metabolism, and exercise performance of skeletal muscle *per se*. We have incorporated these new data in this revision.

Response Fig. 11 Relative gene expression levels in quadriceps from chow diet-fed control and MCK-Musclin mice. Data represent mean \pm SEM ($n = 7-8$ mice per group). No significant difference was obtained using the two-tailed unpaired Student's *t*-test.

*Page 11, line 298: harvested scWAT. Did the authors harvest scWAT or simply

obtained the adipose tissue samples.

The reviewer has raised a good question. As shown in **Response Fig. 12**, the human subcutaneous adipose tissue (scWAT) samples included in this study were collected from the fat layer that is located underneath the Scarpa's Fascia and outside of the abdominal cavity (the blue arrow, the deep abdominal subcutaneous adipose tissue (SAT) as previously described^{8,9}) of patients who underwent surgical repair because of oblique inguinal hernia, incisional hernia, or appendicitis at the Second Affiliated Hospital of Zhejiang University School of Medicine.

Response Fig. 12 The location of the scWAT from human subjects harvested in this study. Figure adapted from Luong et al.⁹, *Biology* 2019 Apr 11;8(2):23. SAT, subcutaneous adipose tissue.

*Page 14, line 401. Data not shown. Why?

We thank the reviewer for the insightful comment. In this study, we observed that Musclin neutralizing antibody (Musclin Ab) treatment robustly lowered body weight gain compared to saline in chow diet-fed WT mice (**Figure 8c**). We were wondering whether food intake or physical activity was altered by Musclin Ab treatment, and thereby contributing to the difference in body weight gain. As such, we have measured the food intake and physical activity using the metabolic cages in mice treated with saline and Musclin Ab, and found that both food intake and physical activity were similar between the two groups. These results suggest that Musclin Ab treatment lowers body weight gain is not due to its potential side effects on food intake and physical activity. We have included these new data in **Supplementary Fig. 7k, I** in the revised manuscript.

Reviewer #3 (Remarks to the Author):

Lu Jin et al. investigated the impact of the myokine musclin on beige fat thermogenesis. They found increased Musclin serum levels in individuals with a BMI>24 and in mouse models of diabetes and obesity. Using transgenic muscle specific (but also systemic AAV-based) Musclin overexpression as well as muscle specific Musclin knock-out mice, the authors demonstrate that Musclin inhibits the development of beige fat during cold exposure and high fat diet administration, triggering exacerbation of obesity under the latter condition. Mechanistically, the authors suggest endocrine crosstalk between skeletal muscle and inguinal white adipose tissue, which they found to be rather selectively bound by Musclin. Using an unbiased proteomics-based approach, the authors identified the transferrin receptor as binding target of Musclin at inguinal WAT. Intracellularly Musclin inhibits the cAMP formation and PKA activation.

This is a very interesting manuscript addressing a timely topic using elegant genetic and translational mouse models with results that could be of high interest for treating obesity or metabolic syndrome. A previously unknown function of Musclin is described. This being said, the authors almost completely neglect previous knowledge describing Musclin as inhibitor of the natriuretic peptide degrading receptor NPRC, and therefore as an amplifier of natriuretic peptide signaling. The authors need to address these points. In addition, some parts of the mechanism remain unclear. More specifically, the authors need to address the following points:

Major:

1) Musclin competes with ANP, BNP and CNP for binding to NPRC. All natriuretic peptide receptors are also expressed in adipose tissue and have a known role in lipolysis and perhaps fat browning (e.g. Bordicchia et al., JCI, 2012). How are the serum levels of ANP (and CNP?) in Musclin overexpressing or knock-out mice? How are cGMP levels changed in iWAT or BAT after manipulation of Musclin during high fat diet or cooling? Are cAMP level changes by Musclin the result of changes of cGMP levels, which could happen for example via PDE3.

We appreciate the reviewer's enthusiasm for our study and the insightful comments and suggestions. As pointed out by the reviewer, Musclin has been shown to function as a ligand to NPRC, a natriuretic peptide (NP) clearance receptor, competing with NPs, including ANP and CNP, to amplify the natriuretic peptide signaling^{1,10}. It is possible that Musclin may elicit its effects on adipocyte thermogenesis and metabolism through regulating the circulation levels of ANP and CNP. To test this, as per the reviewer's suggestion, we have performed ELISA assays of the circulating levels of ANP and CNP in MCK-Musclin and Musclin-MKO mice and their respective control mice. As shown in **Response Fig. 13**, concentrations of both ANP and CNP in the circulation remained largely unaltered in MCK-Musclin and Musclin-MKO mice compared to their respective controls, suggesting that ANP and CNP may not be involved in the regulation of adipose tissue thermogenesis and systemic energy metabolism by Musclin.

Response Fig. 13 Plasma ANP and CNP levels MCK-Musclin and Musclin-MKO mice and their respective controls. **a** Plasma levels of ANP in chow diet-fed WT and MCK-Musclin mice (left) and in Musclin^{flox/flox} and Musclin-MKO (right). Data represent mean \pm SEM ($n = 5-6$ mice per group). **b** Plasma levels of CNP in chow diet-fed WT and MCK-Musclin mice (left) and in Musclin^{flox/flox} and Musclin-MKO (right). Data represent mean \pm SEM ($n = 6-8$ mice per group). No significant difference was obtained using the two-tailed unpaired Student's t -test.

The cyclic GMP (cGMP), generated by the guanylyl cyclase domain of NPRA and NPRB, is a key mediator in the natriuretic peptide-associated biological responses¹¹. According to the reviewer's suggestions, we measured the cGMP levels in iWAT and BAT from control and MCK-Musclin mice in response to either chow diet feeding and cold exposure, or HFD feeding conditions, as well as in iWAT from control and Musclin-MKO mice upon HFD feeding. As shown in **Response Fig. 14**, cGMP levels in iWAT or BAT remained largely unchanged after manipulation of Musclin during HFD feeding or cooling, indicating that cGMP/PDE3 pathway might not be involved in the regulation of cAMP levels by Musclin in adipocytes.

Response Fig. 14 cGMP levels in iWAT or BAT in mice with manipulation of Musclin during high-fat diet feeding or cooling. **a** cGMP content in iWAT (left) and BAT (right) from WT and MCK-Musclin mice after cold treatment at 8°C for 1 week. Data represent mean \pm SEM ($n = 5-7$ mice per group). **b** cGMP content in iWAT (left) and BAT (right) from WT and MCK-Musclin mice fed with HFD for 8 months. Data represent mean \pm SEM ($n = 5-6$ mice per group). **c** cGMP content in iWAT from Musclin^{flox/flox} and Musclin-MKO mice fed with HFD for 5 months. Data represent mean \pm SEM ($n = 6$ mice per group). No significant difference was obtained using the two-tailed unpaired Student's t -test.

2) Are the effects of Musclin on adipocytes dependent on the expression of Npr1 or Npr2?

We thank the reviewer for this insightful comment. Both Npr1 and Npr2 are well known to play important roles in mediating natriuretic peptide signaling. Previous studies have shown that both Npr1 and Npr2 are involved in the regulation of adipocyte differentiation¹². To avoid this confounding effect in our assessment of the involvement of these two receptors in the effects of Musclin on adipocytes, we performed siRNA-mediated knockdown of Npr1 and Npr2 in differentiated C3H10T1/2-derived adipocytes, followed by Musclin treatment and qPCR analyses of genes involved in glucose and lipid metabolism and thermogenesis. As shown in **Response Fig. 15a**, *Npr1* mRNA expression was markedly decreased in differentiated adipocytes transfected with both siRNAs targeting *Npr1* (si*Npr1*-1 and si*Npr1*-2) compared to the control group. As expected, siRNA knockdown of Npr1 exhibited mild effects on adipocyte differentiation as revealed by the QPCR analysis of marker genes of adipocyte differentiation, including *Cebp α* , *Cebp β* , *Ppar γ 1*, and *Ppar γ 2*. Then the control (CTR) or Npr1 knockdown (si*Npr1*-2) mature adipocytes were subjected to Musclin treatment. Interestingly, Musclin treatment elicited similar inhibitory effects on the expression of genes involved in lipid (*Atgl*, *Hsl*, *Mgll*, *Lpl*, *Acaa1b*) and glucose (*Aldoa*, *Pdhhb*) metabolism and thermogenesis (*Prdm16*, *Ucp1*, *Cox8b*, *Cidea*) in the *Npr1* knockdown adipocytes as in the CTR adipocytes (**Response Fig. 15b**).

Similarly, knockdown of the *Npr2* gene in differentiated C3H10T1/2-derived adipocytes by two siRNAs exhibited mild effects on adipocyte differentiation as revealed by the expression of adipocyte differentiation marker genes (**Response Fig. 15c**). Notably, the inhibitory effects of Musclin on the expression of genes involved in glucose and lipid metabolism and thermogenesis remained largely unaffected by *Npr2* knockdown (**Response Fig. 15d**). Expression of genes related to lipid (*Hsl*, *Mgll*, *Lpl*) and glucose (*Aldoa*) metabolism and thermogenesis (*Prdm16*, *Ucp1*, *Cox8b*, *Cidea*) was significantly decreased by Musclin treatment. Nonetheless, consistent with previous studies showing that Npr1 and Npr2 are involved in the regulation of lipolysis and metabolism of adipocytes, we also observed that mRNA levels of several genes related to metabolism and thermogenesis are decreased in Npr1 or Npr2 knockdown adipocytes (si*Npr1*-2+vehicle vs CTR+vehicle in **Response Fig. 15b**, and si*Npr2*-1+vehicle vs CTR+vehicle in **Response Fig. 15d**). Together, these results provide evidence suggesting the Npr1- and Npr2-independent effects of Musclin on adipocyte thermogenic metabolism.

Response Fig. 15 Roles of *Npr1* and *Npr2* in the effects of Musclin on adipocyte metabolism and thermogenesis. **a** qPCR analysis of *Npr1* and adipocyte differentiation marker genes in differentiated C3H10T1/2-derived adipocytes transfected with control siRNA (CTR) or siRNAs targeting *Npr1* (si*Npr1*-1 or si*Npr1*-2). Data represent mean \pm SD ($n = 3$ biological replicates). *** $p < 0.001$, one-way ANOVA with Sidak's multiple comparisons. **b** qPCR analysis of genes involved in lipid and glucose metabolism and thermogenesis in C3H10T1/2-derived adipocytes with indicated treatments. Data represent mean \pm SD ($n = 3$ technical replicates), * $p < 0.05$, ** $p < 0.01$, *** $p < 0.001$, one-way ANOVA with Sidak's multiple comparisons test. **c** qPCR analysis of *Npr2* and adipocyte differentiation marker genes in C3H10T1/2-derived adipocytes transfected with control siRNA (CTR) or siRNAs targeting *Npr2* (si*Npr2*-1 or si*Npr2*-2). Data represent mean \pm SD ($n = 3$ biological replicates). *** $p < 0.001$, one-way ANOVA with Sidak's multiple comparisons. **d** qPCR analysis of genes involved in lipid and glucose metabolism and thermogenesis in C3H10T1/2-derived adipocytes with indicated treatments. Data represent mean \pm SD ($n = 3$ technical replicates), * $p < 0.05$, ** $p < 0.01$, *** $p < 0.001$, one-way ANOVA with Sidak's multiple comparisons test. C3H10T1/2 adipocytes were differentiated for 6 days before transient transfection of siRNAs. About 60-65 h after transfection, cells were treated with vehicle or Musclin-Fc for an additional 6 h, and then harvested for total RNA extraction and qPCR analysis. The data are representative of at least two independent experiments.

3) How does Musclin selectively target iWAT, but not eWAT or other tissues? Is Tfr1 expression different between the different tissues (for example higher in iWAT than eWAT?)

We thank the reviewer for raising such an important question. Since adipose tissue contains not only adipocytes but also other cell types in the stromal vesicular fraction (SVF). To examine adipocytes from which fat depots are targeted by Musclin, we

isolated mature adipocytes from eWAT, iWAT, and iBAT, and performed western blotting analysis of the protein levels of Tfr1. Consistent with our observations that Musclin potently inhibited thermogenic metabolism in iWAT during chronic cold exposure (**Fig. 2m** and **Fig. 3b, c, e, f**), whereas having modest effects on the energy metabolism and thermogenesis in iBAT and eWAT (**Supplementary Fig. 3a-e**), Tfr1 protein levels in adipocytes from iWAT were much higher than those in adipocytes from iBAT and eWAT from WT mice after cold exposure at 6°C for 1 week (**Response Fig. 16**). These results suggest the protein expression levels of Tfr1 receptor in adipocytes during chronic cold exposure might be responsible for the selectivity of Musclin on iWAT to exert its effects on adipose tissue thermogenesis and systemic energy homeostasis.

Response Fig. 16 Protein levels of Tfr1 in mature adipocytes from eWAT, iWAT, and iBAT in mice. Adipose tissues were harvested from WT mice after cold treatment at 6°C for 1 week. Mature adipocytes were isolated from the indicated adipose tissues and subjected to whole-cell protein lysate preparation and western blotting analyses. Total protein lysates of iWAT from Tfr1^{flox/flox} and Tfr1^{flox/flox} Adipoq-CreERT2 (Tfr1-iAKO) mice were used as controls for the validation of the Tfr1 antibody.

4) How are serum Glucose levels changed by Musclin? Is this mediated by its effect on iWAT or is this through its effect on skeletal muscle?

We thank the reviewer for raising these insightful questions. To identify the targets of Musclin signaling *in vivo*, we have generated a fusion protein of SEAP and Musclin (SEAP-Musclin) and performed binding assays on frozen sections of metabolic tissues. The tissue binding sites were determined by histochemical staining for alkaline phosphatase activity. Interestingly, the strongest binding activities on tissue sections were observed on the tissue section of inguinal white adipose tissue (iWAT), accompanied by much weaker binding signals on eWAT, interscapular BAT, and liver sections, while the binding activities on muscle and pancreas tissue sections were almost undetectable (**Fig. 2e**), suggesting that adipose tissues, especially iWAT, might serve as the primary targets of Musclin signaling to exert its effect on adipose tissue thermogenesis.

In this study, we observed that blood glucose levels were significantly higher in MCK-Musclin mice compared to control mice in response to cold exposure at 8°C for

1 week (**Fig. 2o**). Both skeletal muscle and adipose tissues are known to play important roles in the regulation of systemic glucose homeostasis. Consistent with the results in SEAP binding assays, the expression of genes involved in glucose uptake and metabolism was significantly decreased in iWAT from MCK-Musclin mice compared to those from control mice (**Response Fig. 17**), whereas the expression of these genes remained largely unaltered in skeletal muscle (**Supplementary Fig. 3f**). In addition, the expression levels of genes related to lipid metabolism and mitochondrial function were also similar in skeletal muscle from control and MCK-Musclin (**Response Fig. 11**). In addition, we have also examined the exercise performance and histology of skeletal muscles in control and MCK-Musclin mice, and showed that MCK-Musclin has a modest effect on the development and exercise performance of skeletal muscle (**Supplementary Fig. 2**). These data suggest that regulation of blood glucose levels by Musclin is likely mediated by its direct effects on iWAT instead of skeletal muscle.

Response Fig. 17 Expression of genes related to glucose uptake and metabolism in iWAT from control and MCK-Musclin mice housed at 8°C for 1 week. Data represent mean \pm SEM ($n = 5$ mice per group). * $p < 0.05$, ** $p < 0.01$, *** $p < 0.001$, two-tailed unpaired Student's t -test.

5) How are adipocyte cAMP levels controlled by Tfr1?

We thank the reviewer for raising this great question. Starting from the *in vivo* observations that elevation of circulating Musclin levels inhibited cold- and CL316,243 treatment-induced beige fat thermogenesis and impaired systemic energy homeostasis, we went on to demonstrate that Musclin suppressed the transcriptional program related to thermogenic metabolism in beige adipocytes in a cell-autonomous manner (**Fig. 3g**). Further seahorse studies revealed that both basal and forskolin (FSK, an adenylate cyclase agonist)-induced cell respiration rates were robustly attenuated by Musclin (**Fig. 3h**). Consistent with the *in vivo* phenotype and transcriptional regulation, Musclin markedly decreased the cellular levels of cAMP and the phosphorylation levels of PKA substrates in beige adipocytes (**Fig. 6a, b**). However, this inhibitory effect was blocked by pretreatment with 8-Br-cAMP, a cell-permeable cAMP analog (**Fig. 6c**). These data suggest that Musclin inhibits thermogenic metabolism in beige adipocytes likely through modulating adenylate

cyclase activity and its downstream cAMP/PKA signaling.

In this study, through BioID and mass spectrometry, we identified Tfr1 as the new receptor for Musclin in adipocytes. We demonstrated that both basal and isoproterenol (ISO, a nonselective β -adrenoceptor agonist)-induced elevation of cellular cAMP levels was dramatically attenuated by Tfr1 inactivation in beige adipocytes, leading to marked decreases in the phosphorylation levels of PKA substrates (**Fig. 6i, j**). Moreover, ISO-induced elevation of cAMP and its downstream PKA signaling was robustly decreased by either Musclin treatment or Tfr1 knockdown alone, and no further decrease was observed by Musclin treatment in Tfr1 knockdown beige adipocytes (**New Fig. 6k, l**). These results indicate that Tfr1 may regulate intracellular cAMP levels by modulating adenylate cyclase activity. To further test this, differentiated beige adipocytes were treated with holo-transferrin in the presence or absence of SQ22536, an adenylate cyclase inhibitor, followed by intracellular cAMP measurement. As shown in **Response Fig. 18**, holo-transferrin significantly elevated intracellular cAMP levels, which was largely abolished by SQ22536 pretreatment. Taken together, our data suggest that Tfr1 controls adipocyte cAMP levels through modulating adenylate cyclase activity. However, the detailed molecular mechanisms underpinning how Tfr1 regulates the activity of adenylate cyclase in adipocytes remained to be explored in future studies.

Response Fig. 18 Intracellular cAMP levels in differentiated beige adipocytes treated with holo-transferrin in the presence or absence of SQ22536. Data represent mean \pm SD ($n = 3$ biological replicates). ** $p < 0.01$, *** $p < 0.001$, one-way ANOVA with Sidak's multiple comparisons test. SQ, SQ22536.

6) The authors suggest that Musclin triggered effects via Tfr1 are iron independent, have they checked some iron homeostasis (e.g. ferritin levels)?

We thank the reviewer for raising this critical question. Tfr1, as a member of the transferrin receptor family, is known to play an important role in cellular iron homeostasis through endocytosis of transferrin-bound iron^{13,14}. Intriguingly, accumulating evidence indicates that iron-independent mechanisms might also be involved in the regulation of biological functions by Tfr1¹⁵. Accordingly, Tfr1 was

uncovered to play important roles in the regulation of white adipose tissue homeostasis and brown/beige fat development and thermogenesis via both iron-dependent and -independent mechanisms¹⁶⁻¹⁸. To test whether iron-dependent or -independent mechanisms are involved in the Tfr1-mediated regulation of adipocyte thermogenic metabolism by Musclin, we have measured non-heme iron content in both iWAT and BAT from control and MCK-Musclin mice, and observed no significant difference between the two groups (**Response Fig. 19**), suggesting that iron homeostasis may not be involved in the regulation of adipose tissue thermogenic metabolism by Musclin. However, the precise roles of iron-dependent or -independent mechanisms in the regulation of adipose tissue thermogenesis by Musclin-Tfr1 signaling remain to be further investigated. We have added more discussion on this aspect in the revised manuscript (Page 19, lines 550-562).

Response Fig. 19 Non-heme iron content in BAT and iWAT from control and MCK-Musclin mice. Data represent mean \pm SEM ($n = 5$ mice per group).

7) I am still struggling with the seemingly contradictory effect of systemic musclin KO mice (Subbotina E et al., PNAS, 2015) showing less mitochondria in skeletal muscle and the current study implying more mitochondria in adipose tissue. Can the authors elaborate on this?

We appreciate the reviewer for raising this insightful concern. We also noticed the seemingly contradictory effects of Musclin on mitochondrial content and activity in skeletal muscle using whole-body KO mice in a previous study¹⁹ and in adipose tissue using skeletal muscle-specific KO and transgenic mice in this study. The following three reasons may contribute to this discrepancy: (1) The potential effects of Musclin on skeletal muscle development and bone growth caused by systemic inactivation of Musclin, as previously described using whole-body KO mice^{1,5}. Different from the previous study using whole-body KO mice, to rule out these potential confounding factors in the evaluation of Musclin produced and secreted from mature skeletal muscle on systemic energy homeostasis, here we generated mice carrying muscle-specific transgenic expression or KO of Musclin using MCK promoter and MLC-Cre, respectively. Both MCK²⁰ and MLC²¹ promoters have been shown to be mainly expressed in mature skeletal myocytes and have been widely used to generate mature

myocyte-specific transgenic or KO mouse models; (2) Different running protocols and testing conditions were used in two studies. Here, we evaluated all the muscle-related parameters in sedentary mice, while mice were subjected to 5 consecutive days of exercise training prior to assessment in their research. In fact, Subbotina E et al.¹⁹ found that differences in endurance running capacity and markers of mitochondrial biogenesis were much less evident in sedentary WT and Musclin-KO mice than those in exercise-trained WT and Musclin-KO mice, which is consistent with their hypothesis that running endurance affected by Musclin is mediated by its effect on ANP/Pgc1 α pathway-dependent regulation of exercise training-related mitochondrial biogenesis. In this study, using SEAP binding assay on tissue sections from sedentary WT mice, we demonstrated that Musclin mainly binds to adipose tissues, especially subcutaneous iWAT, rather than other metabolic tissues, including skeletal muscle and liver, and showed that Musclin selectively acts on iWAT to suppress thermogenic gene expression programs, decreases mitochondrial content and activity, and thereby leading to impairment of beige fat thermogenesis and systemic energy balance. In accordance, we have examined the both gene expression and protein levels of mitochondrial OXPHOS complexes in skeletal muscle from control and Musclin-MKO mice, and observed no significant difference between the two groups (**Response Fig. 20**). Nonetheless, it is possible that following exercise training, Musclin may also act on skeletal muscle in an autocrine fashion to exert its effect on muscle mitochondrial biogenesis. In this case, both skeletal muscle and adipose tissue may contribute to the effects of Musclin on systemic energy homeostasis. Further studies are required to test these possibilities. (3) Different receptors and associated downstream signal pathways might be another reason for the tissue-specific effects of Musclin on mitochondrial biogenesis in skeletal muscle and adipose tissue. In the previous study, Musclin was shown to bind to Npr3, a clearance receptor for natriuretic peptides, leading to the enhancement of activity-related ANP/cGMP/Pgc1 α -dependent skeletal muscle mitochondrial biogenesis and endurance running activity. However, in this study, we identified Tfr1 as a new receptor for Musclin in adipocytes, and revealed that Tfr1-mediated cAMP signaling plays a pivotal role in the regulation of thermogenic metabolism in subcutaneous white adipose tissue. Moreover, during the revision, we have performed additional studies to provide more evidence suggesting that Npr1, Npr2, and Npr3 are likely not required for the effects of Musclin on adipose tissue thermogenesis. We have updated the “Discussion” section in the revised manuscript (Pages 18-19, lines 515-536).

Response Fig. 20 Gene expression and protein levels of mitochondrial OXPHOS complexes in skeletal muscles. **a** qPCR analysis of mitochondrial complexes-related genes quadriceps muscles from HFD-fed control and Musclin-MKO mice. Data represent mean \pm SEM ($n = 9-11$ mice per group). **b** Total protein lysates of quadriceps muscles from HFD-fed control and Musclin-MKO mice were subjected to western blotting analysis of mitochondrial OXPHOS complex and Hsp90 (as internal control). CI-CV, mitochondrial OXPHOS complex I to V.

Minor:

The authors do not mention at all that Musclin was also discovered as exercise dependent myokine with cardioprotective features and they neglect some of the relevant recent literature (Kattih B et al., JAHA, 2022; Szaroszyk et al., Nat comm, 2022).

We thank the reviewer for raising this important concern. We are sorry for not mentioning these two new studies on the cardioprotective roles of Musclin in our original manuscript. Interestingly, Musclin was shown to attenuate the pathogenesis and progression of heart failure during pathological overload²², and its low circulating level is associated with adverse prognosis of patients undergoing transcatheter aortic valve implantation (TAVI)²³. These important new discoveries further highlight the tissue-specific roles of Musclin under different physiological and disease conditions. We have cited the two papers and discussed these new studies demonstrating that Musclin functions as exercise-dependent myokine with cardioprotective features in the “Discussion” section of the revised manuscript (Page 17, lines 496-502).

Reviewer #4 (Remarks to the Author):

In their manuscript entitled “The muscle-enriched myokine Musclin impairs beige fat thermogenesis and systemic energy homeostasis via Tfr1/PKA signaling” the authors report that mouse and human obesity is associated with elevated muscle expression and circulating levels of Musclin. The muscle expression of Musclin was also detected to be environment temperature sensitive, and beige fat was identified as primary site of Musclin action. Overexpression of Musclin was also detected to inhibit beige fat thermogenesis mimicking the obesity and metabolic disorders,

whereas Musclin ablation promoted beige fat thermogenesis and improved metabolic homeostasis. The authors also tried to find the molecular mechanisms behind this and find that Musclin bind to transferrin receptor 1 (Tfr1) and antagonises Tfr1-mediated cAMP/PKA-dependent thermogenic induction in beige adipocytes. Overall, this is a very large and extensive work utilising multiple different sample types and methods. However, several issues need to be solved before the manuscript is ready for publication.

Major criticism:

Figure 1. b-c: The authors should comply with standard guidelines for RNA-Seq data reporting and should include the actual data as a supplementary table and also should ensure that the actual raw data is submitted by the time of publication to a public data repository (it is difficult to judge this, as no access was made available for the reviewers)

We are deeply grateful to the reviewer for his/her enthusiasm and positive comments on our work. According to the reviewer's suggestion and guidelines for RNA-Seq data reporting, we have included the actual data of the RNA-seq data in **Fig. 1b-c** in **Supplementary table 2**.

All the actual raw RNA-Seq datasets of mouse tissues are available for data presented in **Fig. 1c**, **3b-d**, and **Supplementary Fig. 3a-d**, and have been deposited in the public data repository Mendeley Data (<http://dx.doi.org/10.17632/fx72gjjwvc9.1>). Raw RNA-Seq datasets of human muscle samples are available for the data presented in **Fig. 1b**, and have been deposited in the National Omics Data Encyclopedia (NODE) with accession number OEP003781 (<https://www.biosino.org/node/project/detail/OEP003781>). We have added this information to the "Data and materials availability" section in the revised manuscript and will ensure that all the RNA-Seq raw data will be publicly available to the readers by the time of publication.

Figure 1. g: there is a horizontal white line present in the Musclin plot. Original uncropped images of this WB as well as other WBs should be included as a supplementary images.

We appreciate the reviewer's insightful comment. We have double-checked the original uncropped images for all the immunoblots shown in this manuscript and included them in **Supplementary table 3** (for immunoblots shown in main and Supplementary Figures) and **Supplementary table 7** (for immunoblots shown in Response Figures).

Figure 2. d: could perhaps use some more replicates than one as $n \geq 6$ animals were most likely used in the actual analyses.

Thank the reviewer for his/her suggestion. We have included more skeletal muscle samples from control ($n = 5$) and MCK-Musclin ($n = 7$) mice for the western blotting analyses of Musclin and GAPDH (internal control) protein levels. Moreover, we performed a quantification analysis of the immunoblots and showed that skeletal muscle Musclin protein levels were significantly elevated in MCK-Musclin mice compared to controls. These new data have been incorporated into **Fig. 2f** of the revised manuscript.

Figure 3. a-d: the whole RNA-Seq (quantitative) data should be presented in a supplementary table. Additionally, I have serious concerns of the used cutoff values ($p < 0.05$, $\log_2(\text{fold change}) > 0.3$). These are significantly lower than what normally are used and it is impossible to say if the changes reported actually are significant or biologically relevant.

Per the reviewer's suggestion, the whole RNA-Seq (quantitative) data shown in the original **Fig. 3a-d** have been included in **Supplementary table 4** in the revised manuscript.

We appreciate the reviewer for raising critical concerns about the cutoff values using our original analysis. Low cutoff may include some genes that are only modestly regulated and may confound the following GO pathway analysis. To address this issue, we have re-analyzed these RNA-Seq data using cutoff values of $p < 0.05$ and $\log_2(\text{fold change}) > 0.5$. As expected, the new analysis cutoff significantly decreased the number of genes in each data set for GO analysis. Similar to the results obtained using the cutoff in the previous analysis, the significantly downregulated genes (Set I, 350 genes) in iWAT from MCK-Musclin mice are closely associated with energy generation, fatty acid and glucose metabolism, cellular respiration, and body temperature homeostasis (New **Fig. 3b, c**). The significantly upregulated genes (Set II, 561 genes) are mainly enriched in the immune biological process (New **Fig. 3b, c**). Similarly, we have also re-performed the overlapping analysis with higher cutoff values ($p < 0.05$ and $\log_2(\text{fold change}) > 0.5$) on the differentially expressed genes (DEGs) in iWAT from comparisons between control and MCK-Musclin mice, and DEGs in iWAT from comparisons between control and cold-acclimated mice based on RNA-Seq datasets. As expected, a smaller set of genes (142 out of 1179) upregulated in iWAT of WT mice in response to cold acclimation were reversed by Musclin transgenic expression (New **Fig. 3d**, middle panel), the enriched GO biological processes of reversed genes are similar to the terms obtained in the original manuscript (original **Fig. 3d**), including fatty acid oxidation, generation of energy, purine ribonucleotide metabolism, glucose metabolism, tricarboxylic acid cycle, and cellular respiration. On the other hand, a set of genes (99 out of 571) downregulated by cold acclimation in iWAT of WT mice were reversed by Musclin transgenic expression (New **Fig. 3d**, right panel), mainly enriched in GO biological processes of regulation of cytokine production, lymphocyte differentiation, regulation of immune response, mononuclear cell differentiation, and regulation of T cell activation. We

have incorporated these new results in **Fig. 3b-d**, and updated the text in the revision.

Figure 6. d: The proximity labeling is highly dependent on the localization of the BirA*-tagged protein and choosing an appropriate control requires some thought. The authors use the BirA* tag alone as a control for secreted Musclin-BirA*. To my knowledge the BirA*-tag alone is not well secreted to the cell culture media and therefore is not a great control. Therefore the experiment should be redone using a secreted BirA* control.

Additionally, the mass spectrometry data (after the database searches) should be added as a supplementary table.

We totally agree with the reviewer's comments. According to the reviewer's suggestion, we have constructed a vector that expressed BirA* with a flag tag (for the western blotting analysis of BirA*) in its C-terminus and the same signal peptide (Sp-BirA*-flag) as Musclin-BirA* in its N-terminus. As shown in **Response Fig. 21a**, Sp-BirA*-flag is secreted and readily detectable in the conditioned medium. We then applied the conditioned medium containing either Sp-BirA*-flag or Musclin-BirA* to treat the fully differentiated C3H10T1/2-derived adipocytes, followed by streptavidin pull-down of the cell membrane-associated proteins and cytoplasmic proteins. Through western blotting analyses, similar to the results shown in the original manuscript, Tfr1 was confirmed to be present in the pull-down sample of membrane-associated protein fraction treated with Musclin-BirA*, but barely detectable in the sample treated with Sp-BirA*-flag (**Response Fig. 21b**). As expected, no signal was detected in the pull-down samples of cytoplasmic protein fractions treated with conditioned medium containing either Musclin-BirA* or Sp-BirA*-flag (**Response Fig. 21c**). We have incorporated these new results in **Supplementary Fig. 6b**.

Per the reviewer's suggestion, we have included the mass spectrometry data (after the database searches) of the BioID study in **Supplementary table 6**.

Response Fig. 21 Validation of the interaction between Musclin and Tfr1 by BioID assay. **a** Immunoblots of the conditioned medium of HEK293T cells transiently transfected with plasmids expressing indicated proteins. **b** Immunoblots of the input and streptavidin pull-down samples of the cell membrane-associated fractions. **c** Immunoblots of the input and streptavidin pull-down samples of the

cytoplasmic fractions.

Minor criticism:

At parts the manuscript would benefit from language editing.

Per the reviewer's suggestion, the language of the revised manuscript has been proofread and edited by Dr. Christopher R Wood, a native English speaker, from the College of Life Sciences at Zhejiang University. We have also acknowledged Dr. Christopher R Wood in the "Acknowledgement" section of the revised manuscript.

References

1. Moffatt, P. *et al.* Osteocrin is a specific ligand of the natriuretic Peptide clearance receptor that modulates bone growth. *J Biol Chem* **282**, 36454-62 (2007).
2. West, A.P., Jr. *et al.* Comparison of the interactions of transferrin receptor and transferrin receptor 2 with transferrin and the hereditary hemochromatosis protein HFE. *J Biol Chem* **275**, 38135-8 (2000).
3. Giannetti, A.M. *et al.* The molecular mechanism for receptor-stimulated iron release from the plasma iron transport protein transferrin. *Structure* **13**, 1613-23 (2005).
4. Kleven, M.D., Jue, S. & Enns, C.A. Transferrin Receptors TfR1 and TfR2 Bind Transferrin through Differing Mechanisms. *Biochemistry* **57**, 1552-1559 (2018).
5. Thomas, G. *et al.* Osteocrin, a novel bone-specific secreted protein that modulates the osteoblast phenotype. *J Biol Chem* **278**, 50563-71 (2003).
6. Nishizawa, H. *et al.* Musclin, a novel skeletal muscle-derived secretory factor. *J Biol Chem* **279**, 19391-5 (2004).
7. Zhou, R. *et al.* Endocrine role of bone in the regulation of energy metabolism. *Bone Research* **9**, 25 (2021).
8. Kwok, K.H.M., Lam, K.S.L. & Xu, A. Heterogeneity of white adipose tissue: molecular basis and clinical implications. *Experimental & Molecular Medicine* **48**, e215-e215 (2016).
9. Luong, Q., Huang, J. & Lee, K.Y. Deciphering White Adipose Tissue Heterogeneity. *Biology* **8**, 23 (2019).
10. Kita, S. *et al.* Competitive binding of musclin to natriuretic peptide receptor 3 with atrial natriuretic peptide. *J Endocrinol* **201**, 287-95 (2009).
11. Zois, N.E. *et al.* Natriuretic peptides in cardiometabolic regulation and disease. *Nat Rev Cardiol* **11**, 403-12 (2014).
12. Perez-Ternero, C. *et al.* C-type natriuretic peptide is a pivotal regulator of metabolic homeostasis. *Proceedings of the National Academy of Sciences of the United States of America* **119**, e2116470119 (2022).
13. Levy, J.E., Jin, O., Fujiwara, Y., Kuo, F. & Andrews, C.N. Transferrin receptor is necessary for development of erythrocytes and the nervous system. *Nature Genetics* **21**, 396-399 (1999).
14. Hentze, W.M., Muckenthaler, U.M. & Andrews, C.N. Balancing Acts: Molecular Control of Review Mammalian Iron Metabolism. *Cell* **117**, 285-297 (2004).
15. Senyilmaz, D. *et al.* Regulation of mitochondrial morphology and function by stearylolation of TFR1. *Nature* **525**, 124-8 (2015).
16. Li, J. *et al.* Transferrin Receptor 1 Regulates Thermogenic Capacity and Cell Fate in

- Brown/Beige Adipocytes. *Adv Sci (Weinh)* **7**, 1903366 (2020).
17. Qiu, J. *et al.* Transferrin Receptor Functionally Marks Thermogenic Adipocytes. *Frontiers in Cell and Developmental Biology* **8**, 572459 (2020).
 18. Zhang, Z. *et al.* Adipocyte iron levels impinge on a fat-gut crosstalk to regulate intestinal lipid absorption and mediate protection from obesity. *Cell Metab* **33**, 1624-1639 e9 (2021).
 19. Subbotina, E. *et al.* Musclin is an activity-stimulated myokine that enhances physical endurance. *Proc Natl Acad Sci U S A* **112**, 16042-7 (2015).
 20. Sternberg, E.A. *et al.* Identification of upstream and intragenic regulatory elements that confer cell-type-restricted and differentiation-specific expression on the muscle creatine kinase gene. *Mol Cell Biol* **8**, 2896-909 (1988).
 21. Bothe, G.W., Haspel, J.A., Smith, C.L., Wiener, H.H. & Burden, S.J. Selective expression of Cre recombinase in skeletal muscle fibers. *Genesis* **26**, 165-6 (2000).
 22. Szaroszyk, M. *et al.* Skeletal muscle derived Musclin protects the heart during pathological overload. *Nature Communications* **13**, 149 (2022).
 23. Kattih, B. *et al.* Low Circulating Musclin is Associated With Adverse Prognosis in Patients Undergoing Transcatheter Aortic Valve Implantation at Low-Intermediate Risk. *J Am Heart Assoc* **11**, e022792 (2022).

REVIEWER COMMENTS

Reviewer #1 (Remarks to the Author):

The authors addressed the reviewer's comments satisfactorily.

Reviewer #2 (Remarks to the Author):

I have no additional comments.
Congratulations for this manuscript

Reviewer #3 (Remarks to the Author):

Lu Jin et al. improved their manuscript and responded to my points of criticism, although some critical points remain with regard to the points that I had previously raised.

-Response Fig. 13/14: This data stands in contrast with multiple previous studies (Subbotina E et al., 2015; Szaroszyk M et al., 2022; Miyazaki T et al., 2018), which all showed perturbation of ANP and/or CNP in response to Musclin knock-out or overexpression, and therefore has to be addressed in more detail as the authors have currently done this. I suggest that plasma ANP and CNP should also be measured in Musclin knock-out and overexpressing mice exposed to HFD and /or cold exposure. In addition, cGMP responses in cultured adipocytes to Musclin + increasing amounts of ANP (or CNP) should be measured, in order to verify the authors' claim that Musclin does not act via enhancing cGMP signaling in contrast to other tissues. A positive control (e.g. HUVECs) should be included. See for example Figure 3B in Miyazaki T et al., 2018. Data should also be included in the manuscript as Supplemental material, and should not only be shown to the reviewer.

-Response Figure 16 and 17 should be included in the manuscript (e.g. as supplemental Figure).

Reviewer #4 (Remarks to the Author):

The authors have done a great job on their revision. The manuscript should now be acceptable for publication.

Responses to the reviewers' comments (original comments are in blue)

We are pleased that all four reviewers found our manuscript had been significantly improved and had addressed most of the reviewers' original comments and concerns. We appreciate the insightful comments and constructive suggestions on additional studies from Reviewer #3 to further improve our manuscript. In this revision, we have performed all the studies suggested by Reviewer #3, included the new data in **Supplementary Figures 3a, 6d, and 8a-j**, and added more discussion in the text to carefully address all the remaining concerns from the reviewers. Alterations of the text in the manuscript have been highlighted in red.

Point-to-point responses to the reviewers' comments are detailed below.

Reviewer #1 (Remarks to the Author):

The authors addressed the reviewer's comments satisfactorily.

We really appreciate the reviewer's enthusiasm and positive comments on our study.

Reviewer #2 (Remarks to the Author):

I have no additional comments.
Congratulations for this manuscript.

We really appreciate the reviewer's encouragement and positive comments on our work.

Reviewer #3 (Remarks to the Author):

Lu Jin et al. improved their manuscript and responded to my points of criticism, although some critical points remain with regard to the points that I had previously raised.

-Response Fig. 13/14: This data stands in contrast with multiple previous studies (Subbotina E et al., 2015; Szaroszyk M et al., 2022; Miyazaki T et al., 2018), which all showed perturbation of ANP and/or CNP in response to Musclin knock-out or overexpression, and therefore has to be addressed in more detail as the authors have currently done this. I suggest that plasma ANP and CNP should also be measured in Musclin knock-out and overexpressing mice exposed to HFD and /or cold exposure. In addition, cGMP responses in cultured adipocytes to Musclin + increasing amounts of ANP (or CNP) should be measured, in order to verify the authors' claim that Musclin does not act via enhancing cGMP signaling in contrast to other tissues. A positive control (e.g. HUVECs) should be included. See for example Figure 3B in Miyazaki T et al., 2018. Data should also be included in the manuscript as Supplemental material, and should not only be shown to the reviewer.

We appreciate the reviewer's insightful comments and constructive suggestions on experiments to further improve our work. Per the reviewer's suggestion, we have measured the plasma ANP and CNP levels in Musclin-MKO and MCK-Musclin mice and their respective control mice in response to HFD feeding and /or cold exposure. It has been reported that Musclin transgenic mice exhibited a significant increase in plasma CNP levels at 28 days after myocardial infarction¹. Consistently, a recent study by Szaroszyk M et al. also demonstrated that plasma CNP levels were significantly elevated in mice with AAV6-mediated overexpression of Musclin in skeletal muscle at 3 weeks after transverse aortic constriction (TAC) surgery, as compared to control mice². Conversely, plasma CNP levels were lower in muscle-specific Musclin deficient mice versus control mice after TAC surgery². However, plasma ANP levels were either trending lower or higher in Musclin overexpressing mice after myocardial infarction¹ or TAC surgery², respectively. Moreover, it has been reported that plasma ANP levels were trending lower, but did not reach statistical significance ($P = 0.09$), in Musclin whole-body KO mice compared to WT controls following exercise training³. These results revealed an overall modest effect of Musclin overexpression or deficiency on plasma ANP levels in mice. Consistent with these findings, we observed that plasma ANP levels remained largely unchanged in MCK-Musclin mice compared to control mice in response to cold exposure or HFD feeding (**Supplementary Fig. 8a, b**). Similarly, plasma levels of ANP were similar between Musclin^{fllox/fllox} and Musclin-MKO mice under either cold exposure or HFD feeding conditions (**Supplementary Fig. 8a, b**). Notably, plasma CNP levels were significantly increased in MCK-Musclin mice compared to controls under either cold exposure or HFD feeding conditions (**Supplementary Fig. 8c, d**). In contrast, plasma CNP levels were lower in Musclin-MKO mice than Musclin^{fllox/fllox} in response to either cold exposure or HFD feeding conditions (**Supplementary Fig. 8c, d**).

The cyclic GMP (cGMP), generated by the guanylyl cyclase domain of NPRA (receptor for ANP and BNP) and NPRB (receptor for CNP), is a key mediator of the natriuretic peptide-associated biological responses⁴. It has been reported that Musclin could enhance cardiomyocyte cAMP generation through cGMP-mediated inhibition of cAMP-degrading phosphodiesterase 3 (PDE3)². Per the reviewer's suggestion, we have measured cGMP responses in cultured adipocytes treated with Musclin plus increasing amounts of ANP or CNP, using cellular cGMP response in HUVECs as a positive control. Similar to the data shown in Figure 3B in the previous study by Miyazaki T et al.¹, we observed that Musclin (10^{-5} M) could significantly increase the intracellular cGMP levels in the presence of 10^{-8} M ANP in HUVECs (**Supplementary Fig. 8e**). Moreover, treatment with 10^{-6} M ANP alone also elicited an increase in intracellular cGMP levels compared to 10^{-8} M ANP treatment alone (**Supplementary Fig. 8e**). Previous studies showed that both ANP and CNP could increase the intracellular cGMP levels in adipocytes^{5,6}. Consistently, we also observed that both ANP (**Supplementary Fig. 8f**) and CNP (**Supplementary Fig. 8g**) elevated intracellular cGMP levels in differentiated C3H10T1/2-derived adipocytes in a dose-dependent manner. However, co-treatment with Musclin exhibited no obvious effect on intracellular cGMP levels in response to increasing amounts of ANP or CNP (**Supplementary Fig. 8f, g**).

To further examine the involvement of cGMP signaling in the regulation of adipocyte thermogenic metabolism, we also measured the cGMP levels in iWAT and BAT from control and MCK-Musclin mice in response to either cold exposure or HFD feeding, as well as in iWAT from Musclin^{flox/flox} and Musclin-MKO mice upon HFD feeding. As shown in **Supplementary Fig. 8h-j**, cGMP levels in iWAT or BAT remained largely unchanged after manipulation of Musclin in mice upon HFD feeding or cooling. Taken together, these data indicate that the cGMP/PDE3 signaling pathway might not be involved in the regulation of adipocyte thermogenic metabolism by Musclin. According to the reviewer's suggestion, these new data have been included in **Supplementary Fig. 8**. More thorough discussion of these new results was also added to the "Discussion" section in the revised manuscript.

-Response Figure 16 and 17 should be included in the manuscript (e.g. as supplemental Figure).

Per the reviewer's suggestion, we have included the data shown in **Response Figures 16 and 17** in **Supplementary Fig. 6d** and **Supplementary Fig. 3a**, respectively, in the revised manuscript.

Reviewer #4 (Remarks to the Author):

The authors have done a great job on their revision. The manuscript should now be acceptable for publication.

We are deeply grateful to the reviewer for the encouragement and positive comments on our work.

References

- 1 Miyazaki, T. *et al.* A New Secretory Peptide of Natriuretic Peptide Family, Osteocrin, Suppresses the Progression of Congestive Heart Failure After Myocardial Infarction. *Circ Res* **122**, 742-751, doi:10.1161/circresaha.117.312624 (2018).
- 2 Szaroszyk, M. *et al.* Skeletal muscle derived Musclin protects the heart during pathological overload. *Nature Communications* **13**, 149, doi:10.1038/s41467-021-27634-5 PMID - 35013221 (2022).
- 3 Subbotina, E. *et al.* Musclin is an activity-stimulated myokine that enhances physical endurance. *Proc Natl Acad Sci U S A* **112**, 16042-16047, doi:10.1073/pnas.1514250112 (2015).
- 4 Zois, N. E. *et al.* Natriuretic peptides in cardiometabolic regulation and disease. *Nat Rev Cardiol* **11**, 403-412, doi:10.1038/nrcardio.2014.64 (2014).
- 5 Bae, C. R. *et al.* Adipocyte-specific expression of C-type natriuretic peptide suppresses lipid metabolism and adipocyte hypertrophy in adipose tissues in mice fed high-fat diet. *Sci Rep* **8**, 2093, doi:10.1038/s41598-018-20469-z (2018).

- 6 Katafuchi, T., Garbers, D. L. & Albanesi, J. P. CNP/GC-B system: a new regulator of adipogenesis. *Peptides* **31**, 1906-1911, doi:10.1016/j.peptides.2010.06.025 (2010).

Point-to-point responses to the reviewers' comments are detailed below.

Reviewer #1 (Remarks to the Author):

The authors addressed the reviewer's comments satisfactorily.

We really appreciate the reviewer's enthusiasm and insightful comments on our study, which have significantly expanded the scope and improved the quality of our work.

Reviewer #2 (Remarks to the Author):

I have no additional comments.
Congratulations for this manuscript.

We really appreciate the reviewer's encouragement and positive comments on our study.

Reviewer #3 (Remarks to the Author):

We are pleased to know that the reviewer has no additional comments, and that we have addressed all the reviewer's original comments and concerns. We are deeply thankful to the reviewer for the enthusiasm and insightful comments on expanding the scope and impact of our original manuscript.

Reviewer #4 (Remarks to the Author):

The authors have done a great job on their revision. The manuscript should now be acceptable for publication.

We are deeply grateful to the reviewer for the encouragement and positive comments on our work.